# Evolutionary characterization of lung cancer metastasis

Sonya Hessey[1,2,3,4,139], Abigail Bunkum[1,2,3,139], Ariana Huebner[1,5,6,139], Kerstin Haase[1,2,139], Kristiana Grigoriadis[1,5,6,139], Cristina Naceur-Lombardelli[1,139], Wing Kin Liu[1,2], Caitlin F. Harrigan[7,8,9], Charlotte Grieco[1,6], Daniele Marinelli[2,5,10], Boyue Ding[11], Carlos Martínez-Ruiz[1,6], Piotr Pawlik[1,5], Mark S. Hill[6], Olivia Lucas[1,3,4,6], Corentin Richard[1,2], Oriol Pich[6], Kerstin Thol[1,5], Takahiro Karasaki[1,2,6,12], Sophia Ward[1,6,13], Foteini Athanasopoulou[1,6,13], Monica Sivakumar[1], Selvaraju Veeriah[1], Antonia Toncheva[1], Andrew J. Rowan[6], Paulina Prymas[1], Hayley Bridger[14], Miriam Mitchison[15], Elaine Borg[15], Mary Falzon[15], Ian Proctor[15], Ula Mahadeva[16], Anna Green[16], Martin D. Forster[1,4], Sarah Benafif[4], Tanya Ahmad[4], Siow Ming Lee[1,4], Dionysis Papadatos-Pastos[4], Babu Naidu[17], Gerald Langman[18], Matthew G. Krebs[19], Pedro Oliveira[20], Fiona H. Blackhall[19], Yvonne Summers[19], Jamie Weaver[19], John Le Quesne[21,22,23], Anne Thomas[24], Cathy Richards[25,26], Dean A. Fennell[26,27], Sanjay Jogai[28], Judith Cave[29], Patricia Roxburgh[22,30], Sioban Fraser[21,23], Alan Kirk[31], Kevin G. Blyth[21,22,32], Peter Russell[33], Crispin T. Hiley[1,6], Allan Hackshaw[14], TRACERx Consortium*, TRACERx EVO Consortium*, PEACE consortium*, David A. Moore[1,6,15], Simone Zaccaria[1,3,140✉], Nicholas McGranahan[1,5,140✉], Charles Swanton[1,4,6,140✉] & Mariam Jamal-Hanjani[1,2,4,140✉]

Limited understanding of the biological processes that govern metastatic dissemination hinders its prevention and treatment[1]. Here, using 501 longitudinally collected primary and metastatic tumour samples from 24 patients with non-small cell lung cancer (NSCLC) enrolled in the TRACERx lung study and PEACE autopsy programme, we infer tumour evolution from diagnosis to death. With DNA-sequencing data encompassing 70% of the metastases that were radiologically detected before death and paired multi-region sampled primary tumours, we show that the genomes of metastases diverge markedly from those of their ancestral primary tumour, with additional driver alterations and genome doubling events occurring after metastatic dissemination. In 62.5% of patients, multiple primary tumour subclones disseminated, each founding a distinct metastasis. These metastases served as sources of onward spread: more than half of the metastases sampled were seeded by other metastases. The duration that metastases existed in situ influenced their likelihood of seeding further metastases. Most metastatic migrations started and ended in the same anatomical cavity. The few subclones that exited the thorax to seed metastases disseminated widely and were enriched for somatic copy-number alterations, suggesting that chromosomal instability may facilitate extrathoracic spread. This spatial and temporal evolutionary analysis sheds light on the extent of metastatic diversity and seeding in advanced NSCLC—which tends to be underestimated in single metastasis biopsies—and identifies genomic and clinical mediators of metastatic progression.

Metastasis, the process by which cancer cells spread from their site of origin to a secondary location, is the leading cause of cancer-related mortality[2]. Lung cancer accounts for the largest share of metastatic cancer cases[3], owing to its high incidence[4], frequent presentation as de novo metastatic disease[5] and high rate of relapse after curative-intent surgery for localized disease[6]. Understanding the genetic basis for the mechanisms that enable cancer cells to migrate from the primary site into local tissues, air spaces, lymphatics and/or circulation, and to 'seed' foreign 'soils'[7], could inform strategies to prevent and treat this lethal condition[1].

The migration of cancer cells is a transient event that is difficult to observe in patients in real time; however, a retrospective view of this process can be gleaned by tracking the evolutionary history of subpopulations of cancer cells, or subclones, in primary tumours and metastases using DNA-sequencing data[8–12]. With this approach, studies of ovarian[9], breast[13,14] and prostate cancer[15] have revealed complex metastatic migration patterns that involve multiple tumour subclones migrating bidirectionally between the primary tumour and metastases, and between anatomically distinct metastatic sites. The accuracy and completeness of this view of metastasis is contingent

A list of affiliations appears at the end of the paper. *Lists of authors and their affiliations appear online. ✉e-mail: s.zaccaria@ucl.ac.uk; nicholas.mcgranahan.10@ucl.ac.uk; charles.swanton@crick.ac.uk; m.jamal-hanjani@ucl.ac.uk

on the extent to which the sampled metastases are representative of a patient's disease burden, and the availability of the primary tumour for comparison[16]. Despite its importance, such comprehensive and longitudinal sampling is rarely performed because of its clinical infeasibility in living patients, but it can be achieved by integrating research autopsy programmes[16,17] with prospective clinical studies such as TRACERx (Tracking Non-small Cell Lung Cancer Evolution through Therapy (Rx); ClinicalTrials.gov identifier NCT01888601)[18,19], which performs multi-region profiling of early-stage, operable primary NSCLC.

To address this need, the national, multi-centre, pan-cancer research autopsy programme PEACE (Posthumous Evaluation of Advanced Cancer Environment; NCT03004755) was strategically embedded in centres that recruited patients to TRACERx to enable co-enrolment to both studies and the generation of a clinically annotated tumour tissue resource that spans the complete disease course, from diagnosis to death.

In this study, we reconstructed detailed tumour evolutionary histories and metastatic migration patterns using high-depth whole-exome sequencing (WES) data from longitudinally collected primary tumour, pre-mortem and post-mortem metastasis samples from 24 patients enrolled in both TRACERx and PEACE (Supplementary Fig. 1), to investigate the genetic properties that endow cancer cells with the capacity to metastasize. By integrating migration patterns with serial radiological imaging performed before death, we uncover tumour-intrinsic, temporal and anatomical properties that govern NSCLC metastasis.

## The TRACERx–PEACE cohort

The clinical characteristics of the patients in this cohort, including age (median [range]: 70 [47–87] years), smoking status (19 ex-smokers, 3 current smokers and 2 never-smokers), disease-free survival (DFS; median [interquartile range; IQR]: 11 [6–18] months) and overall survival (OS; median [IQR]: 29 [16–46] months), were broadly representative of patients with comparable stages of operable NSCLC (TNM version 8: 4 stage I, 8 stage II and 12 stage III)[19]. Common NSCLC histological subtypes were represented (9 lung adenocarcinomas (LUAD) and 10 squamous cell carcinomas (LUSC)), and 5 other subtypes, including 2 large cell carcinomas, 2 pleomorphic carcinomas and 1 carcinosarcoma, were present (Supplementary Table 1).

In total, 108 regions from 24 resected primary tumours (median regions per primary [range]: 4 [2–8]), 41 regions from 35 metastases sampled pre-mortem (12 lymph node metastases resected during primary surgery from 7 patients, 17 metastases sampled at relapse from 15 patients and 6 at disease progression from 4 patients) and 352 regions from 233 anatomically distinct metastases collected at autopsy were subjected to WES (median depth: 401.2, IQR: 360.2–441.5, Fig. 1a) and passed quality control (Supplementary Fig. 1 and Methods). At autopsy, metastatic sampling was guided by radiological imaging performed before death and macroscopic examination by the attending pathologist. For most metastases collected pre-mortem or at autopsy, a single metastasis region was sampled (75%, 200/268), but in 25% (68/268), multiple regions from the same metastasis were sampled. The mean number of metastasis regions collected per patient was 16 (range: 3–39) and the mean number of anatomically distinct metastases sampled per patient was 11 (range: 2–37). These encompassed 19 anatomical locations, including common metastasis sites observed in NSCLC[20]: lung (125 regions), lymph node (80 regions), liver (33 regions), musculoskeletal soft tissues (23 regions: 13 chest wall, 3 diaphragm, 2 abdominal wall, 5 other), brain (22 regions), adrenal gland (17 regions) and bone (7 regions) (Fig. 1a). For the 23 patients with available radiological imaging, quality-controlled WES data were available for 70% (112/160) of the metastases that were detected with imaging performed before death (Fig. 1b).

## Intra- and inter-metastasis genetic heterogeneity

To resolve the extent of metastatic heterogeneity and the degree to which metastases resemble the primary tumour from which they originate, we performed a detailed genomic analysis of somatic mutations, somatic copy-number alterations (SCNAs) and whole-genome doubling (WGD) in primary and metastasis regions, and reconstructed the subclonal architecture and phylogenetic history of each patient's disease (Supplementary Fig. 2 and Methods). Subclones were classified into four groups to delineate when they arose relative to metastatic dissemination: truncal (the most recent common ancestor (MRCA) of all sequenced cancer cells); primary-unique (non-truncal subclones present in the primary tumour and undetected in any metastasis); metastasis-unique (non-truncal subclones present in one or more metastases and undetected in the primary tumour); and shared subclonal (non-truncal subclones present in both the primary and one or more metastases).

We detected subclonal diversity within individual metastases (median [range]: 7 [3–37] subclones per metastasis) and between anatomically distinct metastases (79% of metastases contained a subclone that was not detected in any other metastasis). The number of subclones detected per metastasis increased with the number of metastasis regions sampled (Pearson's $R$: 0.6, $P = 7.64 \times 10^{-27}$; Extended Data Fig. 1a), implying that, as with primary NSCLC tumours[18,19,21], single-region sampling can underestimate the subclonal diversity in a metastasis. The number of metastasis-unique subclones identified increased with the number of anatomically distinct metastases sampled per patient (Pearson's $R$: 0.52, $P = 0.01$; Extended Data Fig. 1a), highlighting that samples from anatomically distinct metastases are needed to detect the subclones variably distributed across metastatic sites. The number of metastasis-unique subclones identified per patient in this cohort (median [range]: 28 [4–58]) was 11-fold greater than it was in our previous analysis of metastases sampled at the time of primary surgery or relapse in 126 patients enrolled in TRACERx[11] (median [range]: 2.5 [0–13]); the mean number of metastasis regions sampled per patient was 1.7 (range: 1–6) in that study, compared with 16 (range: 3–39) in this study (Supplementary Fig. 3).

We used two metrics based on either somatic mutations or SCNAs to quantify the genetic diversity between anatomically distinct metastases (inter-metastasis heterogeneity), within individual metastases (intra-metastasis heterogeneity) and between the primary tumour and metastases from the same patient (primary–metastasis heterogeneity; Methods). Metastases from the same patient were more similar to each other than they were to the paired primary (mean inter-metastasis versus mean primary–metastasis heterogeneity per patient: mutation diversity $P = 0.004$, SCNA diversity $P = 6.0 \times 10^{-5}$, Wilcoxon signed-rank test; Fig. 1c), as observed in prostate cancer[15]. Primary–metastasis heterogeneity was positively associated with both intra-primary heterogeneity (mean per patient: mutation diversity Pearson's $R$: 0.57, $P = 0.005$; SCNA diversity Pearson's $R$: 0.51, $P = 0.013$; Extended Data Fig. 1b) and intra-metastasis heterogeneity (mean per patient: mutation diversity Pearson's $R$: 0.55, $P = 0.01$; SCNA diversity Pearson's $R$: 0.33, $P = 0.15$; Extended Data Fig. 1b), suggesting that somatic evolution in the primary tumour, metastases, or both after metastatic dissemination contributes to this genetic divergence. The degree of primary–metastasis divergence did not differ according to the treatment patients received (Supplementary Fig. 4), although such inferences might be limited owing to cohort size. Individual multi-region sampled metastases were less heterogeneous than their paired multi-region sampled primary (mean intra-metastasis versus mean intra-primary heterogeneity per patient: mutation diversity $P = 0.033$, SCNA diversity $P = 0.004$, Wilcoxon signed-rank test; Fig. 1c), but when all metastases sampled from a patient were considered, the heterogeneity among them was not significantly different to that within the paired primary tumour (mean inter-metastasis

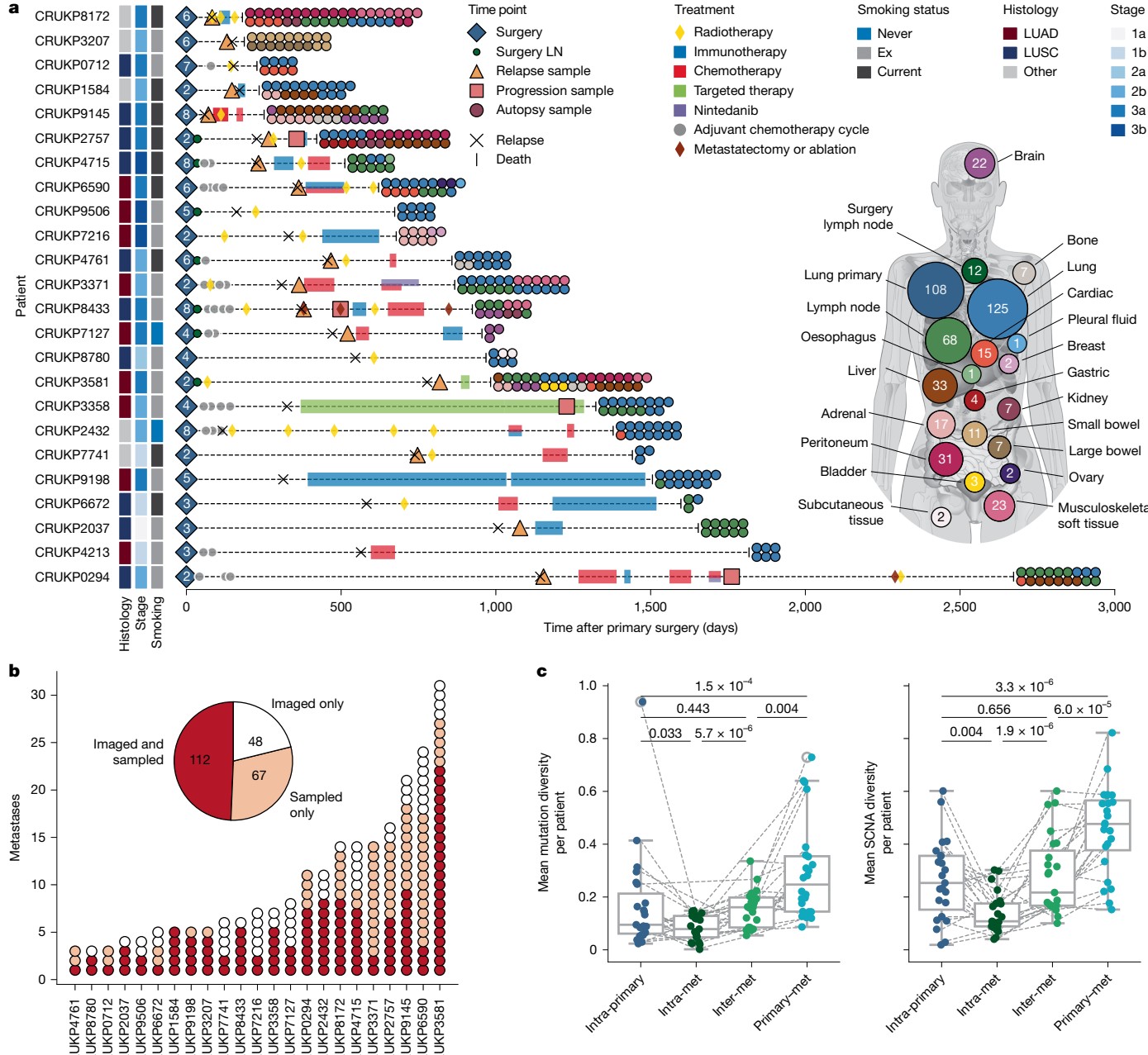

**Fig. 1 | Clinical and sample characteristics of the TRACERx–PEACE cohort.**
**a**, Longitudinal timelines for 24 patients showing clinical events and sampling time points, ordered by overall survival. Tumour histology, stage and smoking status are annotated. The body map summarizes the total number of samples obtained per organ across the cohort. LN, lymph node. **b**, Number of metastases per patient that were imaged only (white), sampled only (peach) or both imaged and sampled (red) among 23 patients with radiological imaging available. **c**, Mean genetic diversity (mutations and SCNAs; Methods) per patient within primary tumours (intra-primary, dark blue; n = 23 patients, 100 regions), within

multi-region sampled metastases (intra-metastasis, dark green; n = 21 patients, 191 regions), between anatomically distinct metastases (inter-metastasis, green; n = 24 patients, 258 metastases) and between primary and metastatic samples (primary–metastasis, blue; n = 24 patients, 24 primaries–258 metastases). Lines connect patients. Wilcoxon signed-rank test. The box plots show the median and IQR with whiskers denoting values within 1.5 times the IQR from the first and third quartiles. Body map illustration in **a** by J. Brock adapted from ref. 11 under a Creative Commons licence CC BY 4.0.

versus mean intra-primary heterogeneity per patient: mutation diversity $P = 0.443$, SCNA diversity $P = 0.656$, Wilcoxon signed-rank test; Fig. 1c).

These data are consistent with primary tumours and metastases continuing to evolve after they diverge. As such, resected primary NSCLC tumours are unlikely to be representative of metastases detected during clinical follow up[22], and, as in other cancer types[9,15,23], the full extent of metastatic heterogeneity is likely to be underestimated when metastatic sampling is limited.

## Mutational processes change over time

Somatic mutations arise from a variety of mutational processes. To assess their contribution to temporal and spatial genomic heterogeneity, we evaluated mutational signatures known to be active in NSCLC across truncal, shared subclonal, primary-unique and metastasis-unique subclones (Methods). Smoking and APOBEC signatures were more prevalent in primary than in metastasis-unique subclones. This was evident when considering the majority aetiology

of each subclone (the mutational process that constituted more than 50% of mutations; Extended Data Fig. 1c and Methods), or the number of subclones in which each mutational process was detected (average percentage of subclones with smoking signature: primary 21% versus metastasis 7%, $P = 0.02$; average percentage of subclones with APOBEC signature: primary 54% versus metastasis 34%, $P = 0.0045$; Wilcoxon signed-rank test; Extended Data Fig. 1d and Methods). In ever-smokers (patients who had smoked 100 or more cigarettes during their lifetime), 83.4% of all mutations attributed to the smoking signature SBS4 occurred in the trunk, consistent with SBS4 being an early process in NSCLC evolution[21]. The lower prevalence of SBS4 in metastases might also reflect a lack of exposure owing to smoking cessation (80% of patients were ex-smokers) or that not all organs are equally exposed[24]. APOBEC activity was highest in shared subclonal and primary-unique subclones (Extended Data Fig. 1c,d). Episodic APOBEC activity was observed throughout tumour evolution[25], including in metastases, characterized by fluctuating levels of APOBEC mutagenesis along phylogenetic branches in 39% (9/23) of patients with APOBEC activity (Extended Data Fig. 1e).

Among metastasis-unique subclones, the majority aetiologies, in order of prevalence, were clock-like (SBS1 and SBS5), APOBEC (SBS2 and SBS13), platinum (SBS31 and SBS35) and smoking (SBS4) (Extended Data Fig. 1c). Platinum-related signatures were detected in metastasis-unique subclones in 64% (9/14) of patients treated with platinum chemotherapy. None of these mutational processes were site-specific: each occurred in multiple metastatic sites (Extended Data Fig. 1f and Supplementary Fig. 5). Mutational signature profiles of metastasis-unique subclones were more similar to each other than they were to ancestral primary subclones ($P = 0.0043$, Wilcoxon signed-rank test; Extended Data Fig. 1g), suggesting that the observed temporal shifts in mutational process activity contribute to primary–metastasis divergence.

## Putative driver alterations occur in metastases

To investigate genetic alterations that might underpin somatic evolution in metastases, we assessed the frequency of known genetic drivers of tumorigenesis. Annotating somatic mutations (Methods), 196 driver mutations were identified: 174 single-nucleotide variants (SNVs), 4 dinucleotide variants (DNVs), and 18 insertion–deletions (indels), of which 70% affected tumour suppressor genes (TSGs) (Fig. 2a). Overall, 49% (97/196) were truncal and thus shared between all primary and metastasis regions, consistent with studies in other cancer types[23,26–29] (Fig. 2b). *KRAS* was the most frequently mutated oncogene (21% (5/24) of patients) and was always truncal. Of the driver mutations, 27% (53/196) were metastasis-unique (detected in metastases, but not in the primary tumour). Most patients (83%, 20/24) had at least one metastasis-unique driver mutation (median [range]: 2 [0–6] per patient; Fig. 2a). The number of metastasis-unique drivers per patient correlated with the total number of metastasis-unique mutations (Pearson's $R = 0.8$, $P = 3.2 \times 10^{-6}$; Fig. 2c), and both were associated with the duration of chemotherapy treatment (Extended Data Fig. 2a). Metastasis-unique subclones with platinum signature activity were more likely to contain a metastasis-unique driver than were their counterparts without ($P = 0.016$, chi-squared test; Fig. 2d), suggesting that in addition to the greater number of drivers that result from treatment-related mutagenesis, treatment may select for or induce these mutations.

Next, we used OncoKB[30,31] to determine whether actionable drivers occur uniquely in metastases (Fig. 2b). No level 1 actionable mutations, which predict response to a drug licensed for NSCLC management, were detected in metastasis-unique subclones. Only one metastasis-unique mutation was annotated as potentially actionable: *SMARCA4* (p.E1083K), identified in a lung metastasis in patient CRUKP3358, was classified as a level 3A mutation (a biomarker for an unlicensed drug with efficacy in clinical trials[32]).

We also examined the timing of focal amplifications and loss of heterozygosity (LOH) affecting cancer-related oncogenes and TSGs, respectively (Methods). Among TSG driver mutations, 46% (63/138) had biallelic inactivation, defined as a mutation affecting one allele with LOH affecting the other (Methods and Fig. 2a). In 76% (48/63) of these cases, both events were detected in the primary tumour; 21% (13/63) occurred sequentially, with the first hit in the primary and the second in a metastasis; and in 3% (2/63) of cases, both events occurred in a metastasis. For example, CRUKP2037 had a truncal driver mutation affecting *ARID1A* (N942S), followed by LOH of the *ARID1A* locus within a lymph node metastasis, as well as a truncal LOH affecting *B2M*, followed by a subclonal *B2M* driver mutation, both in the primary tumour (Extended Data Fig. 2b). The most frequently mutated TSG was *TP53* (75% (18/24) of patients), which is associated with metastatic seeding[11,33]. It was biallelically inactivated in 89% (16/18) of patients: in most cases both hits were truncal (88% (14/16)), whereas in the others, subclonal LOH followed a truncal driver mutation, including on parallel phylogenetic branches (CRUKP8172).

WGD, which is often associated with *TP53* disruption[34] and can mitigate against deleterious alterations[35], occurred in 92% (22/24) of patients. These events occurred throughout tumour evolution (76% (31/41) occurred in the primary—11 truncal, 11 shared subclonal and 9 primary-unique—and 24% (10/41) occurred in metastases; Extended Data Fig. 2c), and were associated with increased primary–metastasis (Pearson's $R$: 0.57, $P = 0.0034$) and inter-metastasis SCNA heterogeneity (Pearson's $R$: 0.67, $P = 0.00051$; Extended Data Fig. 2d). Primary subclonal and metastasis-unique WGD events occurred on parallel phylogenetic branches in 29% (7/24) of patients, suggesting that they confer a fitness advantage in late-stage disease.

Overall, although most driver alterations, including those that are clinically actionable, occurred in primary tumours, metastases accrued additional—often treatment-associated—genetic alterations of potential biological consequence.

## Pervasive metastasis-to-metastasis seeding

In addition to ongoing evolution in metastases, cancer cells migrating between anatomical sites will influence the subclonal landscape of metastatic disease. To elucidate the metastatic migration patterns that underpin advanced NSCLC, we applied the MACHINA algorithm[8] to the phylogenies inferred for each patient to identify the subclones that seeded metastases and their corresponding migration routes (Extended Data Fig. 3).

In 3 patients, seeding subclones originated exclusively in the primary tumour. In the remaining 88% (21/24), seeding subclones were identified in both the primary tumour (mean per patient [range]: 2.8 [1–8]) and one or more metastases (mean per patient [range]: 6.0 [1–16]), herein referred to as primary-to-metastasis seeding subclones and metastasis-to-metastasis seeding subclones, respectively (Fig. 3a). Metastasis-to-primary reseeding was not considered because no patients had radiologically detectable metastatic disease at the time of primary tumour resection. On average, 4 anatomically distinct metastases were seeded by the primary tumour per patient (range: 1–22). In 62.5% (15/24) of patients, these primary-seeded metastases were seeded by distinct primary-to-metastasis seeding subclones, in contrast with previous studies that detected only a single primary seeding subclone in most patients[9,11,36]. The number of seeding subclones identified per patient correlated with the number of anatomically distinct metastases (Pearson $R$: 0.48, $P = 0.018$) and the number of metastasis regions sampled (Pearson $R$: 0.52, $P = 0.009$; Extended Data Fig. 5a), indicating that seeding subclone prevalence can be underestimated when metastasis sampling is limited. Overall, however, a greater number of metastases were seeded by other metastases than by the primary tumour: 60% (156/258) of metastases were seeded by metastasis-to-metastasis seeding subclones, 38% (98/258) were seeded by primary-to-metastasis

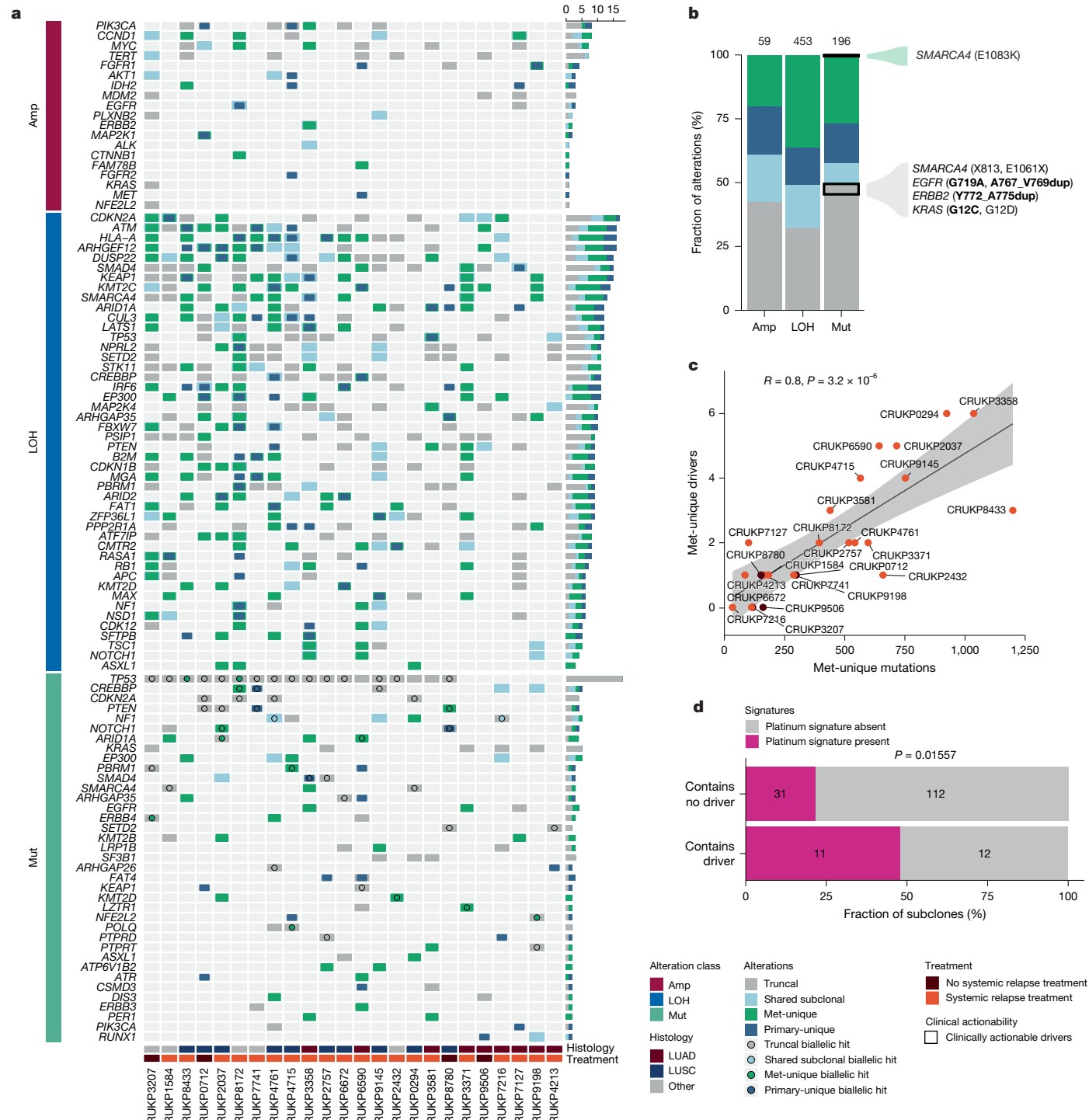

**Fig. 2 | Evolutionary timing of putative driver alterations. a**, Summary of the focal copy-number amplifications (Amp) affecting oncogenes (top), focal LOH affecting TSGs (middle) and putative driver mutations (Mut: SNVs, indels and DNVs; bottom) across patients. Only mutations occurring in at least two patients or showing biallelic inactivation (Methods) in one patient are shown. Alterations are classified as truncal (grey), shared subclonal (light blue), primary-unique (dark blue) or metastasis-unique (green). Mutations that co-occur with LOH are marked with circles. Horizontal bars indicate total occurrences per gene and class. **b**, Alteration class distribution for events in **a**. Clinically actionable events are highlighted in boxes; bold font denotes biomarkers for approved lung cancer therapies (OncoKB level 1) and regular font denotes biomarkers with compelling clinical evidence (OncoKB level 3A). *n* = 24 patients. **c**, Correlation between the number of metastasis-unique mutations and the number of metastasis-unique driver mutations per patient; points coloured by the relapse treatment status of each patient. **d**, Fraction of metastasis-unique subclones in platinum-treated patients (*n* = 14) with detectable platinum-associated mutational signatures, stratified by the presence or absence of a driver mutation. Chi-squared test.

seeding subclones and 2% (4/258) were seeded by both (10 low-purity metastases were excluded from tree building and migration analyses; Methods and Fig. 3b). For example, in CRUKP2037, only one of the six

thoracic lymph node metastases sampled at autopsy was seeded by the primary; the remaining five were seeded by subclones from other established thoracic lymph node metastases (Extended Data Fig. 3).

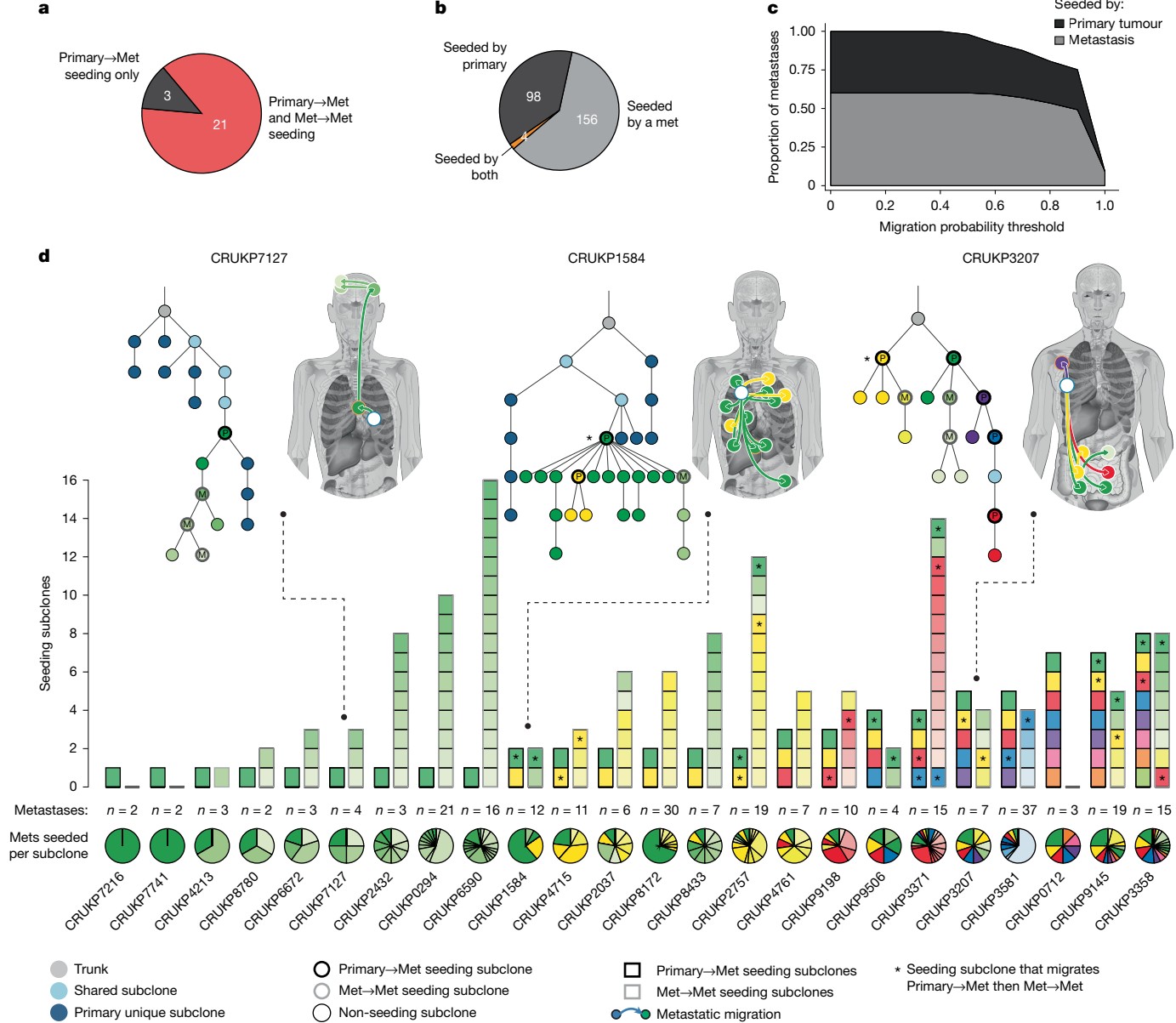

**Fig. 3 | Metastatic seeding patterns. a**, Number of patients whose metastases were seeded exclusively by the primary tumour (Primary->Met seeding only, black) or by both the primary tumour and other metastases (Primary->Met and Met->Met seeding, red). **b**, Number of metastases seeded by the primary tumour (black), by another metastasis (grey) or by both sources (yellow). **c**, Proportion of metastases seeded by other metastases (grey) or by the primary tumour (black) across increasing migration probability thresholds (Methods). **d**, Number of primary-to-metastasis (black outline) and metastasis-to-metastasis (grey outline) seeding subclones per patient. Each primary-to-metastasis seeding subclone is assigned a distinct colour (concordant between bars and phylogenetic

tree nodes). Metastasis-to-metastasis seeding subclone colours match the colour of their primary-to-metastasis seeding ancestor, with lighter shades assigned to each new metastasis-to-metastasis seeding subclone in that lineage. Pie wedges show the fraction of metastases seeded by each correspondingly coloured seeding subclone. Metastases on body maps are coloured by their seeding subclone, with arrows indicating migration routes. Subclones contributing to both primary-to-metastasis and metastasis-to-metastasis spread are marked with an asterisk. P, primary-to-metastasis seeding subclone; M, metastasis-to-metastasis seeding subclone. Body map illustration in **d** by J. Brock adapted from ref. 11 under a Creative Commons licence CC BY 4.0.

Consistent with studies in other cancer types[9,15,36], most metastases were seeded by a single migrating subclone (72.5%, 187/258), as opposed to multiple subclones that migrated together or in sequence (Extended Data Fig. 4). Primary-to-metastasis migrations involved a single subclone more frequently than metastasis-to-metastasis migrations ($P = 7.73 \times 10^{-4}$, chi-squared test; Supplementary Fig. 6a). Migrations that started and ended in the same organ involved multiple subclones more often (43.6%, 41/94) than migrations between organs (26.8%, 49/183, $P = 0.007$, chi-squared test; Supplementary Fig. 6b), implying that a greater number of subclones were capable of migrating within, as opposed to between, organs.

Three independent analyses provide orthogonal evidence for the inferred metastasis-to-metastasis migrations. First, we developed a probabilistic approach to assign a confidence level to each metastatic migration (Methods and Extended Data Fig. 5b). Metastasis-to-metastasis migrations remained predominant when considering only the migrations inferred with highest confidence (Fig. 3c). Second, we used standard-of-care radiological imaging to determine the first time each metastasis was radiologically detected. Metastases seeded by the primary were detected earlier than metastases seeded by other metastases (median 272 versus 745 days from surgery, respectively, $P = 2.41 \times 10^{-5}$, Mann–Whitney $U$ test; Extended Data Fig. 5c).

Third, we evaluated migration directionality by quantifying LOH events present in the seeding but not in the seeded metastasis—an implausible scenario given LOH irreversibility (Methods). Metastases linked by a metastasis-to-metastasis migration had significantly more conserved LOH events (which are not an input to MACHINA[8]) compared with alternative seeding sources (inferred metastasis source versus primary source $P = 3.29 \times 10^{-28}$; inferred metastasis source versus alternative metastasis source $P = 0.011$; Mann–Whitney $U$ test; Extended Data Fig. 5d). Furthermore, the metastatic migration patterns inferred were highly consistent with those obtained using different combinations of algorithms for phylogenetic[37] and metastatic migration inference[38] (Supplementary Fig. 7).

These patterns reveal that in NSCLC, multiple primary subclones seed metastases, initiating a cascade of metastasis-to-metastasis seeding that promotes metastatic progression.

## Seeding capacity across subclone lineages

To test whether metastatic capacity differs among seeding subclones, we used the number of metastases seeded by each seeding subclone as a surrogate measure of its seeding capacity (Methods). In 93% (14/15) of the patients with multiple primary-to-metastasis seeding subclones, no significant differences were observed in the number of metastases seeded by each subclone (Extended Data Fig. 5e). In the patient in which a difference in seeding capacity was detected, CRUKP8172, one of the 2 primary seeding subclones seeded 21 of the 30 sampled metastases (Monte Carlo likelihood ratio test $P = 0.00001$; Fig. 3d).

Despite the fact that most of the primary-to-metastasis seeding subclones had a similar metastatic capacity, only 45% (30/67) of them produced descendants that seeded additional metastases, raising the possibility that the capacity to seed metastases from the primary tumour differs from that required for metastasis-to-metastasis seeding. We therefore assessed whether subclones from the same primary tumour, after establishing metastases, had an equal likelihood of further spread. In 67% (10/15) of patients, significant differences were observed in the number of metastasis-to-metastasis migrations that descended from each primary-to-metastasis seeding subclone (Fig. 3c and Extended Data Fig. 5e). For example, in CRUKP9198, three primary subclones each seeded one or two metastases (Monte Carlo likelihood ratio test $P = 1.0$), but their subsequent spread differed: one did not give rise to any metastasis-to-metastasis migrations; another had a descendant that seeded one metastasis; and the third had descendants that seeded twelve metastases via metastasis-to-metastasis migrations (Extended Data Fig. 5f). Thus, in this small cohort, the capacity to seed further metastases is not uniformly inherited by descendants of primary-to-metastasis seeding subclones.

## Duration in situ associates with seeding

To investigate determinants of metastatic capacity, we examined patient, metastasis and subclone characteristics associated with metastasis-to-metastasis seeding. Patients in whom metastasis-to-metastasis seeding was predominant (more than 50% of all migrations) relapsed later (median DFS 14 versus 5 months, $P = 0.036$, Mann–Whitney $U$ test) and had significantly longer OS (median 32 versus 15 months, $P = 0.019$, Mann–Whitney $U$ test), compared with patients in whom it was not (less than 50%; Extended Data Fig. 6a). Their clinical demographics and treatment histories were otherwise similar (Extended Data Fig. 6a). In fact, OS exhibited a positive linear association with the proportion of migrations that were metastasis-to-metastasis (Pearson's $R$: 0.46, $P = 0.025$; Fig. 4a), raising the possibility that metastasis-to-metastasis seeding is related to the duration metastases are in situ.

To investigate this, we evaluated the seeding capacity of each metastasis with respect to when it arose during the disease course. Metastases

that seeded other metastases were identified significantly earlier on radiological imaging: 36.8% (14/38) of metastases detected on the first relapse scan seeded metastases, compared with 16.4% (10/61) detected only after the last scan, at autopsy ($P = 0.03$, Fisher's exact test; Fig. 4b). Consistent with having a longer duration in situ, these metastases also accrued more somatic alterations than metastases that did not seed (mutations $P = 5.7 \times 10^{-5}$, SCNAs $P = 0.002$, Wilcoxon signed-rank test; Fig. 4c and Methods). Likewise, primary-to-metastasis seeding subclones with metastasis-to-metastasis seeding descendants emerged earlier in tumour evolution than those without metastasis-to-metastasis seeding descendants (mutation distance from the trunk $P = 0.053$, SCNA distance from the trunk $P = 0.025$, one-sided Wilcoxon signed-rank test; Extended Data Fig. 6b). In fact, 68% (141/207) of the metastases that did not seed other metastases contained fewer mutations than the mutational burden typically observed at the point when metastasis-to-metastasis seeding subclones emerged in the same patient (Extended Data Fig. 6c–e and Methods). This suggests that many non-seeding metastases could potentially seed metastases with longer time in situ.

Two explanations could underlie this finding. First, metastases with a longer duration in situ might reach larger sizes, such that an increased number of cancer cells have the potential to seed. Indeed, metastases that seeded other metastases had significantly greater maximum radiological volumes than those that did not ($P = 0.025$, Wilcoxon signed-rank test; Fig. 4d and Methods). Second, subclones capable of seeding metastases might emerge from the reservoir of subclonal diversity that accrues over time. In keeping with this, metastases that seeded other metastases had more subclones ($P = 0.028$, Mann–Whitney $U$ test; Extended Data Fig. 6f) and were more likely to have branched phylogenies than were metastases that did not ($P = 9.8 \times 10^{-8}$, Fisher's exact test; Fig. 4e). These data imply that the greater number of cancer cells and/or subclones afforded by a longer duration in situ is associated with the likelihood of metastasis-to-metastasis seeding.

## Anatomical constraints of seeding

Anatomical location may also influence the likelihood of metastases seeding further metastases; for example, owing to differences in organ vascularity or lymphatic drainage[39]. Metastasis-to-metastasis seeding varied across anatomical locations: 28.4% of lung and 30.6% of intrathoracic lymph node metastases seeded other metastases, compared with only 8.3% of peritoneal and 10.7% of liver metastases (Fig. 4f). These differences mostly reflected the temporal order in which metastases arose. The prevalence of metastasis-to-metastasis seeding from each site strongly correlated with the proportion of metastases at that site with 'sufficient' time in situ to seed others, defined by comparing the mutation burden of each metastasis with the number of mutations accrued before the emergence of metastasis-to-metastasis seeding subclones in the same patient (Pearson's $R$: 0.80, $P = 0.01$; Fig. 4g and Methods). In particular, intrathoracic metastases (encompassing mediastinal lymph node and lung metastases; Methods), which were detected significantly earlier on radiological imaging than were extrathoracic metastases ($P = 3.94 \times 10^{-4}$, Mann–Whitney $U$ test; Extended Data Fig. 7a), seeded other metastases more frequently (intrathoracic versus extrathoracic metastases that seeded other metastases: 26.4% (32/121) versus 13.9% (19/137), $P = 0.013$, Fisher's exact test; Fig. 4h). Thus, the tendency we observe for NSCLC metastases to emerge in an anatomical sequence (intrathoracic early; extrathoracic late) is reflected in metastatic seeding patterns.

Next, we investigated whether the anatomical location in which seeding originates influences the location of the resulting metastases. Overall, 71.3% (92/129) of metastasis-to-metastasis migrations remained within the anatomical cavity in which they originated ($P = 2.40 \times 10^{-5}$, Fisher's exact test; Fig. 4i); that is, intrathoracic metastases

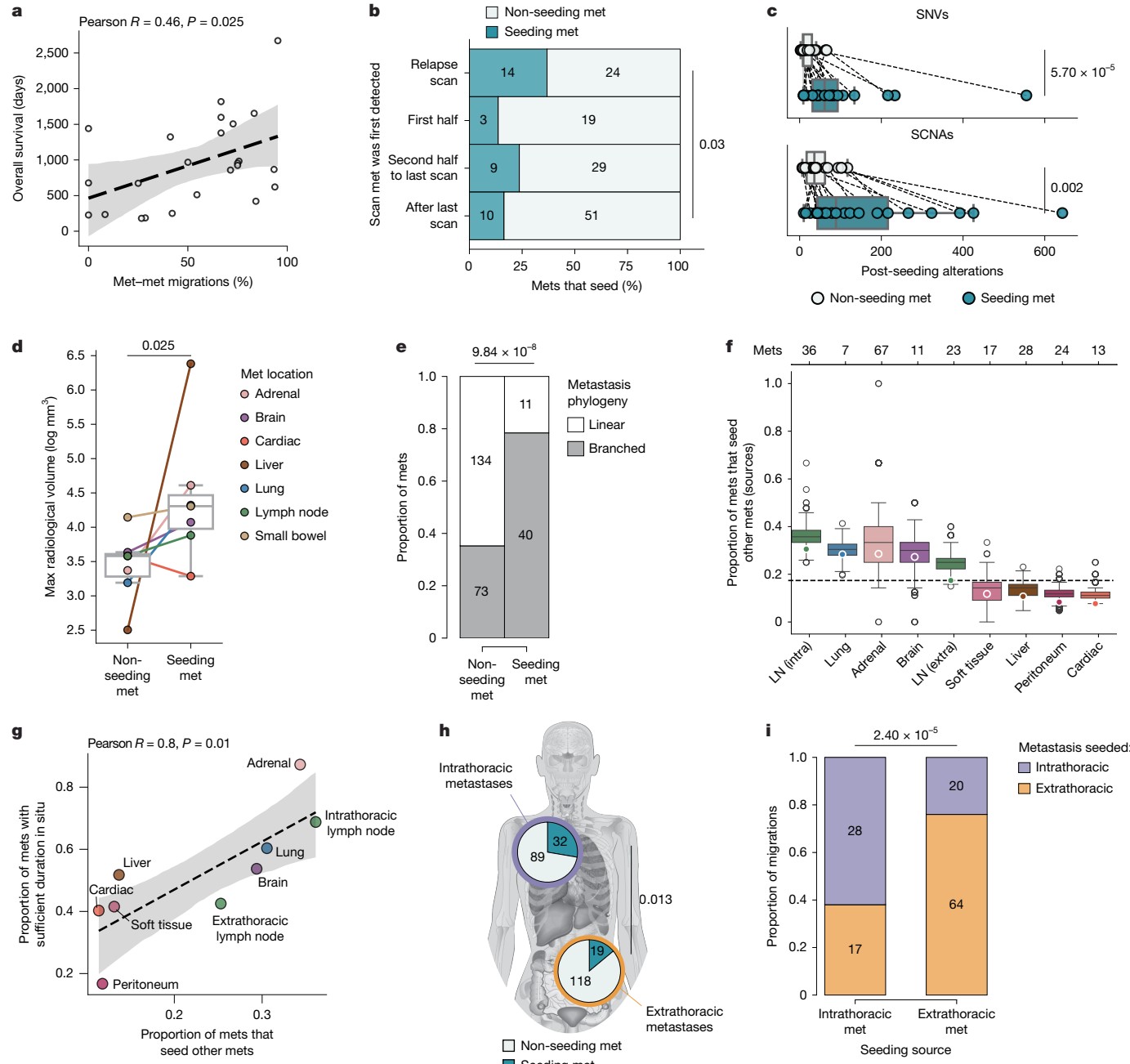

**Fig. 4 | Metastasis-to-metastasis seeding associates with duration in situ and anatomical site. a**, Correlation between the percentage of migrations that constitute metastasis-to-metastasis per patient and overall survival (days). **b**, Proportion of seeding (dark teal) and non-seeding (light teal) metastases stratified by time of first detection on imaging: relapse scan; scan in first half of the post-relapse period; scan in second half; or after the last scan before death (at autopsy). Metastases without known timing of first appearance or seeding status were excluded. *n* = 23 patients; Fisher's exact test comparing earliest and latest categories. **c**, Burden of SNVs and SCNAs accumulated after seeding (that is, in descendants of the seeding subclone) for seeding (dark teal) and non-seeding (light teal) metastases. Dots indicate median per patient; lines connect patients. *n* = 20 patients with both metastasis types; Wilcoxon signed-rank test. **d**, Maximum radiological volume measured on any clinical scan for seeding versus non-seeding metastases. Dots indicate median per organ site; lines connect organ sites. *n* = 23 patients, 129 metastases; Wilcoxon signed-

rank test. **e**, Proportion of seeding and non-seeding metastases with branched (grey) versus linear (white) metastasis-unique subclone phylogenies. *n* = 24 patients; Fisher's exact test. **f**, Prevalence of metastasis-to-metastasis seeding across anatomical locations. Intra, intrathoracic; extra, extrathoracic. **g**, Correlation between the proportion of metastases in each anatomical site with 'sufficient' duration in situ (Methods) and the prevalence of seeding metastases from that site. *n* = 24 patients, 258 metastases. **h**, Proportion of non-seeding (light teal) or seeding (dark teal) metastases in intrathoracic (purple edge) and extrathoracic organs (orange edge). *n* = 24 patients, 258 metastases; Fisher's exact test. **i**, Origin of seeding for intrathoracic (purple) and extrathoracic (orange) metastases, stratified by intrathoracic versus extrathoracic source. *n* = 12 patients with both metastasis types; Fisher's exact test. The box plots show the median and IQR with whiskers denoting values within 1.5 times the IQR from the first and third quartiles. Body map illustration in **h** by J. Brock adapted from ref. 11 under a Creative Commons licence CC BY 4.0.

predominantly seeded intrathoracic metastases, and extrathoracic metastases predominantly seeded extrathoracic metastases. This observation was consistent across patients (Extended Data

Fig. 7b), maintained when restricting to the highest confidence migrations (Methods and Extended Data Fig. 7c) and not explained by a propensity for within-organ spread (59% (54/92) of migrations

within the same anatomical cavity were not within the same organ). Primary-to-metastasis seeding also differed with respect to anatomical cavities. Only one-third (16/48) of primary seeding subclones seeded extrathoracic metastases; however, those that did were more likely to seed multiple metastases than were subclones that seeded intrathoracic metastases ($P = 0.007$, Fisher's exact test; Extended Data Fig. 7d). For example, in CRUKP1584, the primary tumour seeded 11 metastases—7 intrathoracic and 4 extrathoracic. The primary subclone capable of exiting the thorax seeded eight metastases, whereas the subclone that remained within the thorax seeded three (Fig. 3d). The same pattern was evident among metastasis-to-metastasis seeding subclones that originated within intrathoracic metastases (Extended Data Fig. 7e), suggesting that the process of cancer cells migrating between anatomical cavities differs from the process of migrating within them.

## Extrathoracic spread tracks chromosomal instability

Despite sufficient time in situ (as defined in Extended Data Fig. 6c,d and Methods), more than half of metastases did not seed further metastases, implying that time is necessary but not sufficient for metastatic seeding. Genetic alterations that affect cancer cell phenotypes might influence their ability to seed metastases[40]. Previously, we found that primary tumour chromosomal instability (CIN) is associated with the likelihood of metastatic relapse[18,19], the detection of multiple primary seeding subclones[11] and extrathoracic spread[19,41]. In this cohort, in which extensive metastatic sampling has increased the detection of seeding subclones, primary-to-metastasis seeding subclones similarly contained more SCNAs ($P = 9.6 \times 10^{-5}$, linear mixed effects (LME) model; Fig. 5a,b) but not SNVs (Supplementary Fig. 8a) than non-seeding subclones from the same primary. In addition, the percentage of primary subclones that seeded metastases was strongly associated with the median SCNA burden (Pearson's $R$: 0.62, $P = 0.0013$) and rate of SCNA acquisition (per mutation) across all primary subclones (Pearson's $R$: 0.72, $P = 7.5 \times 10^{-5}$; Fig. 5c), suggesting the degree of CIN in primary subclones correlates with their likelihood to seed.

This association extended to late-stage disease: metastasis-to-metastasis seeding subclones had significantly more SCNAs (p-value = $3.0 \times 10^{-7}$, LME model; Fig. 5d) and had a higher rate of SCNA acquisition (Supplementary Fig. 8b) than non-seeding metastasis subclones from the same patient. Although some subclones had SCNAs affecting genes that are implicated in metastasis[11,41]—such as the focal amplification of *CCND1* in the primary subclone that seeded a chest wall metastasis in CRUKP3207, and in a left frontal lobe brain metastasis that seeded a right frontal lobe metastasis in CRUKP8433—no significant enrichment in the rate of SCNAs affecting driver genes or driver mutations was detected in seeding compared with non-seeding subclones (Supplementary Fig. 9). This cohort might be underpowered to detect such differences, or this finding could suggest that CIN supports metastasis through alternative means, such as by generating subclonal diversity or by altering the tumour microenvironment[42].

Given the distinct seeding patterns observed between intrathoracic and extrathoracic spread, we examined the characteristics of subclones that seeded metastases in each location. Primary subclones that seeded extrathoracic metastases, but not those that seeded intrathoracic metastases, had significantly more SCNAs, compared to non-seeding primary subclones (Fig. 5e). We confirmed this observation in the published cohort of TRACERx patients[11] for whom paired primary and metastasis data and extrathoracic relapse status based on imaging were available (Extended Data Fig. 7f and Supplementary Fig. 10). Metastasis-to-metastasis seeding followed the same pattern: intrathoracic metastases that seeded extrathoracic metastases contained subclones with higher SCNA burdens, compared with intrathoracic metastases that seeded intrathoracic metastases ($P = 7.2 \times 10^{-4}$, LME model; Fig. 5f). These data correlate subclone SCNA burden with extrathoracic seeding capacity, suggesting that CIN supports an aspect of the metastatic process that is specific to this route of spread.

## Discussion

Previous genomic studies vary considerably in their conclusions about metastatic heterogeneity[9,43–46] and seeding[11,15,47]. Sampling-related variation could account for some of these differences, given that data generated from limited samples might fail to accurately represent often-widespread metastatic disease[16]. Here, DNA-sequencing data from 108 primary tumour regions paired with 393 metastasis regions, encompassing the majority of metastases radiologically detected before death in 24 patients, enabled in-depth characterization of the subclonal landscape of NSCLC metastases and the cellular migrations that founded them.

The predominant seeding pattern involved the dissemination of multiple primary subclones before surgery or from postoperative residual disease (62.5% of patients), each giving rise to a distinct metastasis (72.5% of metastases were founded by a single subclone). Metastasis-to-metastasis seeding from the resultant metastases, however, were inferred to account for most sampled metastases (60%), suggesting that clinical interventions aimed at minimizing existing metastatic disease could prevent further metastatic progression. Moreover, the latency period associated with metastasis-to-metastasis seeding indicates that there could be a window of opportunity for such interventions. Consistent with this hypothesis, local consolidative therapy (LCT) with radiotherapy or surgery for metastases that persist after systemic therapy improves the outcomes of patients with metastatic NSCLC[48–50] and other cancer types[51] in phase II trials. However, in a 2024 study, LCT did not produce any survival benefit in patients with NSCLC who were treated predominantly with immunotherapy[52], highlighting the unresolved challenge of identifying patients who will truly benefit from this approach. Our data offer a biological rationale for this treatment strategy in appropriately selected patients, and, potentially, in selected metastases, such as those with high CIN.

Corroborating insights from lineage-tracing models of metastasis[53–55], the cellular attributes required to seed metastases, and the likelihood of doing so, varied according to the route of spread. Within-cavity seeding—in which direct invasion, spread through airspaces[56,57] and lymphatic and circulatory migration are all possible routes of spread[58]—was more frequent and feasible than was seeding between cavities, which requires transit in the circulation. In colorectal cancer, local lymph node metastases develop through evolutionary mechanisms that are fundamentally different from those of distant metastases[36,59,60]. Here, CIN, which we[18,19,41] and others[61] have previously found to be associated with metastatic capacity, was a distinguishing feature of subclones that seeded extrathoracic metastases. This raises the possibility that the metastatic advantage conferred by CIN in NSCLC relates to a process specific to extrathoracic seeding, such as circulatory spread or adaptation to a non-thoracic microenvironment, and highlights the need for further functional studies that are designed to elucidate the mechanisms of spread along different anatomical routes.

The generalizability of these results to untreated patients (88% received systemic therapies in this cohort), patients who present with de novo metastatic disease and other cancer types is unknown. The size of the cohort limited our ability to assess the effects of treatments on metastasis evolution, seeding and recurrent genomic events. Furthermore, bulk WES, although performed at high depth, can underestimate subclonal diversity, when compared with whole-genome and/ or single-cell sequencing technologies, and precludes investigation of the roles of structural variants, extrachromosomal DNA, the tumour microenvironment and other non-genetic processes in metastasis. Our follow-on study, TRACERx EVO (NCT05628376), endeavours to address these limitations. It aims to recruit 600 patients with NSCLC, small cell lung cancer or pleural mesothelioma across the spectrum of cancer

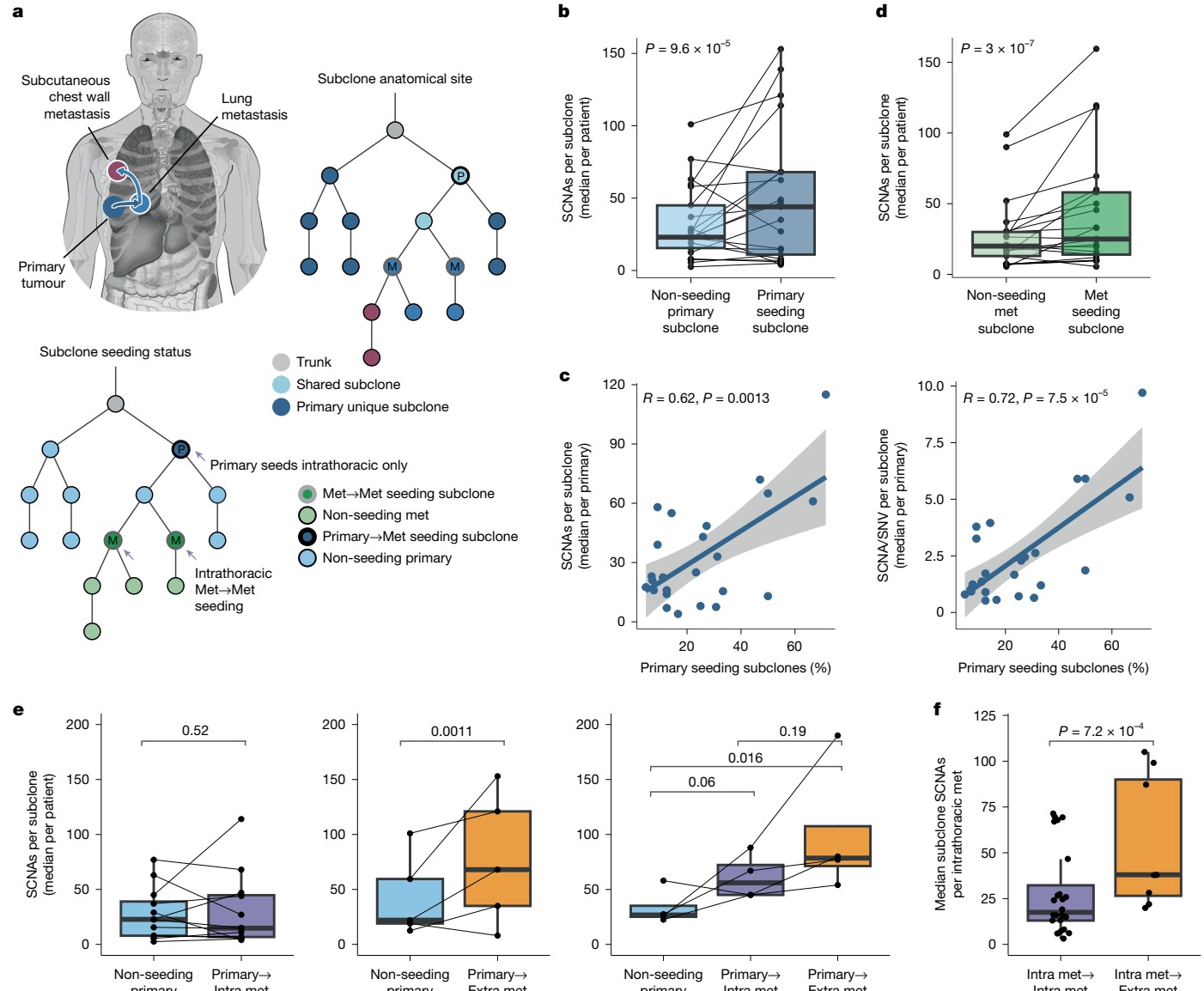

**Fig. 5 | Extrathoracic seeding subclones are enriched for chromosomal instability. a**, Example patient (CRUKP8780), depicting anatomical sites of subclones (top right tree), their migration patterns (body map) and the related classification of seeding and non-seeding (bottom left tree). **b**, SCNA burden per primary-to-metastasis seeding subclone (dark blue) and non-seeding primary subclone (light blue). LME model with subclone mutation burden as covariate and patient as random effect. $n$ = 21 patients, 274 subclones. **c**, Correlation between the percentage of primary subclones that seed metastases and the median subclone SCNA burden (left) or the SCNA/SNV ratio (right). $n$= 24 patients. **d**, SCNA burden per metastasis-to-metastasis seeding subclone (dark green) and non-seeding metastasis subclone (light green). LME model as in **b**. $n$ = 21 patients, 625 subclones. **e**, Primary subclone SCNA burden stratified by subclone seeding status: non-seeding (blue), seeded intrathoracic metastasis (purple) or seeded extrathoracic metastasis (orange), shown for patients with intrathoracic only (left; $n$ = 12 patients, 129 subclones), extrathoracic only (middle; $n$ = 5 patients, 70 subclones) or both intrathoracic and extrathoracic (right; $n$ = 4 patients, 75 subclones) metastases seeded by the primary. Dots, median per patient; Lines, connect primary tumours. LME model as for **b**. **f**, Median SCNA burden per subclone in intrathoracic metastases that seed intrathoracic metastases (purple) and intrathoracic metastases that seed extrathoracic metastases (orange). LME model as in **b**. $n$ = 15 patients, 32 metastases. The box plots show the median and IQR with whiskers denoting values within 1.5 times the IQR from the first and third quartiles. Body map illustration in **a** by J. Brock adapted from ref. 11 under a Creative Commons licence CC BY 4.0.

stages (I–IV), and to carry out up to 100 research autopsies using the PEACE study infrastructure, performing high-depth whole-genome sequencing on the collected samples.

This work demonstrates the extensive genetic diversity that constitutes metastatic disease, revealing that NSCLC progression is propagated by a multitude of primary and metastasis subclones with seeding capacity. It thus highlights the value of this longitudinal, clinically annotated dataset that, by facilitating further interrogation of this complexity, will foster greater understanding of the metastatic process, and inform strategies to curtail it.

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

¹Cancer Research UK Lung Cancer Centre of Excellence, University College London Cancer Institute, London, UK. ²Cancer Metastasis Laboratory, University College London Cancer Institute, London, UK. ³Computational Cancer Genomics Research Group, University College London Cancer Institute, London, UK. ⁴Department of Oncology, University College London Hospitals, London, UK. ⁵Cancer Genome Evolution Research Group, University College London Cancer Institute, London, UK. ⁶Cancer Evolution and Genome Instability Laboratory, Francis Crick Institute, London, UK. ⁷Department of Computer Science, University of Toronto, Toronto, Canada. ⁸Vector Institute for Artificial Intelligence, Toronto, Canada. ⁹Ontario Institute for Cancer Research, Toronto, Canada. ¹⁰Department of Experimental Medicine, Sapienza University of Rome, Rome, Italy. ¹¹Department of Medical Physics and Biomedical Engineering, University College London, London, UK. ¹²Department of Thoracic Surgery, Graduate School of Medicine, University of Tokyo, Tokyo, Japan. ¹³Genomics Science Technology Platform, Francis Crick Institute, London, UK. ¹⁴Cancer Research UK and UCL Cancer Trials Centre, London, UK. ¹⁵Department of Cellular Pathology, University College London Hospitals, London, UK. ¹⁶Department of Cellular Pathology, Guy's and St Thomas' NHS Foundation Trust, London, UK. ¹⁷Birmingham Acute Care Research Group, Institute of Inflammation and Ageing, University of Birmingham, Birmingham, UK. ¹⁸University Hospitals Birmingham NHS Foundation Trust, Birmingham, UK. ¹⁹Division of Cancer Sciences, University

of Manchester and The Christie NHS Foundation Trust, Manchester, UK. [20]Department of Pathology, The Christie NHS Foundation Trust, Manchester, UK. [21]Cancer Research UK Scotland Institute, Glasgow, UK. [22]School of Cancer Sciences, University of Glasgow, Glasgow, UK. [23]Pathology Department, Queen Elizabeth University Hospital, NHS Greater Glasgow and Clyde, Glasgow, UK. [24]Leicester Cancer Research Centre, College of Life Sciences, University of Leicester, Leicester, UK. [25]Department of Histopathology, University Hospitals of Leicester NHS Trust, Leicester, UK. [26]University of Leicester, Leicester, UK. [27]University Hospitals of Leicester NHS Trust, Leicester, UK. [28]University Hospital Southampton NHS Trust, Southampton, UK. [29]Department of Oncology, University Hospital Southampton NHS Foundation Trust, Southampton, UK. [30]Beatson West of Scotland Cancer Centre, NHS Greater Glasgow and Clyde, Glasgow, UK. [31]Golden Jubilee National Hospital, Clydebank, UK. [32]Queen Elizabeth University Hospital, NHS Greater Glasgow and Clyde, Glasgow, UK. [33]Princess Alexandra Hospital, Princess Alexandra Hospital NHS Trust, Harlow, UK. [139]These authors contributed equally: Sonya Hessey, Abigail Bunkum, Ariana Huebner, Kerstin Haase, Kristiana Grigoriadis, Cristina Naceur-Lombardelli. [140]These authors jointly supervised this work: Simone Zaccaria, Nicholas McGranahan, Charles Swanton, Mariam Jamal-Hanjani.

**TRACERx Consortium**

Mariam Jamal-Hanjani[1,2,4,140], Charles Swanton[1,4,6,140], Nicholas McGranahan[1,5,140], Simone Zaccaria[1,3,140], David A. Moore[1,6,15], Sonya Hessey[1,2,3,4,139], Abigail Bunkum[1,2,3,139], Ariana Huebner[1,5,6,139], Kerstin Haase[1,2,139], Kristiana Grigoriadis[1,5,6,139], Cristina Naceur-Lombardelli[1,139], Wing Kin Liu[1,2], Charlotte Grieco[1,6], Carlos Martínez-Ruiz[1,6], Piotr Pawlik[1,5], Olivia Lucas[1,3,4,6], Oriol Pich[6], Kerstin Thol[1,5], Takahiro Karasaki[1,2,6,12], Sophia Ward[1,6,13], Monica Sivakumar[1], Selvaraju Veeriah[1], Andrew J. Rowan[6], Elaine Borg[15], Mary Falzon[15], Martin D. Forster[1,4], Sarah Benafif[1], Tanya Ahmad[4], Siow Ming Lee[1,4], Dionysis Papadatos-Pastos[4], Babu Naidu[17], Gerald Langman[18], Matthew G. Krebs[19], Pedro Oliveira[20], Fiona H. Blackhall[19], Yvonne Summers[19], John Le Quesne[21,22,23], Dean A. Fennell[26,27], Judith Cave[29], Alan Kirk[31], Kevin G. Blyth[21,22,32], Peter Russell[33], Crispin T. Hiley[1,6], Allan Hackshaw[14], Jason F. Lester[34], Amrita Bajaj[27], Apostolos Nakas[27], Azmina Sodha-Ramdeen[27], Claire Wilson[35], Molly Scotland[27], Rebecca Boyles[27], Sean Dulloo[26,27], Sridhar Rathinam[27], Gurdeep Matharu[36], Jacqui A. Shaw[36], Ekaterini Boleti[37], Heather Cheyne[38], Gillian Price[39,40], Keith M. Kerr[40,41], Mohammed Khalil[38], Shirley Richardson[38], Tracey Cruickshank[38], Jack French[38], Kayleigh Gilbert[42], Akshay J. Patel[43], Aya Osman[18], Gary Middleton[18,44], Helen Shackleford[18], Madava Djearaman[18], Mandeesh Sangha[18], Angela Leek[45], Adam Atkin[46], Anshuman Chaturvedi[47,48], Antonio Paiva-Correia[49], Colin R. Lindsay[19,48], Eustace Fontaine[46], Felice Granato[46], Jack Davies Hodgkinson[45], Juliette Novasio[46], Katherine D. Brown[47,48], Kandadai Rammohan[46], Leena Joseph[46], Mathew Carter[46], Nicola Totton[45], Paul Bishop[46], Philip A. J. Crosbie[46,48,50], Sara Waplington[46], Jonathan Tugwood[48,51], Caroline Dive[48,51], Hugo JWL Aerts[52,53,54], Gareth A. Wilson[6], Aino-Maija Leppä[6], Alexander A. Azizi[1,6], Lydia Y. Liu[1,6], Jonas Demeulemeester[55,56,57], Miklos Diossy[58,59,60], Nicolai J. Birkbak[1,6,61,62,63], Peter Van Loo[64,65], Rachel Rosenthal[6], Roberto Salgado[66,67], Roland F. Schwarz[68,69], Tom L. Kaufmann[69,70], Zoltan Szallasi[58,59,71], Alexander M. Frankell[72], Angela Dworník[73], Angeliki Karamani[73], Karen Grimes[1,5], Benny Chain[73], Carla Castignani[74,75], Chris Bailey[6], Cian Murphy[6], Clare E. Weeden[76], Clare Puttick[1,5,6], David R. Pearce[73], Despoina Karagianni[1,77], Dimitra Brempou[73], Emilia L. Lim[78], Emma C. Colliver[73], Emma Hazelwood[72], Emma Nye[79], Erik Sahai[76], Eva Grönroos[76], Francisco Gimeno-Valiente[1], Gemma Foulds[6], George Kassiotis[76,81], Georgia Moth[1], Georgia Stavrou[73], Helen L. Lowe[73], Ieva Usaite[1], Iva Mladenova[1,3,5,13], Jacki Goldman[76], James L. Reading[1,82], James R. M. Black[1,6], Jayant K. Rane[6,73], Jeanette Kittel[1,2], John A. Hartley[73], Jorge Martin Arana[73], Karl S. Peggs[73,83], Katey S. S. Enfield[6,84,85], Katherine Honan[72], Kayalvizhi Selvaraju[73], Kexin Koh[1,2], Krupa Thakkar[1], Leah Ensell[73], Lucrezia Patruno[1,3], Maise Al Bakir[1,6], Mansi Shah[73], Maria Litovchenko[73], Maria Zagorulya[6], Michalina Magala[1], Michelle M. Leung[1,5,6], Mickael Escudero[76], Mihaela Angelova[6], Nnennaya Kanu[1], Oliver Shutkever[1,2], Philip Hobson[80], Richard Kevin Stone[79], Rija Zaidi[1,3], Robert Bentham[1,6], Robert Goldstone[13], Roberto Vendramin[1,6,86], Sadegh Saghafinia[1], Samuel Gamble[73], Seng Kuong Anakin Ung[73], Sergio A. Quezada[1,77], Sharon Vanloo[1], Sian Harries[1,6,13], Stefan Boeing[76], Stephan Beck[75], Supreet Kaur Bola[1,77], Teresa Marafioti[15], Theepan Visakan[1], Thomas B. K. Watkins[87], Thomas Patrick Jones[5], Victoria Spanswick[73], Vittorio Barbè[88], Wei-Ting Lu[76,89], William Hill[76], Woody Z. Zhang[72], Yin Wu[73], Yutaka Naito[76], Zoe Ramsden[76], Catarina Veiga[11], Charles-Antoine Collins-Fekete[11], Francesco Fraioli[90], Gary Royle[91], Paul Ashford[92], Alexander James Procter[93], Arjun Nair[93,94], Asia Ahmed[93], David Lawrence[95], Davide Patrini[95], Emilie Martinoni Hoogenboom[96], Fleur Monk[96], James W. Holding[96], Junaid Choudhary[96], Kunal Bhakhri[96], Magali N. Taylor[93], Maria Chiara Pisciella[96], Neal Navani[97,98], Pat Gorman[96], Reena Khiroya[15], Ricky M. Thakrar[97,98], Robert CM Stephens[96], Sam M. Janes[99], Steve Bandula[96], Zoltan Kaplar[100,101], Aoife Walker[14], Camilla Pilotti[14], Rachel Leslie[14], Salomey Kellett[14], Anca Grapa[102], Hanyun Zhang[103], Khalid AbdulJabbar[104], Xiaoxi Pan[105], Yinyin Yuan[105], David Chuter[106], Mairead MacKenzie[106], Aiman Alzetani[107], Patricia Georg[108], Serena Chee[108], Eric Lim[109,110], Alexandra Rice[110], Anand Devaraj[110], Andrew G. Nicholson[110,111], Chiara Proli[110], Daniel Kaniu[110], Harshil Bhayani[110], Hema Chavan[110], Hilgardt Raubenheimer[110], Lyn Ambrose[110], Mpho Malima[110], Nadia Fernandes[110], Paulo De Sousa[110], Pratibha Shah[110], Sarah Booth[110], Silviu I. Buderi[110], Simon Jordan[110], Sofina Begum[110], Madeleine Hewish[112,113], Sarah Danson[114,115], Michael J. Shackcloth[116], Lily Robinson[33], Andrew Kidd[117], Craig Dick[118], Jennifer Whiteley[31], Mathew Thomas[31], Mohammed Asif[31], Nikos Kostoulas[31] & Rocco Bilancia[31]

[34]Singleton Hospital, Swansea Bay University Health Board, Swansea, UK. [35]Leicester Medical School, University of Leicester, Leicester, UK. [36]Cancer Research Centre, University of Leicester, Leicester, UK. [37]Royal Free London NHS Foundation Trust, London, UK. [38]Aberdeen Royal Infirmary NHS Grampian, Aberdeen, UK. [39]Department of Medical Oncology, Aberdeen Royal Infirmary NHS Grampian, Aberdeen, UK. [40]University of Aberdeen, Aberdeen, UK. [41]Department of Pathology, Aberdeen Royal Infirmary NHS Grampian, Aberdeen, UK. [42]Whittington Hospital NHS Trust, London, UK. [43]Guy's and St Thomas' NHS Foundation Trust, London, UK. [44]Institute of Immunology and Immunotherapy, University of Birmingham, Birmingham, UK. [45]Manchester Cancer Research Centre Biobank, Manchester, UK. [46]Wythenshawe Hospital, Manchester University NHS Foundation Trust, Wythenshawe, UK. [47]The Christie NHS Foundation Trust, Manchester, UK. [48]Cancer Research UK Lung Cancer Centre of Excellence, University of Manchester, Manchester, UK. [49]Cellular Pathology Department, Wythenshawe Hospital, Manchester University NHS Foundation Trust, Wythenshawe, UK. [50]Division of Infection, Immunity and Respiratory Medicine, University of Manchester, Manchester, UK. [51]Cancer Research UK Manchester Institute Cancer Biomarker Centre, University of Manchester, Manchester, UK. [52]Artificial Intelligence in Medicine (AIM) Program, Mass General Brigham, Harvard Medical School, Boston, MA, USA. [53]Department of Radiation Oncology, Brigham and Women's Hospital, Dana-Farber Cancer Institute, Harvard Medical School, Boston, MA, USA. [54]Radiology and Nuclear Medicine, CARIM and GROW, Maastricht University, Maastricht, The Netherlands. [55]Integrative Cancer Genomics Laboratory, VIB Center for Cancer Biology, Leuven, Belgium. [56]VIB Center for AI and Computational Biology, Leuven, Belgium. [57]Department of Oncology, KU Leuven, Leuven, Belgium. [58]Danish Cancer Institute, Copenhagen, Denmark. [59]Computational Health Informatics Program, Boston Children's Hospital, Boston, MA, USA. [60]Department of Physics of Complex Systems, ELTE Eötvös Loránd University, Budapest, Hungary. [61]Department of Molecular Medicine, Aarhus University Hospital, Aarhus, Denmark. [62]Department of Clinical Medicine, Aarhus University, Aarhus, Denmark. [63]Bioinformatics Research Centre, Aarhus University, Aarhus, Denmark. [64]Department of Genetics, University of Texas MD Anderson Cancer Center, Houston, TX, USA. [65]Department of Genomic Medicine, University of Texas MD Anderson Cancer Center, Houston, TX, USA. [66]Department of Pathology, ZAS Hospitals, Antwerp, Belgium. [67]Division of Research, Peter MacCallum Cancer Centre, Melbourne, Australia. [68]Institute for Computational Cancer Biology (ICCB), Center for Integrated Oncology (CIO), Cancer Research Center Cologne Essen (CCCE), Faculty of Medicine and University Hospital Cologne, University of Cologne, Cologne, Germany. [69]Berlin Institute for the Foundations of Learning and Data (BIFOLD), Berlin, Germany. [70]Institute for Computational Cancer Biology (ICCB), Faculty of Medicine and University Hospital Cologne, University of Cologne, Cologne, Germany. [71]Department of Bioinformatics, Semmelweis University, Budapest, Hungary. [72]Somatic Evolution Monitoring Lab, Early Cancer Institute, CRUK Cambridge Centre, University of Cambridge, Cambridge, UK. [73]University College London Cancer Institute, London, UK. [74]Cancer Genomics Laboratory, Francis Crick Institute, London, UK. [75]Medical Genomics, University College London Cancer Institute, London, UK. [76]Francis Crick Institute, London, UK. [77]Immune Regulation and Tumour Immunotherapy Group, Cancer Immunology Unit, Research Department of Haematology, University College London Cancer Institute, London, UK. [78]Department of Biochemistry and Molecular Biology and Edwin S.H. Leong Centre for Healthy Aging, Faculty of Medicine, The University of British Columbia, Vancouver, British Columbia, Canada. [79]Experimental Histopathology, Francis Crick Institute, London, UK. [80]Flow Cytometry STP, Francis Crick Institute, London, UK. [81]Department of Infectious Disease, Faculty of Medicine, Imperial College London, London, UK. [82]Pre-Cancer Immunology Laboratory, Department of Haematology, University College London Cancer Institute, London, UK. [83]Department of Haematology, University College London Hospitals, London, UK. [84]Basic and Translational Research, BC Cancer Research Institute, Vancouver, British Columbia, Canada. [85]Pathology and Laboratory Medicine, University of British Columbia, Vancouver, British Columbia, Canada. [86]Tumour Immunogenomics and Immunosurveillance Laboratory, University College London Cancer Institute, London, UK. [87]Department of Pathology, Stanford University School of Medicine, Stanford, CA, USA. [88]Stem Cells and Organoids Platform, Francis Crick Institute, London, UK. [89]Department of Oncology, University of Oxford, Oxford, UK. [90]Institute of Nuclear Medicine, Division of Medicine, University College London, London, UK. [91]Department of Medical Physics and Bioengineering, University College London Cancer Institute, London, UK. [92]Institute of Structural and Molecular Biology, University College London, London, UK. [93]Department of Radiology, University College London Hospitals, London, UK. [94]UCL Respiratory, Department of Medicine, University College London, London, UK. [95]Department of Thoracic Surgery, University College London Hospitals NHS Trust, London, UK. [96]University College London Hospitals, London, UK. [97]Lungs for Living Research Centre, UCL Respiratory, University College London, London, UK. [98]Department of Thoracic Medicine, University College London Hospitals, London, UK. [99]Lungs for Living Research Centre, UCL Respiratory, Department of Medicine, University College London, London, UK. [100]Integrated Radiology Department, North-Buda St John's Central Hospital, Budapest, Hungary. [101]Institute of Nuclear Medicine, University College London Hospitals, London, UK. [102]Institute of Cancer Research, London, UK. [103]Garvan Institute of Medical Research, Sydney, New South Wales, Australia. [104]Case45, London, UK. [105]University of Texas MD Anderson Cancer Center, Houston, TX, USA. [106]Independent Cancer Patient's Voice, London, UK. [107]Department of Thoracic Surgery, NIHR Southampton Biomedical Research Centre, University Hospital Southampton NHS Foundation Trust, Southampton, UK. [108]University Hospital Southampton NHS Foundation Trust, Southampton, UK. [109]Academic Division of Thoracic Surgery, Imperial College London, London, UK. [110]Royal Brompton and Harefield Hospitals, part of Guy's and St Thomas' NHS Foundation Trust, London, UK. [111]National Heart and Lung Institute, Imperial College, London, UK. [112]Royal Surrey Hospital, Royal Surrey Hospitals NHS Foundation Trust, Guildford, UK. [113]University of Surrey, Guildford, UK. [114]University of Sheffield, Sheffield, UK. [115]Sheffield Teaching Hospitals NHS Foundation Trust, Sheffield, UK. [116]Liverpool Heart and Chest Hospital, Liverpool, UK. [117]Institute of Infection, Immunity and Inflammation, University of Glasgow, Glasgow, UK. [118]NHS Greater Glasgow and Clyde, Glasgow, UK.

**TRACERx EVO Consortium**

Mariam Jamal-Hanjani[1,2,4,140], Charles Swanton[1,4,6,140], Nicholas McGranahan[1,5,140], David A. Moore[1,6,15], Ariana Huebner[1,5,6,139], Cristina Naceur-Lombardelli[1,139], Carlos Martínez-Ruiz[1,6], Olivia Lucas[1,3,4,6], Sophia Ward[1,6,13], Selvaraju Veeriah[1], Dionysis Papadatos-Pastos[4], Fiona H. Blackhall[19], Yvonne Summers[19], Crispin T. Hiley[1,6], Allan Hackshaw[14], Aoife Walker[14], Salomey Kellett[14], Zainab Kalokoh[14], Maise Al Bakir[1,6], Sharon Vanloo[1], Sian Harries[1,6,13], Georgia Moth[1], Elizabeth Keene[1], Theepan Vikasan[1], Michalina Magala[1], Jacki Goldman[76], Karen Ambrose[76], Mike Gavrielides[76], Nitzan Rosenfeld[119], Amrit Roshan[119], Cecilie Agergaard Soerensen[120], Ben Solomon[67], Lavinia Tan[67], Adam Atkin[46], Ana Parreira[47,48], Angela Leek[45], Colin R. Lindsay[19,48], Corinne Faivre-Finn[47,48], Fabio Gomes[47,48], Igor Gomez-Randulfe[47,48], Jack Webster[47,48], Katherine D. Brown[47,48], Laura Cove-Smith[47,48], Leena Joseph[46], Pamela Maroa[46], Paul Taylor[47,48], Philip A. J. Crosbie[46,48,50], Raffaele Califano[19,48], Sara Tenconi[46], Adam Peryt[120,121], Aman Coonar[120,121], Amanda Stone[120,121], Caroline Sanganee[120,121], Martin Goddard[120,121], Stephen Preston[121], Giuseppe Aresu[120,121], Jane Lichfield[120,121], Julia Knight[120,121], Lauren DSA[120,121], Maria Manuela Urda[120,121], Maria Nizami[120,121], Robert Rintoul[120,121], Zoe Armstrong[120,121], Abiya Mathew[122], Damalie Namwanja[122], Nicky Thomson[122], Philip Earwaker[122] & Lily Robinson[33]

[119]Queen Mary University of London, London, UK. [120]University of Cambridge, Cambridge, UK. [121]Royal Papworth Hospital NHS Foundation Trust, Cambridge, UK. [122]Cambridge University Hospitals NHS Foundation Trust, Cambridge, UK.

**PEACE consortium**

Mariam Jamal-Hanjani[1,2,4,140], Charles Swanton[1,4,6,140], Nicholas McGranahan[1,5,140], Simone Zaccaria[1,3,140], David A. Moore[1,6,15], Sonya Hessey[1,2,3,4,139], Abigail Bunkum[1,2,3,139],

Ariana Huebner[1,5,6,139], Kerstin Haase[1,2,139], Kristiana Grigoriadis[1,5,6,139], Cristina Naceur-Lombardelli[1,139], Wing Kin Liu[1,2], Carlos Martínez-Ruiz[1,6], Piotr Pawlik[1,5], Olivia Lucas[1,3,4,6], Oriol Pich[6], Kerstin Thol[1,5], Takahiro Karasaki[1,2,6,12], Sophia Ward[1,6,13], Selvaraju Veeriah[1], Andrew J. Rowan[6], Hayley Bridger[14], Miriam Mitchison[15], Mary Falzon[15], Anna Green[16], Martin D. Forster[1,4], Sarah Benafif[4], Tanya Ahmad[4], Siow Ming Lee[1,4], Dionysis Papadatos-Pastos[4], Babu Naidu[17], Gerald Langman[18], Matthew G. Krebs[19], Pedro Oliveira[20], Fiona H. Blackhall[19], Yvonne Summers[19], John Le Quesne[21,22,23], Cathy Richards[25,26], Dean A. Fennell[26,27], Sanjay Jogai[28], Judith Cave[29], Patricia Roxburgh[22,30], Sioban Fraser[21,23], Kevin G. Blyth[21,22,32], Crispin T. Hiley[1,6], Allan Hackshaw[14], Kai-Keen Shiu[96], John Bridgewater[96], Daniel Hochhauser[73,96], Sam M. Janes[99], Peter Van Loo[64,65], Katey S. S. Enfield[6,84,85], Sergio A. Quezada[1,77], Stephan Beck[75], Tariq Enver[73], David R. Pearce[73], Ron Sinclair[96], Zoe Rhodes[96], Teresa Marafioti[15], Mark Linch[73], Sebastian Brandner[123], Heather Shaw[96,124], Gerhardt Attard[73,96], Faye Gishen[6,42,125], Maise Al Bakir[1,6], Nnennaya Kanu[1], Francisco Gimeno Valiente[1], Sian Harries[1,6,13], Emilia L. Lim[78], James L. Reading[1,82], Benny Chain[73], Adrienne Flanagan[73], Emma C. Colliver[6], Mihaela Angelova[6], James R. M. Black[1,6], William Hill[76], Alexander M. Frankell[72], Roberto Salgado[66,67], Vittorio Barbè[88], Supreet Kaur Bola[1,77], Osvaldas Vainauskas[73], Anna Wingate[73], Daniel Wetterskog[73], A. M. Mahedi Hasan[73], Stefano Lise[73], Gianmarco Leone[73], Jorge Martin Arana[73], Dimitria Brempou[73], Anuradha Jayaram[73,96], Constantine Alifrangis[96], Ursula McGovern[96], Samuel Gamble[73], Seng Kuong Anakin Ung[73], Roberto Vendramin[1,6,86], Jayant K. Rane[6,73], Angela Dwornik[73], Kerry Bowles[96], Jeanette Kittel[1,2], Rija Zaidi[1,3], Athanasia Vargiamiou[96], Lucrezia Patruno[1,3], Christopher Aled Chamberlain[1,77], Oliver Shutkever[1,2], Welles Robinson[1,77], Iain McNeish[126], Nataly Ojeda Mosquera[1,77], Jiali Liu[1,77], Felix O'Farrell[1,77], Chenelle Marcel[96], Samra Turajlic[102,127,128], James Larkin[102,127], Lisa Pickering[127], Andrew Furness[127], Kate Young[127], Will Drake[119,129], Kim Edmonds[127], Nikki Hunter[127], Mary Mangwende[127], Lauren Grostate[127], Lavinia Spain[76,127], Scott Shepherd[76,127], Haixi Yan[76,127], Benjamin Shum[127,128], Zayd Tippu[76,127], Brian Hanley[76,102,127], Charlotte Spencer[76,127], Max Emmerich[76,127], Camille Gerard[76,127], Eleanor Carlyle[127], Steve Hazell[127], Hardeep Mudhar[115], Christina Messiou[102,127], Arash Latifoltojar[102,127], Annika Fendler[76], Fiona Byrne[76], Husayn Pallikonda[76], Irene Lobon[76], Alexander Coulton[76], Anne-Laure Cattin[76], Daqi Deng[76], Hugang Feng[76], Nadia Yousaf[127], Sanjay Popat[127], Charlotte Milner-Watts[127], Emma Nye[79], Aida Murra[127], Justine Korteweg[127], Lauren Terry[127], Jennifer Biano[127], Kema Peat[127], Emma Turay[127], Peter Hill[127,130], Marija Miletic[127], Anadil Javaid[127], Jennifer Thomas[127], Bakir Kudic[127], Orla McGowan[127], Dharmista Ramesh[127], Oznur Saka[127], Sinem Arslan[127], Laura Marandino[127], Reina Ammar[127], Gurneet Kapur[127], Dilruba Kabir[127], David McMahon[127], Alexius John[127], Foteini Kalofonou[127], Debra Josephs[43], Sheeba Irshad[43], James Spicer[131], Ruby Stewart[43], Natasha Wright[43], Ruxandra Mitu[43], Deborah Enting[43], Sarah Rudman[43], Sharmistha Ghosh[43], Eleni Lena Karapanagiotou[43], Elias Pintus[43], Andrew Tutt[43], James D. Brenton[132], Nicola Thompson[133], Rebecca Fitzgerald[134], Merche Jimenez-Linan[133], Elena Provenzano[133], Anna Paterson[133], Kieren Allinson[133], Grant D. Stewart[135], Ultan McDermott[133,136], Tim Maughan[137], Olaf Ansorge[136], Peter Campbell[136], Caroline Dive[48,51], Fabio Gomes[47,48], Mat Carter[47], Jacqui A. Shaw[36], Claire Wilson[35], Charlotte Poile[26], Kudazyi H. Kutywayo[27], Maurice R. Dungey[26], Jens Claus Hahne[138], Shobhit Baijal[18], Charlotte Ferris[18], Hollie Bancroft[18], Amy Kerr[18], Gary Middleton[18,44], Joanne Webb[18], Salma Kadiri[18], Bernard Olisemeke[18], Rodelaine Wilson[18], Aya Osman[18], Ian Tomlinson[18], Luke Nolan[28], Samantha Holden[28], Tania Fernandes[28], David Chuter[106], Mairead McKenzie[106], Aoife Walker[14], Rachel Leslie[14] & Shivani Patel[14]

[123]University College London Queen Square Institute of Neurology, London, UK. [124]Mount Vernon Cancer Centre, Northwood, UK. [125]UCL Medical School, University College London, London, UK. [126]Imperial College London, London, UK. [127]The Royal Marsden Hospital, London, UK. [128]Cancer Dynamics Laboratory, Francis Crick Institute, London, UK. [129]St Bartholomew's Hospital, Barts Health NHS Trust, London, UK. [130]Imperial College London NHS Foundation Trust, London, UK. [131]King's College London, London, UK. [132]Cancer Research UK Cambridge Institute, University of Cambridge, Cambridge, UK. [133]Addenbrooke's Hospital, Cambridge University Hospitals, Cambridge, UK. [134]Early Cancer Institute, Department of Oncology, University of Cambridge, Cambridge, UK. [135]Department of Surgery, University of Cambridge, Cambridge, UK. [136]Wellcome Sanger Institute, Hinxton, UK. [137]MRC Oxford Institute for Radiation Oncology, University of Oxford, Oxford, UK. [138]National Institute for Health Research Biomedical Research Centre and Cancer Research UK Experimental Cancer Medicine Centre, University of Leicester, Leicester, UK.

## Methods

### Patient cohort

**The PEACE study.** PEACE is a pan-cancer, UK-wide research autopsy programme (https://clinicaltrials.gov/study/NCT03004755) designed to investigate the biology of metastatic disease and drug resistance. The study was sponsored by the University College London (UCL) Clinical Trials Centre and approved by the Health Research Authority National Research Ethics Service Committee London–Dulwich on 15 August 2013, in accordance with the UK Human Tissue Act 2004, with research ethics committee reference 13/LO/0972. Informed consent was provided by patients during life or by a person in a qualifying relationship after death.

Eligibility was defined by the following inclusion criteria: (1) age 18 years or over; (2) confirmed solid malignancy with metastatic disease (where the site of origin is known or unknown), with the exception of primary brain tumours, in which there might not be evidence of metastatic disease; and (3) oral and written informed consent from patient to enter the study and to undergo tissue collection after death or from a nominated representative or a person in a qualifying relationship after the patient has died. Exclusion criteria were: (1) medical or psychiatric condition that would preclude informed consent; (2) history of intravenous drug abuse within the past five years; or (3) confirmed diagnosis of known high-risk infections (for example, HIV/AIDS-positive, hepatitis B or C, tuberculosis and Creutzfeldt–Jacob disease), unless the patient case is of particular scientific interest and was agreed in advance with local mortuary staff and pathologist.

**The TRACERx–PEACE lung cohort.** The TRACERx study (https://clinicaltrials.gov/ct2/show/NCT01888601) is a prospective observational cohort study approved by an independent research ethics committee (13/LO/1546). The inclusion and exclusion criteria, clinical data acquisition and tissue and plasma sampling procedures have been described[18,19]. In brief, the TRACERx study includes patients with histopathologically confirmed early-stage I–IIIB NSCLC who underwent primary surgery. Patients are followed up after surgery, during which longitudinal clinical data, plasma and, in the case of disease relapse or progression, tissue samples are collected.

Forty-nine patients with NSCLC were enrolled in both TRACERx and PEACE; 41 died and 33 underwent a research autopsy (Supplementary Fig. 1). Research autopsies were not performed owing to lack of death notification ($n = 6$), post-mortem withdrawal of consent ($n = 1$) or COVID-19 restrictions ($n = 1$).

No tumour was identified in four patients who underwent an autopsy after pathological assessment. In three patients, WES data from all primary or all autopsy samples failed quality control. WES data were unavailable for two patients at data lock. The final cohort comprised 24 patients.

Patients were assigned study identifiers that were subsequently converted to linked identifiers (CRUKP prefix) to maintain anonymity. Tissue and blood samples were barcoded and tracked in a centralized database overseen by the sponsor (UCL Clinical Trials Centre).

### Research autopsy sample procurement

Research autopsies were performed as soon as possible after death (median [IQR]: 79 h [56.0–157.3]; median time to refrigeration: 3.7 h [3.2–5.4]) at the recruiting-site affiliated mortuary (University College London Hospitals (UCLH), Guy's and St Thomas' Hospital, Birmingham Heartlands Hospital, Leicester Royal Infirmary Hospital or the Christie Hospital).

Tissue sampling was led by a pathologist who was provided with the patient's pre-mortem clinical history and imaging. None of the patients (or persons in a qualifying relationship) in this cohort elected to restrict research autopsy sampling. All macroscopically visible metastases were sampled where feasible, and multiple regions of individual metastases were sampled where feasible. Sample annotations that distinguished regions from an individual metastasis (that is, a multi-region sampled metastasis) from macroscopically distinct metastases were assigned to allow intra- and inter-metastasis analyses. Labelled specimens were photographed where feasible.

Where possible, metastases were bisected longitudinally: one half was snap-frozen in liquid nitrogen and stored at −80 °C, and the other half was formalin-fixed and paraffin-embedded (FFPE). Fresh sterile instruments were used for each sample. Body fluids (pleural, peritoneal and cerebrospinal) were centrifuged and cell pellets snap-frozen separately. Peripheral blood was collected pre-mortem or at autopsy from the femoral vein or cardiac ventricle.

### Central histopathological review

Diagnostic histopathological slides from the primary tumours were centrally reviewed as previously described[18]. Haematoxylin and eosin (H&E)-stained slides created from the metastasis FFPE blocks were digitally archived and assessed for tumour content, necrosis, autolysis and lymphocyte content by a pathologist.

### Intrathoracic and extrathoracic classification

Metastases were classified as intrathoracic or extrathoracic on the basis of the anatomical site recorded on the pre-mortem sample histopathology reports or by the autopsy pathologist. Where anatomical origin was uncertain, radiological imaging was reviewed (see Supplementary Note).

**Intrathoracic metastases.** Intrathoracic metastases included mediastinal lymph nodes, mediastinal soft tissue, lung, lung surgical bed, pleura and chest wall if the metastasis radiologically arose from within the pleural boundary.

**Extrathoracic metastases.** Extrathoracic metastases included axillary, cervical, supraclavicular and abdominopelvic lymph nodes, chest wall if the metastasis radiologically arose from outside the pleural boundary, other subcutaneous or soft tissue musculoskeletal masses, cardiac (pericardium and myocardium), diaphragm (unless radiological evidence of direct pleural or intrapulmonary extension), bone, liver, brain, gastric, adrenals, kidney, peritoneum and bladder.

### Clinical outcome data

DFS was defined as the period from the date of registration to the time of radiological confirmation of the recurrence of the primary tumour registered for TRACERx or the time of death by any cause.

OS was defined as the period from the date of registration to the time of death by any cause.

### Radiological data curation and analysis

Anonymized clinical imaging scans and reports (CT, PET–CT and MRI) were available for 23 out of 24 patients under the TRACERx study protocol, spanning baseline primary imaging, relapse, up to the last scan before death. Radiologically visible, measurable metastases were contoured using ITK-SNAP v.4.2.0 by a clinical oncologist to produce three-dimensional tumour volumes, a selection of which were reviewed by a second, senior clinical oncologist. Radiological lesions were manually mapped to sequenced metastasis samples where possible. Where multiple sequenced samples mapped to one radiological lesion, the genomic feature predominant to the set (mode) was used in analyses.

**Time to first detection.** Time to first detection was defined as days from primary surgery to the first detection on imaging. Metastases that were not detected on imaging but were sampled at autopsy were assigned a time to first detection halfway between the last scan performed and the date of death.

**Scan period.** Scan dates were normalized per patient by the number of days between relapse imaging and death. Scans after the relapse scan and before the last scan before death were assigned to the first half (≤50%) or second half (>50%) of the metastatic period.

**Maximum tumour volume.** Maximum tumour volume was defined as the largest volume recorded for a metastasis on any longitudinal scan.

## DNA extraction and WES
DNA extraction and WES were performed as previously described[18] for both primary tumour and metastasis samples. Paired germline DNA was resequenced in the same run as subsequently sequenced metastases.

All primary tumour regions and pre-mortem metastasis biopsies and metastatectomies collected underwent WES. Research-autopsy metastasis samples with adequate histopathological tumour content and DNA integrity number (DIN) > 4, as measured using an Agilent TapeStation system, underwent WES. In patients with a large number of suitable samples, samples were selected to capture the anatomical distribution of metastatic disease while maximizing tissue quality. Overall, 376 out of 601 research autopsy metastases were selected for WES (Supplementary Fig. 1).

## Post-sequencing quality control
Primary and metastasis samples that failed copy-number calling and variant calling are summarized in Supplementary Fig. 1. Metastases that passed variant but not copy-number calling ($n = 10$) were included in analyses that do not involve phylogenies (which require both copy-number and mutation calls).

## Bioinformatic pipeline
WES data were analysed using the previously described bioinformatic pipeline to perform alignment, somatic mutation calling, copy-number detection and signature artefact quality control with the following modifications: (i) DNVs were defined by two criteria: First, a proportion test was performed to determine whether the frequencies of two SNVs were significantly similar. For all putative DNVs with a significant test, reads were extracted to calculate the proportion of reads overlapping between the bases. DNVs were called when ≥90% of reads contained both variants in at least one sample from the patient. (ii) Refphase[62] was used to infer haplotype-specific copy-number alterations, and to rescue low-purity tumour regions, using the multi-region data. (iii) Driver mutation annotation was updated as detailed in 'Detection of driver alterations'.

## WGD detection
WGD events were identified and assigned to phylogenies using ParallelGDDetect as described previously[19].

## Genomically independent tumours
Sequenced primary and metastasis regions were deemed genomically related or independent to the other samples collected from the patient as previously described[19].

Multiple genomically independent tumours were detected for three patients. In CRUKP1584, a synchronous lung tumour resected at the time of primary surgery was genomically independent from the other tumour resected at the same time and all metastases subsequently sampled, consistent with a second primary that did not metastasize. In CRUKP8172 and CRUKP7741, metastases sampled at autopsy (a lung and an oesophageal sample, respectively) were genomically independent from the corresponding primary and metastasis samples, consistent with second primaries or metastases from undetected second primaries. Because paired primary–metastasis samples were available for these three genomically independent tumours, they were excluded from the final cohort.

## Subclone and phylogenetic tree reconstruction
**Mutation clustering and tree building.** CONIPHER[63] was used to identify clusters of somatic mutations that occurred in the same tumour subclone and to reconstruct tumour phylogenetic trees. Two functionalities of CONIPHER were important in the context of the number of samples available per patient. First, mutations were pre-clustered by presence (more than one mutant read) or absence across the tumour regions available per patient and PyClone[64] was applied to each mutation group independently, making the mutation clustering step scalable[11,63]. Second, CONIPHER enumerated all possible phylogenetic tree topologies compatible with the pigeonhole principle and the crossing rule[65] for each set of mutation clusters. The sum condition error (SCE) was computed for each solution to quantify the extent to which the evolutionary constraints imposed by the topology were violated. The tree topology with the lowest SCE was selected for further analysis and the multiple solutions were used to assign metastatic migration probabilities (see 'Metastatic migration probabilities').

**Subclone clonality.** Phylogenetic trees were used to classify mutation clusters as truncal or subclonal. The truncal cluster corresponded to the mutation cluster ancestral to all others, or the MRCA. Remaining clusters were classified as subclonal and their presence or absence in primary and metastasis regions was further subclassified: primary-unique (detected in the primary, undetected in any metastasis), metastasis-unique (detected in one or more metastases, undetected in the primary) or shared subclonal (detected in the primary tumour and in one or more metastases).

Somatic mutations, SCNAs and WGDs were classified as truncal, shared subclonal, primary-unique or metastasis-unique on the basis of clusters they were assigned to by CONIPHER[63], ALPACA[41] and ParallelGDDetect[19], respectively.

**Inference of subclone proportions.** Subclone proportions per sample were inferred from the mutation cluster cancer cell fraction (CCF) and the phylogenetic tree topology. Leaf node CCFs represent the terminal subclone proportions. Internal node proportions were calculated by subtracting the summed CCFs of the descendant clusters from the parent cluster CCF iteratively from the leaf nodes to the trunk. Subclones with proportions ≤5% were considered extinct (>5% were termed extant).

## Inference of subclone copy-number profiles
ALPACA[41] uses the subclonal and phylogenetic structure of tumours derived from SNV frequencies to infer subclone-specific copy-number profiles. ALPACA was run with default settings, using as input the phylogenetic tree and subclone proportions derived from CONIPHER, the allele-specific fractional copy-number estimates from Refphase[62], and estimated confidence intervals. The burden of SCNAs per subclone was computed as the total number of break points detected per subclone and the ratio of SNVs to SCNAs was used to quantify the rate of SCNA acquisition per subclone.

## Detection of driver alterations
Somatic mutations were annotated using OncoKB[30,31], openCRAVAT[66] (https://www.opencravat.org/) and the Ensembl Variant Effect Predictor (v.114)[67]. Mutations were classified as putative drivers if they fulfilled any of the following criteria: (1) classified as a loss-of-function event by LOFTEE[68] in a gene annotated as a TSG in the COSMIC Cancer Gene Census (v.102)[69] (https://cancer.sanger.ac.uk); (2) called by SpliceAI[70] (using a threshold of 0.8) in a gene listed in COSMIC (v102); (3) predicted to be a driver mutation by BoostDM[71]; (4) classified as a driver by CHASMplus[72] at a false discovery rate < 0.05, using the histology-specific models for LUAD and LUSC tumours; or (5) annotated as oncogenic in OncoKB[30,31].

OncoKB was further used to assign therapeutic levels of clinical actionability (levels 1–3B)[30,31].

Putative SCNA drivers were defined by intersecting loci of significant amplification or deletion identified using GISTIC (v.2.0)[73] in a study of 1,000 lung cancer tumours[74] with COSMIC Cancer Gene Census genes associated with mutation types A (amplification) or D (deletion). The oncogenes *ALK*, *TERT* and *FGFR1* and TSGs *MAP2K4* and *TSC1* were also considered. Analyses were restricted to alterations most likely to have a functional consequence: focal amplifications in oncogenes and LOH affecting TSGs.

### Biallelic hits affecting TSGs
Potential biallelic inactivation of TSGs was assessed for each driver mutation. Subclone-specific copy numbers inferred by ALPACA[41] were used to determine whether LOH occurred in the same, ancestral or descendant subclone relative to the SNV, allowing LOH→SNV, SNV→LOH or coincident events.

Tumour samples containing subclones with two or more events affecting a TSG were further evaluated with a binomial test of the null hypothesis that all remaining allele copies were mutated. The expected variant allele frequency (VAF) of the SNV under the null hypothesis was calculated as

$$\mathrm{VAF}_{\mathrm{expected}} = \frac{m \times \mathrm{CCF} \times \rho}{2(1-\rho) \, + \, \mathrm{CN}_\mathrm{T}\rho},$$

where $\rho$ denotes tumour purity, CCF the cancer cell fraction, $\mathrm{CN}_\mathrm{T}$ the tumour copy number and $m$ the mutation multiplicity; $m = \mathrm{CN}_\mathrm{T}$ under the null hypothesis in LOH regions. The cumulative distribution function (CDF) of the binomial distribution was obtained using the total coverage at the gene locus as the number of trials and the $\mathrm{VAF}_{\mathrm{expected}}$ as success probability. Finally, the $P$ value was taken as $2 \times \mathrm{CDF}$ when $\mathrm{CDF} < 0.5$, or $2 \times (1 - \mathrm{CDF})$ when $\mathrm{CDF} > 0.5$, respectively, and multiple testing correction was applied using the Holm–Sidak method. A biallelic event was called at $P < 0.05$ in any sample.

### Subclone-specific mutational signature activity
COSMIC signature activity was estimated using SigProfilerAssignment[75] for mutation clusters defined by CONIPHER. Signatures were fitted directly to clusters with 50 or more mutations. For each cluster with fewer than 50 mutations, signatures were fitted to mutations resampled (1,000 iterations) from the index cluster (60% probability), a neighbouring cluster on the phylogenetic tree (20% probability) and a cluster matching the index cluster clonality class (primary-unique, shared subclonal or metastasis-unique) (20% probability). Signature activities for these clusters were defined as the mean across bootstrap estimates. Mutation clusters with high uncertainty (standard deviation > 0.1 on the estimated activity of two or more signatures) were excluded.

**Signature prevalence and majority aetiology.** Signatures used for this analysis were in line with previously identified signatures in NSCLC[19]. Signature activity was grouped by aetiology: clock-like (SBS1 and SBS5), smoking (SBS4), APOBEC (SBS2 and SBS13), other (SBS17b) and platinum chemotherapy (SBS31 and SBS35). SBS31 and SBS35 were evaluated only in patients treated with platinum chemotherapy. Aetiology-level activity was defined as the summed activity of constituent signatures per subclone.

Aetiology prevalence refers to the proportion of subclones in which an aetiology is detected (activity ≥ 0.06). An aetiology was designated the majority aetiology of a subclone if activity exceeded 0.5; otherwise, the subclone was classified as having no majority aetiology.

**Episodic APOBEC activity.** A signature was considered to have emerged if it was inactive in the parent and active in the child subclone. Along each phylogenetic trunk-to-leaf lineage, episodic APOBEC activity was defined as either (i) activity present in the trunk and emerging at least once thereafter, or (ii) emergence occurring two or more times. This stringent definition required activity transitions from inactive to active.

**Mutational signature distance.** Mean pairwise cosine distances between signature activity vectors were calculated for each patient to compare metastasis-unique subclones with their ancestral primary subclones (trunk, shared subclones or primary-unique).

### Mutation and SCNA diversity measurement
Two genomic distance metrics were calculated per patient using either mutations (SNVs, DNVs and indels) or SCNAs. SCNA diversity was defined as the segment-length-weighted L1 norm between allele-specific fractional copy number for each pair of tumour regions, normalized by ploidy to account for WGD. SNV diversity was defined as the L1 norm between the SNV CCF values for each pair of tumour regions. Truncal events were included only if the MRCA was inferred as the seeding subclone. Truncal SCNAs comprised segments with a gain or loss relative to ploidy present in all samples; truncal SNVs were those assigned to the MRCA by CONIPHER[63].

Intra-primary and intra-metastasis heterogeneity were quantified as the mean of the pairwise SNV or SCNA diversity measurements between regions of the same primary tumour or regions from a single metastasis, respectively. Inter-metastasis heterogeneity was quantified as the mean of the pairwise SNV or SCNA diversity measurements between all anatomically distinct metastases per patient. Primary–metastasis heterogeneity was quantified as the mean of the pairwise SNV or SCNA diversity measurements between all primary and metastasis regions per patient.

### Inferring metastatic migration patterns
Metastatic migrations were reconstructed by applying MACHINA[8] to CONIPHER[63]-derived phylogenies in the polytomy-resolution mode. Three seeding models of increasing complexity were evaluated: primary-only seeding, single-source metastasis-to-metastasis seeding and multi-source metastasis-to-metastasis seeding. The most parsimonious solution was selected per patient. Primary reseeding was not considered, because primaries were resected before detectable metastatic relapse. Inferred migrations specified the tumour in which the migration originated, the tumour in which the migration terminated and the subclones that migrated. Where MACHINA inferred a metastasis-unique subclone (detected in metastases and undetected in the primary tumour according to sample CCFs) to be present in the primary (reflecting the most parsimonious solution), these subclones were reclassified as shared subclonal for downstream analyses. Ten regions from ten metastases failed copy-number quality control and were excluded from tree building and migration analyses.

**Metastatic migration probabilities.** To account for phylogenetic uncertainty, MACHINA was run (as above) on the 100 lowest-SCE phylogenies per patient enumerated by CONIPHER. Migration probabilities were defined as the proportion of solutions in which a given migration was observed.

**Conserved LOH.** LOH events (which are not input to MACHINA) were used to assess the accuracy of migration inferences. Assuming that LOH events are irreversible, clonal LOH (that is, present in all cancer cells of a sample) detected in samples A and B (conserved LOH) was considered compatible with a migration occurring from A to B, whereas clonal LOH detected in A but not in B was not (non-conserved LOH).

For all possible primary–metastasis and metastasis–metastasis pairs, allele-specific clonal LOH segments were identified using a purity-adjusted fractional copy-number threshold < 0.1, and the

fraction of LOH segments conserved across the genome was calculated, normalized by the total number of segments with LOH. Pairs were categorized according to whether or not a migration was inferred between them.

### Assessment of duration in situ
The number of mutations and SCNAs private to a metastasis that descended from the subclone(s) that seeded that metastasis were used as a surrogate for its duration in situ.

**'Sufficient' duration for metastasis-to-metastasis seeding.** The number of mutations accumulated before the emergence of metastasis-to-metastasis seeding subclones was used a molecular proxy for 'sufficient' duration in situ for metastasis-to-metastasis seeding. For each patient, the 90th percentile of the number of mutations accumulated before the emergence of metastasis-to-metastasis seeding subclones was calculated. The mean of this distribution was used as a patient-specific threshold to indicate 'sufficient' time in situ for metastasis-to-metastasis seeding. Metastases with total mutation counts below this threshold were classified as having 'insufficient' time in situ for metastasis-to-metastasis seeding (Extended Data Fig. 6).

### Comparisons of seeding and non-seeding subclones
Subclones inferred by MACHINA[8] to migrate were defined as seeding subclones. These were classified as primary-to-metastasis or metastasis-to-metastasis seeding subclones according to the migration source. All other subclones were defined as non-seeding subclones (Fig. 5a). Truncal clones (both seeding and non-seeding) were excluded from seeding versus non-seeding subclone comparisons.

### Subclone metastatic capacity
Metastatic capacity was approximated by the number of metastases seeded per subclone, reasoning that seeding several metastases might be indicative of greater metastatic capacity than seeding one or few. For each patient, we tested whether primary-to-metastasis subclones seeded a comparable number of metastases under a multinomial model using Monte Carlo testing ($P > 0.05$ indicating compatibility with equal probabilities). An analogous analysis assessed whether descendants of primary-to-metastasis subclones were equally likely to give rise to metastasis-to-metastasis seeding.

### Statistical information
Statistical analyses were performed in R or Python. Two-tailed Wilcoxon tests were used for distribution comparisons (paired (most commonly within patient comparisons): Wilcoxon signed-rank; unpaired: Mann–Whitney $U$), unless otherwise specified. Group comparisons used two-tailed Fisher's exact tests, binomial tests, Monte Carlo likelihood ratio tests or LME models, as appropriate. Sample sizes are depicted and/or stated for all analyses.

### Reporting summary
Further information on research design is available in the Nature Portfolio Reporting Summary linked to this article.

### Data availability
The WES data derived from the TRACERx and PEACE studies and analysed in this work have been deposited in the European Genome–Phenome Archive (EGA), which is hosted by the European Bioinformatics Institute (EBI) and the Centre for Genomic Regulation (CRG), under accession code EGAS00001008217 and the associated dataset EGAD00001015763; access is controlled by the PEACE data access committee. Details on how to apply for access are available on the linked pages. All processed data used to create the figures are available via Zenodo at https://doi.org/10.5281/zenodo.15755949 (ref. 76).

### Code availability
The code for the alignment pipeline and the subsequent processing pipeline can be found on GitHub (https://github.com/FrancisCrickInstitute/peace-alignment and https://github.com/FrancisCrickInstitute/peace-pipeline, respectively). All code to reproduce figures is available via Zenodo at https://doi.org/10.5281/zenodo.15755949 (ref. 76).

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

**Acknowledgements** S.H. is supported by the National Institute for Health and Care Research (NIHR) UCLH Biomedical Research Centre and a NIHR Clinical Lectureship, was previously supported by a Cancer Research UK (CRUK) City of London Clinical Research Training Fellowship and received funds from the Rosetrees Trust. A.B. is supported by a CRUK UCL Centre Non-Clinical Training Award (CANTAC721\100022). C.F.H. is supported by the NSERC doctoral postgraduate scholarship, the Data Sciences Institute at the University of Toronto doctoral fellowship and Mitacs Globalink Research Internship funding. C.G. is a CRUK Clinical Research Training Fellow, supported by the CRUK City of London Centre Award (SEBCATP-2024/100003). C.M.-R. is supported by the Rosetrees Trust (M630) and the Wellcome Trust. P. Pawlik is supported by CRUK. M.S.H. is supported by CRUK (TRACERx: C11496/A17786). O.L. is supported by a Clinical Lectureship at UCL funded by the CRUK Lung Cancer Centre of Excellence and previously by a Wellcome Trust Clinical Research Fellowship (225491/Z/22/Z). T.K. is supported by the Japan Society for the Promotion of Science (JSPS) Overseas Research Fellowships programme (202060447). A.J.R. is supported by the Francis Crick Institute. M.D.F. is supported by the NIHR UCL–UCLH Biomedical Research Centre and runs early-phase studies in the NIHR UCLH Clinical Research Facility supported by the UCL Experimental Cancer Medicine Centre (ECMC). S.M.L. is partially supported by the NIHR UCL–UCLH Biomedical Research Centre. M.G.K. is supported by the NIHR Manchester Biomedical Research Centre (NIHR203308). The study was supported by the NIHR Manchester Clinical Research Facility (NIHR203956). F.H.B. is supported by the NIHR Manchester Clinical Research Facility. C.T.H. is supported by the NIHR UCLH Biomedical Research Centre. D.A.M. is supported by the CRUK Lung Cancer Centre of Excellence (C11496/A30025). S.Z. is a CRUK Career Development Fellow (RCCCDF-Nov21\100005) and is further supported by the Rosetrees Trust (M917). N.M. receives funding from CRUK (DRCPFA-Nov23/100003) and has received funding from the Wellcome Trust and the Royal Society (211179/Z/18/Z) relevant to this work. N.M. also receives funding from the CRUK Lung Cancer Centre of Excellence, the Rosetrees Trust and the NIHR UCLH Biomedical Research Centre. C.S. is a Royal Society Napier Research Professor (RSRP\R\210001). His work is supported by the Francis Crick Institute, which receives its core funding from CRUK (CC2041), the UK Medical Research Council (CC2041) and the Wellcome Trust (CC2041). C.S. is funded by CRUK (TRACERx (C11496/A17786), PEACE (C416/A21999) and the CRUK Cancer Immunotherapy Catalyst Network); the CRUK Lung Cancer Centre of Excellence (C11496/A30025); the Rosetrees Trust; the Butterfield and Stoneygate Trusts; the Novo Nordisk Foundation (ID16584); a Royal Society Professorship Enhancement Award (RP/EA/180007 and RF\ERE\231118); the NIHR UCLH Biomedical Research Centre; the UCL ECMC; the Breast Cancer Research Foundation (US) (BCRF-23-157); a CRUK Early Detection and Diagnosis Primer Award (grant EDDPMA-Nov21/100034); and a Mark Foundation for Cancer Research ASPIRE Award (grant 21-029-ASP) and ASPIRE Phase II award (grant 23-034-ASP). C.S. is in receipt of an ERC Advanced Grant (PROTEUS) from the European Research Council under the European Union's Horizon 2020 research and innovation programme (grant agreement 835297). M.J.-H. received a CRUK Career Establishment Award and has received funding from CRUK, the IASLC International Lung Cancer Foundation, the

Lung Cancer Research Foundation, the Rosetrees Trust, the UK and Ireland Neuroendocrine Tumour Society (UKI NETS) and the NIHR UCLH Biomedical Research Centre. The TRACERx study (NCT01888601) is sponsored by UCL (UCL/12/0279). TRACERx is funded by CRUK (C11496/A17786) and coordinated through the CRUK and UCL Cancer Trials Centre, which has a core grant from CRUK (C444/A15953). The PEACE study (NCT03004755) is sponsored by UCL (UCL/13/0165). PEACE is funded by CRUK (C416/A21999) and coordinated through the CRUK and UCL Cancer Trials Centre. This work was supported by the CRUK Lung Cancer Centre of Excellence and the CRUK City of London Centre, as well as the UCL ECMC and the NIHR UCLH Biomedical Research Centre. We acknowledge the patients and relatives who participated in the TRACERx study and the PEACE national autopsy programme. We also thank the members of the TRACERx, TRACERx EVO and PEACE consortia for participating in these studies, especially all site personnel, investigators and funders who supported the generation of the data in these studies. We acknowledge the support of staff at the Scientific Computing and Genomics science technology platforms at the Francis Crick Institute, and thank J. Brock from Research Illustration for his help.

**Author contributions** S.H., A.B., A. Huebner, K.H., K.G. and C.F.H. performed bioinformatic analyses. S.H., W.K.L., B.D. and C.T.H. analysed radiological imaging data. C.G., D.M., C.M.-R., C. Richard, O.L., O.P., K.T. and P. Pawlik assisted with bioinformatic analyses. A. Huebner, C.M.-R. and M.S.H. developed the bioinformatic pipeline to process raw data. S.H., W.K.L., T.K., C.T.H., C.N.-L., P. Prymas and A. Toncheva assisted with clinical data annotation. D.A.F., Y.S., F.H.B., J.C., K.G.B., M.D.F., S.B., D.P.-P., S.M.L., T.A., O.L., B.N., M.G.K., P. Russell, A.K. and C.T.H. coordinated clinical aspects of the TRACERx study, patient recruitment and follow-up. M.D.F., S.B., D.P.-P., S.M.L., T.A., B.N., M.G.K., J.W., J.L.Q., A. Thomas, C. Richards., S.J., P. Roxburgh and O.L. coordinated clinical aspects of the PEACE study, patient recruitment and follow-up. D.A.M. and M.S. reviewed pathology. C.N.-L., S.V., S.W., F.A., P. Prymas, A. Toncheva and A.J.R. assisted with sample collection and sample processing. D.A.M., M.M., E.B., M.F., I.P., U.M., A.G., G.L., P.O., C. Richards, S.J. and S.F. performed research autopsies. A. Hackshaw and H.B. helped oversee the running of the TRACERx and PEACE studies. S.Z., N.M., C.S. and M.J.-H. jointly designed and supervised the study. S.H., A.B., A. Huebner, K.H., K.G., C.N.-L., C.F.H., C.G., S.Z., N.M., C.S and M.J.-H. wrote the manuscript.

**Competing interests** S.V. is a co-inventor on a patent of methods for detecting molecules in a sample (US patent no. 10578620; Methods for detecting molecules in a sample). P. Roxburgh has received grant support from CRUK, Beatson Cancer Charity, Wellbeing of Women, AstraZeneca, Atrios and Tesaro; consultancy from Roche, Abbvie and MSD; and speaker fees from AstraZeneca, GSK and pharma&. M.D.F. acknowledges grant support from CRUK, AstraZeneca, Boehringer Ingelheim, MSD, Merck and NIHR; holds appointments with DRA and HCA; serves in unpaid leadership roles at the Ruth Strauss Foundation; is an advisory board member for Transgene; and has consulted for Achilles, Amgen, AstraZeneca, Bayer, BeiOne, Boxer, BMS, Celgene, EQRx, Guardant Health, GSK, Immutep, Janssen, Merck, MSD, Oxford VacMedix, Pharmamar, Regeneron, Roche, Societe Generale, Syncorp, Takeda and UltraHuman. S.B. acknowledges educational grant support from Roche. T.A. acknowledges educational grant support from Boehringer Ingelheim, Roche, Takeda and Johnson & Johnson, and has consulted for AstraZeneca, Daichii Sanko, Johnson & Johnson, Merck, Pfizer, Roche and Takeda. M.G.K. has received funding grants from BerGenBio, Novartis and Roche; advisory board or consultancy fees from Astellas, Bayer, Guardant Health, Johnson & Johnson, Roche, Seattle Genetics and Zai Lab; speaker fees from BMS, Johnson & Johnson and Roche; and support for attending meetings and/or travel from BMS, Eisai, Johnson & Johnson, Roche and Zai Lab. C. Richards. Eli Lilly (Advisory Board), Pfizer, AstraZeneca, Servier laboratories and Dako (talks), Ipsen (conference funding). D.A.F. reports grants from Aldeyra, Boehringer Ingelheim, Astex Therapeutics, Bayer, BMS, GSK, RS Oncology, Clovis, Eli Lilly, BMS, MSD and GSK, and personal fees from Atara, BMS, Boehringer Ingelheim, Cambridge Clinical Laboratories, Targovax, Roche and RS Oncology. C.T.H. has received speaker fees from AstraZeneca and Merck and research funding from Roche and AstraZeneca, and has a paid advisory role for Genesiscare UK. A. Hackshaw has received fees for being a member of independent data monitoring committees for Roche-sponsored clinical trials, and academic projects coordinated by Roche. D.A.M. reports speaker fees from AstraZeneca and Takeda; consultancy fees from AstraZeneca, Thermo Fisher Scientific, Takeda, Amgen, Janssen, MIM Software, Bristol Myers Squibb and Eli Lilly; and has received educational support from Takeda and Amgen. N.M. holds patents related to determining HLA LOH (PCT/GB2018/052004), determining B cell fraction in mixed samples (PCT/EP2024/062999), determining lymphocyte abundance in mixed samples (PCT/EP2022/070694), identifying responders to cancer treatment (PCT/GB2018/051912), targeting neoantigens (PCT/EP2016/059401), identifying patient response to immune checkpoint blockade (PCT/EP2016/071471) and predicting survival rates of patients with cancer (PCT/GB2020/050221), and has a patent pending in determining HLA disruption (PCT/EP2023/059039). C.S. acknowledges grant support from AstraZeneca, Boehringer Ingelheim, Bristol Myers Squibb, Pfizer, Invitae (previously ArcherDX; collaboration in minimal residual disease sequencing technologies), Ono Pharmaceutical, Personalis and Roche–Ventana. He is chief investigator for the AstraZeneca MeRmaiD 1 and 2 clinical trials and is the steering committee chair. He is also co-chief investigator of the NHS Galleri trial funded by GRAIL and a paid member of GRAIL's scientific advisory board. He has received consultancy fees from Achilles, Bicycle Therapeutics (paid board member, and chair of clinical advisory group), Genentech, Medicxi, China Innovation Centre of Roche (CICoR; formerly Roche Innovation Centre – Shanghai), Metabomed (until July 2022), Relay Therapeutics (scientific advisory board member), Saga Diagnostics (scientific advisory board member) and the Sarah Cannon Research Institute. C.S. has received honoraria from Amgen, AstraZeneca, Bristol Myers Squibb, GlaxoSmithKline, Illumina, MSD, Novartis, Pfizer and Roche–Ventana. C.S. has previously held stock or options in GRAIL, currently has stock or options in Bicycle Therapeutics and Relay Therapeutics and previously held stock and was co-founder of Achilles Therapeutics. C.S. declares patent applications relating to methods for lung cancer detection (PCT/US2017/028013 and US20190106751A1); targeting neoantigens (PCT/EP2016/059401); identifying patent response to immune checkpoint blockade (PCT/EP2016/071471); identifying patients who respond to cancer treatment (PCT/GB2018/051912); determining HLA LOH (PCT/GB2018/052004); predicting survival rates of patients with cancer (PCT/GB2020/050221); methods and systems for tumour monitoring (PCT/EP2022/077987); and analysis of HLA allele transcriptional deregulation (PCT/EP2023/059039). C.S. is an inventor on a European patent application (PCT/GB2017/053289) relating to assay technology to detect tumour recurrence. This patent has been licensed to a commercial entity and under their terms of employment, C.S. is due a revenue share of any revenue generated from such licence(s). M.J.-H. has received funding from CRUK, the National Institutes of Health (NIH) National Cancer Institute, International Association for the Study of Lung Cancer (IASLC) Foundation, Lung Cancer Research Foundation, Rosetrees Trust, UKI NETs and NIHR; has consulted for Astex Pharmaceutical and Achilles Therapeutics; was a member of the Achilles Therapeutics scientific advisory board and steering committee; and has received speaker honoraria from Pfizer, Astex Pharmaceuticals, Oslo Cancer Cluster, Bristol Myers Squibb and Genentech. M.J.-H. is listed as a co-inventor on a European patent application relating to methods to detect lung cancer (PCT/ US2017/028013); this patent has been licensed to commercial entities, and, under terms of employment, M.J.-H. is due a share of any revenue generated from such licence(s). M.J.-H. is also listed as a co-inventor on the GB priority patent application (GB2400424.4) titled 'Treatment and prevention of lung cancer'.

**Additional information**
**Correspondence and requests for materials** should be addressed to Simone Zaccaria, Nicholas McGranahan, Charles Swanton or Mariam Jamal-Hanjani.

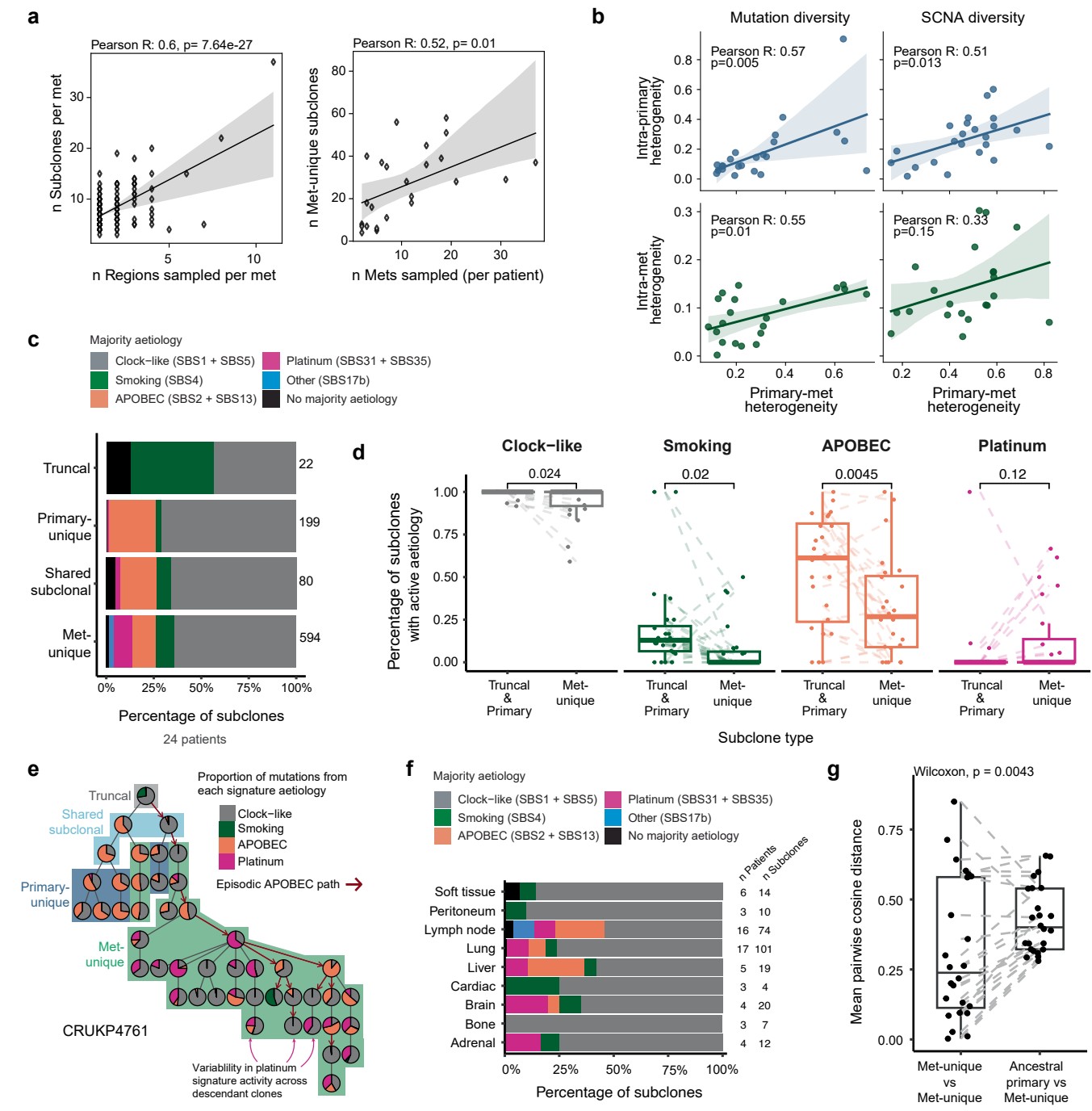

**Extended Data Fig. 1 | Mutational processes evolve during metastatic progression. a**, Correlation between the number of regions sampled and the number of subclones detected per metastasis (left), and between the number of anatomically distinct metastases sampled and the number of metastasis-unique subclones detected per patient (right). **b**, Correlation of primary–metastasis heterogeneity (x-axis) with intra-primary (dark blue) and intra-metastasis (dark green) heterogeneity based on mutation or SCNA diversity. **c**, Prevalence of mutational processes over time, measured as the percentage of truncal, primary-unique, shared subclonal and metastasis-unique subclones in which a given signature accounts for >50% of mutations (ie. the majority aetiology). **d**, Percentage of primary (truncal, shared subclonal and primary-unique) and metastasis-unique subclones per patient in which each signature

aetiology was detected. Dashed lines connect patients. Wilcoxon signed-rank test. **e**, The phylogenetic tree and estimated signature activities for each subclone (node) expressed as the fraction of mutations assigned to each signature (pie wedges) for CRUKP4761. Branches coloured based on the presence (bold red) or absence (black) of episodic APOBEC mutagenesis (Methods). **f**, The majority aetiology of metastasis-unique subclones that are unique to a single anatomical location. **g**, Mean cosine distance per patient between signature compositions of metastasis-unique subclones and their ancestral primary subclones (shared subclonal). Dashed lines connect patients. Wilcoxon signed-rank test. The box plots show the median and IQR with whiskers denoting values within 1.5 times the IQR from the first and third quartiles.

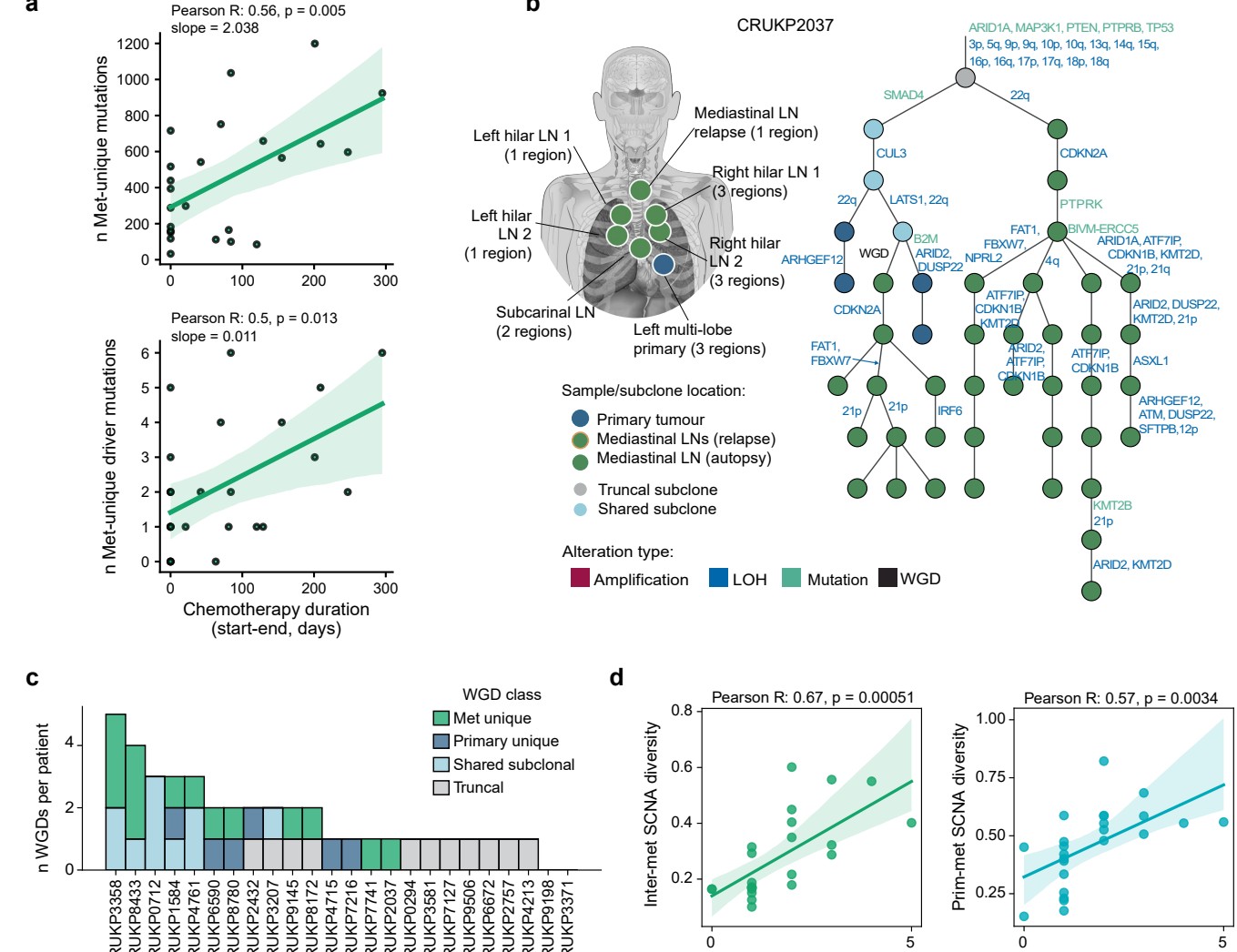

**Extended Data Fig. 2 | Effects of treatment and genome doubling on metastases. a**, Association between cumulative chemotherapy duration (sum of days across all adjuvant and metastatic regimens) and the number of metastasis-unique mutations (top) and metastasis-unique driver mutations (bottom). 24 patients. **b**, CRUKP2037 body map and phylogenetic tree showing the evolutionary timing and anatomical distribution of driver mutations (SNVs, indels and DNVs, teal), focal LOH affecting TSGs and arm-level LOH

events, culminating in biallelic disruption of *MAP3K1* (5q), *PTEN* (10q), *TP53* (17p) and *B2M* (15q) (blue) and genome doubling (WGD, black) events. **c**, Number and evolutionary timing of WGD events per patient. **d**, Correlation between the number of WGD events and inter-metastasis heterogeneity (green) or primary–metastasis SCNA diversity (blue) per patient. Body map illustration in **b** by J. Brock adapted from ref. 11 under a Creative Commons licence CC BY 4.0.

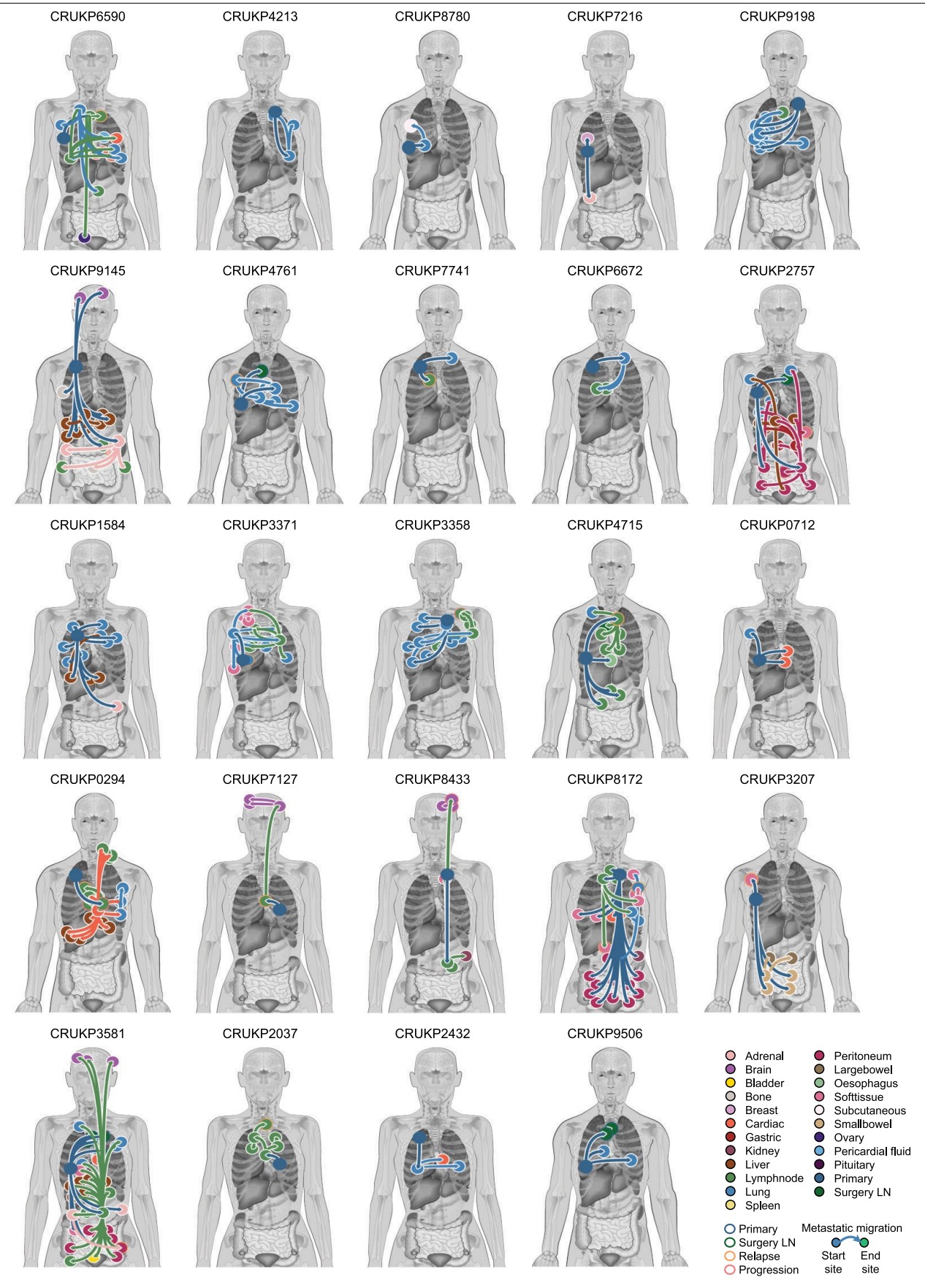

**Extended Data Fig. 3 | Metastatic migration patterns.** Body maps depicting the anatomical site of origin and the site of the metastasis seeded for each inferred metastatic migration (arrows, coloured according to the migration starting site). Body map illustration by J. Brock adapted from ref. 11 under a Creative Commons licence CC BY 4.0.

**Extended Data Fig. 4** | See next page for caption.

Extended Data Fig. 4 | Origin and number of seeding subclones per migration. **a**–**c**, The number of subclones (indicated by the number of small circles), their anatomical site of origin and the site of the metastasis seeded (indicated by the colour of the small circle and large circle respectively) are displayed for migrations (arrow) involving a single subclone only (**a**), multiple subclones that all originated from the same site (**b**) and multiple subclones that originated from different anatomical sites (**c**). Migrations are distinguished based on the site of origin of the subclones involved (primary-to-metastasis migrations; black arrow, metastasis-to-metastasis; grey arrow, migrations involving subclones from both the primary and a metastasis; yellow) and are ordered according to whether the seeding subclones originated in the same organ as the metastasis and by the number of seeding subclones. Subclones linked by a line originated from the same metastasis.

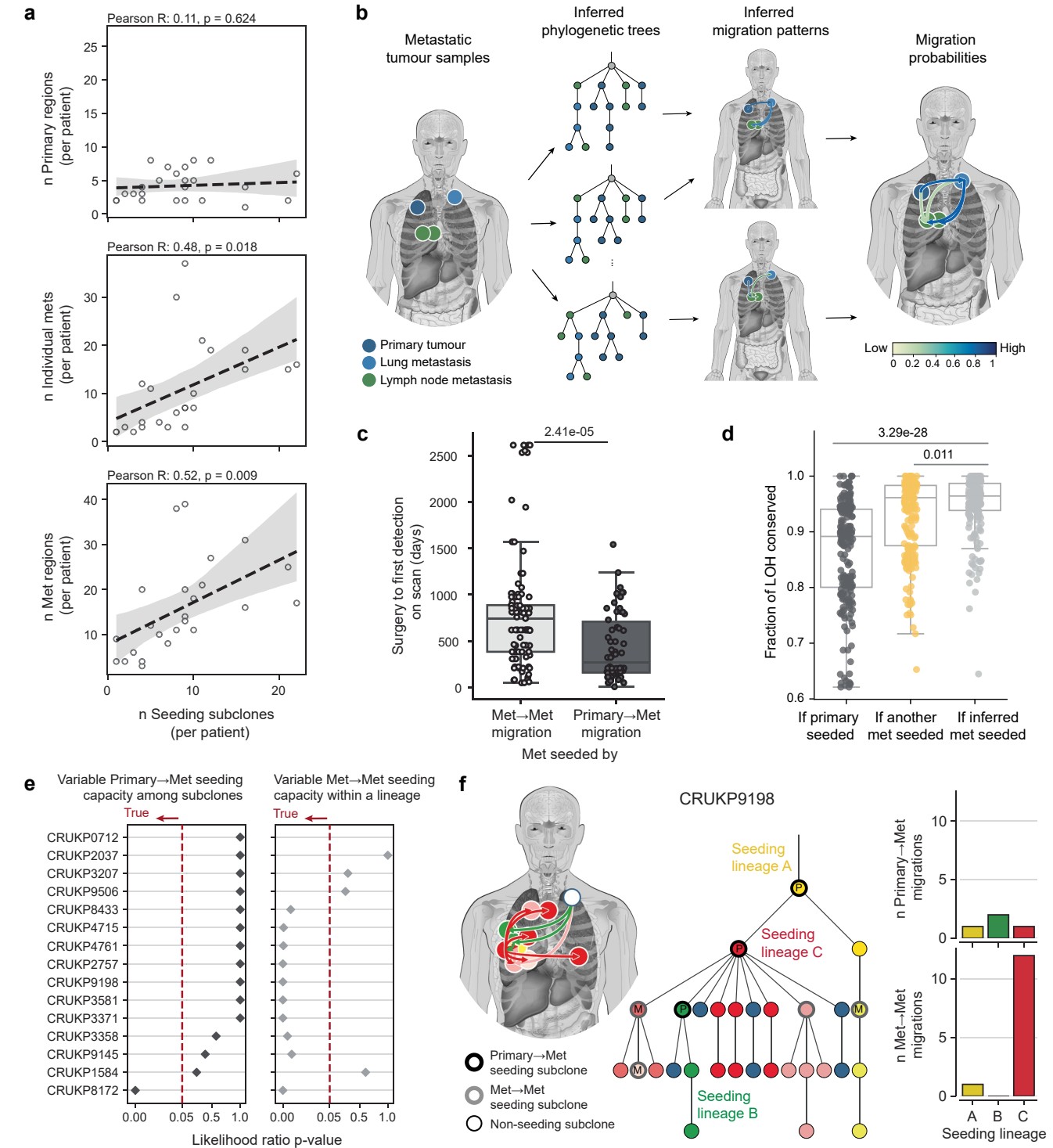

**Extended Data Fig. 5 | Orthogonal evidence for metastasis-to-metastasis seeding. a**, Correlation between sampling extent (number of primary regions, individual metastases, and metastasis regions per patient) and the number of seeding subclones identified per patient. **b**, Schematic of metastatic migration probability estimation. For each patient, MACHINA was run on the 100 most plausible phylogenetic tree topologies. The frequency of solutions supporting each migration was used to derive migration probabilities. **c**, Days from surgery to first radiological detection of metastases seeded by the primary tumour (black) or another metastasis (grey). 23 patients. Mann–Whitney *U* test. **d**, Fraction of LOH events conserved between each metastasis and its inferred seeding source (grey), an alternative metastasis (yellow), or the primary tumour (black). 24 patients. Mann–Whitney *U* test. **e**, Monte Carlo likelihood

ratio tests of the null hypotheses that primary-to-metastasis seeding subclones from the same primary tumour seed an equal number of metastases (left) and give rise to an equal number of metastasis-to-metastasis migrations (right). **f**, Phylogenetic tree and metastatic migration patterns for CRUKP9198 showing 3 primary-to-metastasis seeding subclones (lineages A–C). Although each primary lineage seeded a similar number of metastases (top), lineage C generated substantially more metastasis-to-metastasis migrations than A or B (bottom). The box plots show the median and IQR with whiskers denoting values within 1.5 times the IQR from the first and third quartiles. Body map illustrations in **b**,**f** by J. Brock adapted from ref. 11 under a Creative Commons licence CC BY 4.0.

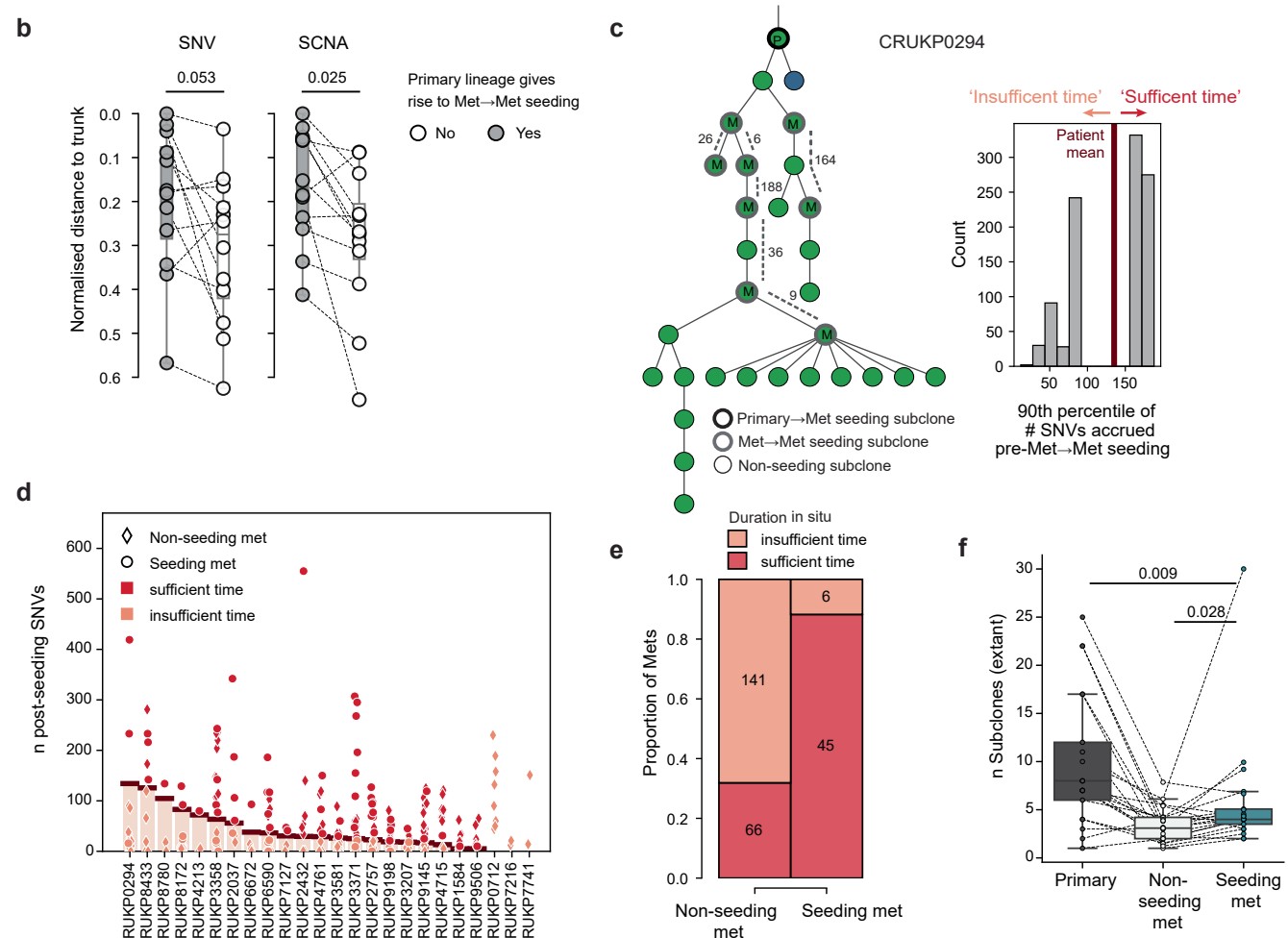

**a**

| | Characteristic | | | | | | | | | | | | | |
|---|---|---|---|---|---|---|---|---|---|---|---|---|---|---|
| | Sex | | Age | Smoking Status | | | Adjuvant treatment | TNM Stage | | | Histology | | | Disease free survival (days) | Overall survival (days) |
| | Female | Male | | Never Smoked | Ex-Smoker | Smoker | | IA-B | IIA-B | IIIA-B | LUAD | LUSC | Other | | |
| <50% Met→Met N = 10[1] | 5 (50%) | 5 (50%) | 74 (65, 80) | 0 (0%) | 6 (60%) | 4 (40%) | 3 (30%) | 1 (10%) | 4 (40%) | 5 (50%) | 3 (30%) | 3 (30%) | 4 (40%) | 167 (148, 330) | 462 (228, 968) |
| >50% Met→Met N = 14[1] | 9 (64%) | 5 (36%) | 65 (59, 74) | 2 (18%) | 7 (64%) | 2 (18%) | 10 (71%) | 3 (21%) | 4 (29%) | 7 (50%) | 6 (43%) | 7 (50%) | 1 (7.1%) | 414 (308, 582) | 969 (858, 1598) |
| p-value[2] | 0.7 | | 0.2 | 0.5 | | | 0.1 | 0.9 | | | 0.2 | | | 0.036 | 0.019 |

[1] n (%); Median (Q1, Q3)  [2] Fisher's exact test; Mann-Whitney U test

**Extended Data Fig. 6 | Metastasis-to-metastasis seeding associates with time in situ. a**, Clinical features of patients with predominant (>50% of total migrations, n = 14) or non-predominant (<50%, n = 10) metastasis-to-metastasis migrations. **b**, Number of SNVs and SCNAs accrued between the trunk and primary-to-metastasis seeding subclones that did (grey) or did not (white) give rise to subsequent metastasis-to-metastasis seeding, normalized by the total number of subclonal SNVs or SCNAs in the primary tumour, 12 patients with both types of subclones, one-sided Wilcoxon signed-rank test. **c**, Strategy to estimate the patient-specific threshold for sufficient time in situ for metastasis-to-metastasis seeding (example: patient CRUKP0294). The 90th percentile of mutations (numbers) accrued in the interval (dashed lines) between metastasis founding and metastasis-to-metastasis seeding subclone emergence (grey outlined nodes) was resampled to generate a distribution of the expected mutation burden accumulated before onward seeding (histogram). that

represents the likely number of mutations that would be accrued in a metastasis prior to it seeding another metastasis. The post-seeding mutation burden of each metastasis was compared with the patient-specific distribution mean (dark red) to classify each metastasis as having sufficient or insufficient time in situ to seed another metastasis. **d**, Classification of seeding (circles) and non-seeding (diamonds) metastases as having sufficient (red) or insufficient (peach) time in situ based on the patient-specific threshold (dark red line). **e**, Proportion of non-seeding and seeding metastases with sufficient or insufficient duration in situ for metastasis-to-metastasis seeding. **f**, Mean number of extant subclones per patient in the primary tumour, non-seeding metastases, or seeding metastases. Mann–Whitney U test. The box plots show the median and IQR with whiskers denoting values within 1.5 times the IQR from the first and third quartiles.

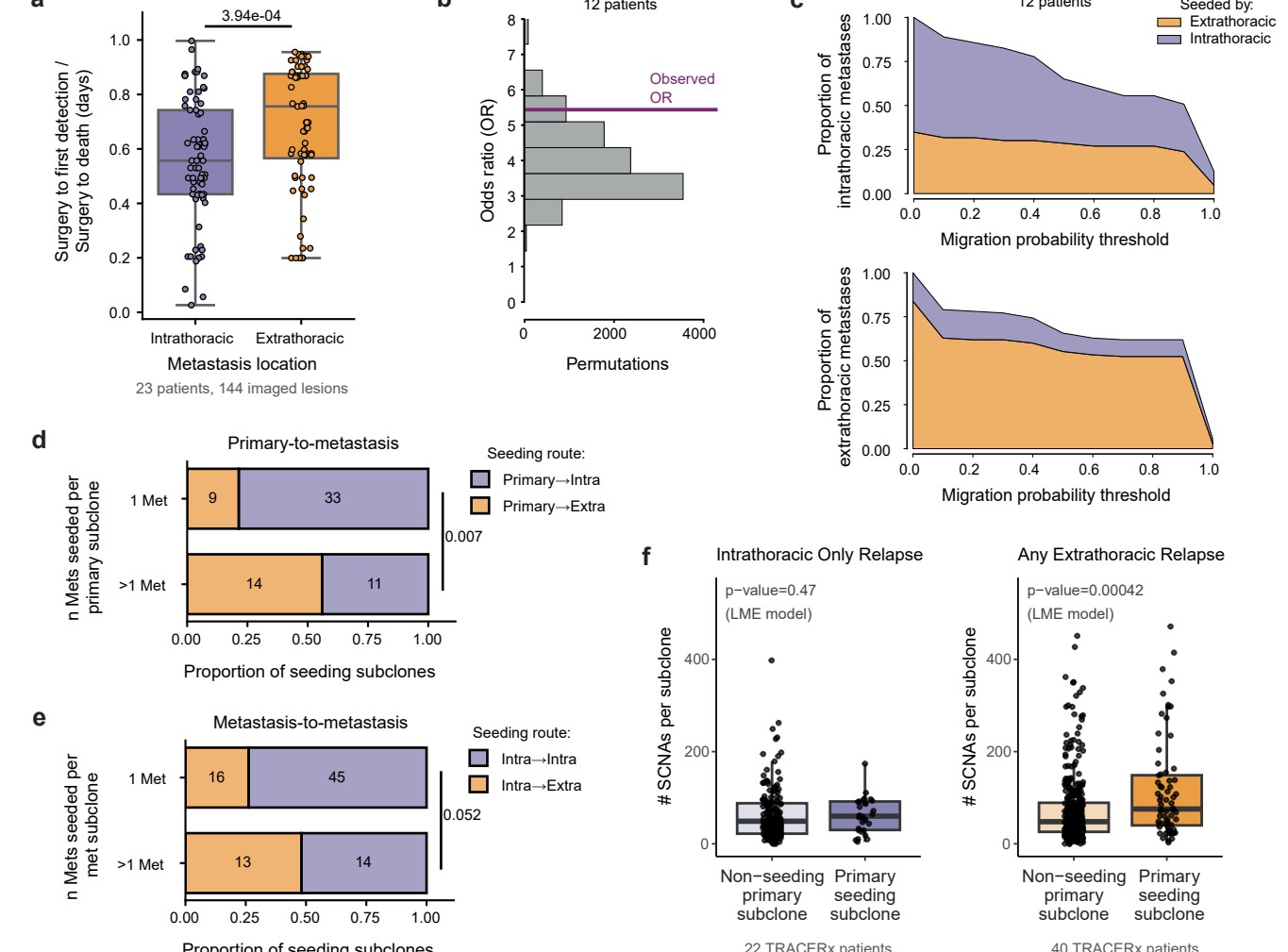

**Extended Data Fig. 7 | Features of intrathoracic and extrathoracic metastases. a**, Days from surgery to first detection of metastasis on imaging, normalized by survival from surgery, for intrathoracic (n = 73) and extrathoracic (n = 71) metastases from 23 patients. Mann–Whitney *U* test. **b**, Observed Fisher's exact test odds ratio (OR) between metastasis location (intrathoracic or extrathoracic) and seeding source (intrathoracic or extrathoracic) compared with a null distribution from 10,000 within-patient permutations of seeding source. The observed OR (purple) exceeds what would be expected by chance (grey). Only patients with both intrathoracic and extrathoracic metastases included (n = 12). **c**, Proportion of intrathoracic (top) and extrathoracic (bottom) metastases seeded by intrathoracic (purple) or extrathoracic (orange) metastases across increasing migration probability thresholds (0–1).

**d,e**, Proportion of subclones that seeded one or multiple (>1) metastases stratified by location of the metastasis seeded (intrathoracic, purple; extrathoracic, orange) for primary-to-metastasis seeding subclones (n = 24 patients, **d**) and metastasis-to-metastasis seeding subclones from intrathoracic metastases (n = 20 patients, **e**). Fisher's exact test. **f**, Site of relapse based on radiological imaging was known for 62 of the 126 patients in the published TRACERx 421 relapse cohort. SCNA burden of seeding and non-seeding primary subclones was compared for patients with intrathoracic-only relapse (n = 22, left) and patients with ≥1 extrathoracic metastasis at relapse (n = 40, right). LME model with subclone mutation count as covariate and patient as random effect. The box plots show the median and IQR with whiskers denoting values within 1.5 times the IQR from the first and third quartiles.

Nicholas McGranahan
Charles Swanton

# Reporting Summary

## Statistics

For all statistical analyses, confirm that the following items are present in the figure legend, table legend, main text, or Methods section.

| n/a | Confirmed | |
|---|---|---|
| ☐ | ☒ | The exact sample size (*n*) for each experimental group/condition, given as a discrete number and unit of measurement |
| ☐ | ☒ | A statement on whether measurements were taken from distinct samples or whether the same sample was measured repeatedly |
| ☐ | ☒ | The statistical test(s) used AND whether they are one- or two-sided<br>*Only common tests should be described solely by name; describe more complex techniques in the Methods section.* |
| ☐ | ☒ | A description of all covariates tested |
| ☐ | ☒ | A description of any assumptions or corrections, such as tests of normality and adjustment for multiple comparisons |
| ☐ | ☒ | A full description of the statistical parameters including central tendency (e.g. means) or other basic estimates (e.g. regression coefficient) AND variation (e.g. standard deviation) or associated estimates of uncertainty (e.g. confidence intervals) |
| ☐ | ☒ | For null hypothesis testing, the test statistic (e.g. $F$, $t$, $r$) with confidence intervals, effect sizes, degrees of freedom and $P$ value noted<br>*Give P values as exact values whenever suitable.* |
| ☒ | ☐ | For Bayesian analysis, information on the choice of priors and Markov chain Monte Carlo settings |
| ☒ | ☐ | For hierarchical and complex designs, identification of the appropriate level for tests and full reporting of outcomes |
| ☐ | ☒ | Estimates of effect sizes (e.g. Cohen's *d*, Pearson's *r*), indicating how they were calculated |

*Our web collection on statistics for biologists contains articles on many of the points above.*

## Software and code

Policy information about availability of computer code

| Data collection | No software was used to collect data |
|---|---|
| Data analysis | The code for the alignment pipeline and the subsequent processing pipeline can be found on github (https://github.com/FrancisCrickInstitute/peace-alignment) and (https://github.com/FrancisCrickInstitute/peace-pipeline), respectively. All code to reproduce analyses and figures is available at https://doi.org/10.5281/zenodo.15755949 |

For manuscripts utilizing custom algorithms or software that are central to the research but not yet described in published literature, software must be made available to editors and reviewers. We strongly encourage code deposition in a community repository (e.g. GitHub). See the Nature Portfolio guidelines for submitting code & software for further information.

## Data

Policy information about availability of data

All manuscripts must include a data availability statement. This statement should provide the following information, where applicable:

- Accession codes, unique identifiers, or web links for publicly available datasets
- A description of any restrictions on data availability
- For clinical datasets or third party data, please ensure that the statement adheres to our policy

The whole exome sequencing data collected from the TRACERx study and PEACE study used in this manuscript has been deposited at the European Genome–Phenome Archive (EGA), which is hosted by the European Bioinformatics Institute (EBI) and the Centre for Genomic Regulation (CRG) under accession code EGAS00001008217 and the associated dataset EGAD00001015763; access is controlled by the PEACE data access committee.

# Field-specific reporting

Please select the one below that is the best fit for your research. If you are not sure, read the appropriate sections before making your selection.

☒ Life sciences    ☐ Behavioural & social sciences    ☐ Ecological, evolutionary & environmental sciences

For a reference copy of the document with all sections, see nature.com/documents/nr-reporting-summary-flat.pdf

# Life sciences study design

All studies must disclose on these points even when the disclosure is negative.

| | |
|---|---|
| Sample size | This manuscript focuses on 24 patients with non-small cell lung cancer (NSCLC) enrolled in the TRACERx lung study and the PEACE study, which is an autopsy programme.<br>501 tumour regions were sampled longitudinally, encompassing 108 primary tumour regions, 41 regions from metastases sampled pre-mortem and 352 regions from metastases sampled at autopsy. |
| Data exclusions | Patients were recruited to the TRACERx study according to the inclusion and exclusion criteria detailed in Reporting Summary of https://doi.org/10.1038/s41586-023-05783-5.<br><br>Patients were recruited to the PEACE study according to the following inclusion and exclusion criteria:<br>Inclusion criteria<br>- Age 18 years or older<br>- Confirmed diagnosis of any form of solid malignancy with metastatic disease (where the site of origin is known or unknown), with the exception of primary brain tumour in which there may not be evidence of metastatic disease<br>- Oral and written informed consent from patient to enter the study and to undergo tissue harvesting after death or informed consent from a nominated representative or a person in a qualifying relationship after the patient has died.<br>Exclusion criteria<br>- Medical or psychiatric condition that would preclude informed consent<br>- History of intravenous drug abuse within the last 5 years<br>- Confirmed diagnosis for high-risk infections (e.g. HIV/AIDS-positive, hepatitis B/C, tuberculosis and Creutzfeldt-Jacob disease) unless patient case is of a particular scientific interest and agreed in advance with local mortuary staff and pathologist.<br><br>Supplementary Figure 1 contains a flow chart that details the samples taken forward for sequencing and the samples of sufficient quality to be included in downstream analyses. |
| Replication | The TRACERx study and PEACE study are prospective, longitudinal, observational studies. As such, the results in this manuscript are not the result of an experimental set up. |
| Randomization | Not applicable in these observational studies. |
| Blinding | Not applicable in these observational studies.<br>Patients were not allocated to an intervention and were followed up according to routine clinical practice.<br>Molecular profiling results were not fed back to patients. As such, there is no risk in this information influencing their behaviours or outcomes. |

# Reporting for specific materials, systems and methods

We require information from authors about some types of materials, experimental systems and methods used in many studies. Here, indicate whether each material, system or method listed is relevant to your study. If you are not sure if a list item applies to your research, read the appropriate section before selecting a response.

## Materials & experimental systems

| n/a | Involved in the study |
|---|---|
| ☒ | Antibodies |
| ☒ | Eukaryotic cell lines |
| ☒ | Palaeontology and archaeology |
| ☒ | Animals and other organisms |
| ☐ | ☒ Human research participants |
| ☐ | ☒ Clinical data |
| ☒ | Dual use research of concern |

## Methods

| n/a | Involved in the study |
|---|---|
| ☒ | ChIP-seq |
| ☒ | Flow cytometry |
| ☒ | MRI-based neuroimaging |

# Human research participants

Population characteristics | Supplementary Table 1 summarises the clinical and demographic information for all 24 patients included in this study.

Recruitment | For the TRACERx study, when patients are initially diagnosed with stage I-III lung cancer and then referred for surgical resection, a research nurse identifies them on a clinic/operating list. The patient has an initial eligibility assessment and is then provided with written information about the TRACERx study and the contact details of the research nurse/practitioner to ask any questions.

For the PEACE study, patients with metastatic disease are approached sensitively during routine clinic appointments or during inpatient hospital admissions (where appropriate) about their interest in participating in a research autopsy programme. If interested, they are provided with written information about the PEACE study and the contact details of the research nurse/practitioner to ask any questions. Notification of a PEACE participant's death is performed by family members or individuals in a qualifying relationship who are also provided the contact details of the research nurse/practitioner.

Ethics oversight | The TRACERx study was approved by the NRES Committee London with the following details:
Study title: TRAcking non small cell lung Cancer Evolution through therapy (Rx)
REC reference: 13/LO/1546
Protocol number: UCL/12/0279
IRAS project ID: 138871

The PEACE study was approved by the NRES Committee London with the following details:
Study title: PEACE (Posthumous Evaluation of Advanced Cancer Environment) Study
REC reference: 13/LO/0972
Protocol number: UCL/13/0165
IRAS project ID: 125424

Written informed consent was obtained from all participants (separately for the TRACERx study and the PEACE study).

Note that full information on the approval of the study protocol must also be provided in the manuscript.

# Clinical data

Clinical trial registration | TRAcking non small cell lung Cancer Evolution through therapy (Rx); NCT01888601
PEACE (Posthumous Evaluation of Advanced Cancer Environment) Study; NCT03004755

Study protocol | https://clinicaltrials.gov/study/NCT01888601
https://clinicaltrials.gov/study/NCT03004755

Data collection | Clinical and pathological data were collected from the patients from the time of they enrolled in the TRACERx study through to death in accordance with the TRACERx study protocol. Data collection was overseen by the sponsor of the study (Cancer Research UK & UCL Cancer Trials Centre) and takes place in hospitals across the United Kingdom. A centralised database called MACRO is used for this purpose.

Outcomes | The pre-defined outcome measures for the TRACERx study are detailed in the Reporting Summary of https://doi.org/10.1038/s41586-023-05783-5.
By enabling extensive tumour sampling at the time of death, the PEACE study builds on the longitudinal samples collected through the TRACERx study for patients who participated in both studies. The analyses conducted in this manuscript are hypothesis generating.

