## [Peer Review File · Nature]

Evolutionary characterisation of lung cancer metastasis

Corresponding Author: Professor Mariam Jamal-Hanjani

Version 0:

Reviewer comments:

Referee #1

(Remarks to the Author)

This manuscript presents a comprehensive evolutionary analysis of metastatic progression of non-small cell lung cancer (NSCLC). The investigators have obtained samples from primary and metastatic sites (resection and biopsy specimens while alive and post-mortem) from 23 patients/subjects. The analysis based on whole exome sequencing of multiple samples from primary and metastatic sites (dominant ones- lung, lymph nodes and liver; infrequent- bones notably). The paper combines the efforts of the TRACERx and PEACE programs. Using phylogenetic reconstruction, the authors map metastatic dissemination, genomic alterations associated with seeding potential between primary and metastatic sites and across various metastatic sites and provide insights into the role of chromosomal instability in metastatic process. This paper leverages the strengths of the TRACERx program and the knowledge gained from previous studies from this group. The major strengths of this paper, apart from the tour de force comprehensive analyses using whole exome, the use of metastatic disease samples for genomic analyses. Published work from The Cancer Genome Atlas and other projects have largely focused on primary untreated surgically resected samples. Our knowledge of genomic alterations in metastatic tumor samples from patients with solid tumor malignancies (the common cause of cancer related morbidity and mortality) is limited. This paper adds a significant and rich body of new knowledge to the field. That said, there are some concerns with the paper as written.

1. The analysis is restricted to whole exome sequencing. The limitations associated with lack of analyses of tumor microenvironment should be acknowledged
2. While this is a commendable effort, nearly 50% of patients with NSCLC present with metastatic disease and would not have been eligible for the study. That important distinction should be acknowledged as well.
3. A more detailed description of signature assignment when both overlap- like APOBEC/platinum-based signature should be provided. It is not clear what the dominant signature in metastatic disease is- did the dominant signatures (apart from platinum signature) vary depending on the sites of metastasis, latency from primary- metastases?
4. A clearer description of what the autopsy component analyses provided that would be clinically relevant compared to the Tracer X data (ante-mortem) would be useful since the TRACERx has data from metastatic samples at least in some instances.
- 5 With regard to the statement that the degree of primary-metastasis genetic divergence did not differ according to the systemic treatment patients received (lines 212-213- Supplementary Figure 3) one possible explanation is that the exposure to targeted therapy is limited in this cohort. Despite the impressive number of tissue samples, no of patients in this study is still relatively small.
6. A cross validation with other algorithms (Phylogic NDT, Canopy etc.) would strengthen the observations from MACHINA algorithm
7. The EGFR mutations reported as an example (EGFR 84Q and EGFR G452A) are not canonical mutations known to play

an oncogenic role in lung cancer.

8. It is noted that the most (83%) patients harbored one metastasis unique driver alteration and nearly 1/4 of all driver alterations present only in metastatic tumor specimens. While this is an important observation, there should be some discussion on actionable alterations.

9. It is not clear that some metastases are fated to be source, and others sink. It appears from their own data, it is a question of time for a sink metastasis to be yet another source. Given the limited amount of tumor burden the host can live with, there will always be a trail here with some metastases, the newer born ones, will appear as "sink" metastases- an illusion of sorts. This section should be carefully rewritten or more strongly supported and eloquently defended to provide an alternate point of view than this reviewer's. We ought to be careful before the terms such source and sink metastases (as appealing as they are) are established without strong foundational data.

10. Finally there should be some detailed discussion of specific biological pathways enriched in these metastases prone clusters that are shared across various organ sites/unique to certain organ sites.

(Remarks on code availability)

Referee #2

(Remarks to the Author)

This is an impressive, well-designed and -executed, study. The novelty in terms of biological insights over scattered previous work is limited, but the systematic execution and massive analyses put findings in a structured setting and convincingly confirm existing hypotheses and quantifies different metastatic seeding mechanisms. The study does provide only limited mechanistic insights in the biological underpinnings of the metastatic seeding patterns that are observed, but despite that, this work is of interest for a broad public.

Novelty:

The primary novelty of this study is the use of a uniquely comprehensive dataset with extensive sampling across metastatic sites, enabling high-resolution quantification of metastasis-to-metastasis (met-to-met) seeding. The authors report that over 60% of metastases may arise from other metastases rather than the primary tumor, with notable differences between intra-thoracic and extra-thoracic sites—the latter more often seeded directly by the primary tumor. While the concept of met-to-met seeding is not entirely new, the resolution achieved here is unprecedented due to the number of metastases profiled.

Major comment:

That said, the reliance on whole-exome sequencing (WES) imposes limitations, particularly given the relatively low number of mutations and copy number alterations detectable compared to whole-genome sequencing (WGS). Given the importance for the main conclusions that are coupled to the inferences that are made, it would be valuable to assess whether the conclusions hold in a subset of cases profiled with WGS. Alternatively, a bit more validation of their analytical/bioinformatic framework to infer met-to-met versus prim-to-met seeding approach could be provided.

Minor comments

- The authors suggest that high CNV burden in the primary tumor is associated with a greater likelihood of seeding to extra-thoracic sites. However, it's unclear how feasible it is to detect high CNV burden reliably from a single biopsy of the primary tumor. Given spatial heterogeneity, what is the probability that a single sample would be representative enough for such an inference? Or are all the primary tumor subclones high CNV on such cases? Or is this claim based on data from within patients, showing systematically that the primary subclones that seed metastases are those that have highest CNV burden? Do the connecting lines in figure 5b indicate metastases from the same patient and is the conclusion based on this figure? In that case, I am not convinced that the conclusion really holds up (and definitely not in a diagnostic setting as suggested in line 610) as about the same number of lines go up and down in the second panel top row. I think this means that in about the same number of patients either the highest or the lowest CNVburden subclones are the main seeding source.

- Can the authors clarify whether there were cases in which metastases likely originated from a different, unsampled primary? Were these cases removed?

- Can you explain the decrease in SMOKING signature? I guess this references to relative decrease (not absolute?) due to increase of other mechanisms, but the authors should clarify

- In line 217 there is a claim about cumulative heterogeneity. It appears to me that this conclusion depends very much on the number of primary and metastatic biopsies that are evaluated and that a conclusion can only be justified when these numbers are equal. So instead of cumulative heterogeneity a balanced comparison is probably better suited.

- Lines 222-226. This has been reported before and should be referenced properly.

- The section Putative driver alterations is confusing to me and it should be considered framing this differently. The problem is that the authors do not clearly discriminate between mutations in driver genes (which can be drivers and passengers) and driver gene mutations. This latter category only includes those mutations that highly likely contribute to tumorigenesis, e.g. hotspot mutations in oncogenes and biallelic inactivation of tumor suppressors. These analyses are in there, but I think too much importance (and incorrect conclusions) are connected to just the mutation in driver genes analyses

- The conclusion in line 559 is just one possible mechanism. The way it is written now, it suggests that CIN is actively involved (directly or indirectly) in the ability of cells to migrate. It is also very well possible that the cell population diversity

caused by CIN is more likely to yield a cell with the right characteristics to seed and/or survive seeding. Consider rephrasing to accommodate both a direct and indirect effect of CIN.

Code:

Description of methods are insufficient to reproduce variant calling and filtering (e.g. lines 859-863 is insufficient to reproduce the exact same indel calls from the source sequencing data). I recommend creating a GitHub repository that separates source code and pipelines/scripts from intermediate data files. This would help clearly distinguish the analysis logic from the data itself. Additionally, including a README file with step-by-step instructions for processing the data (including settings of critical tool parameters that were not default settings) and reproducing the figures would be needed. At the moment, it's necessary to manually navigate through many scripts and infer which corresponds to each analysis, which could be streamlined significantly.

Line 874: it is unclear what the parameter setting was purity. Do they mean minimum purity to be 0.5? This would be very stringent and would exclude many samples from calling. When it is meant that for varscan it was assumed that purity was 0.5 and that this setting was set by the user and not empirically determined it should not only be explained why this was done, but also what the consequences of this could be, particularly for determining (sub)clonality of variants, which is an essential basis for most of the paper.

Looking at the data analysis approach and filtering choices made, it is not unlikely that recall and precision are dependent on the purity of the sample. This is not at all analysed or discussed in the manuscript. Have the authors attempted to determine the tumor purity from the WES data and are there biases between primary and metastatic lesions (or intra and extra thoracic) and are there correlations between tumor purity and CNA, SNV burden and subclonal counts.

Lines 899-901: this is a strange way of determining DNVs. Why not use direct observations based on co-occurrence in the same raw read and/or phasing information provided by standard calling tools?

Line 906: considering replacing blacklist with the more neutral term exclusion list

Line 955-958: why not use a phylogenetic approach? Or edit distance?

Data:

EGAS00001008217 is not yet publicly available so it could not be checked if all relevant data will be available through this route.

Figures:

There are many (interesting) comparisons and statistics. However, for most comparisons, it does not become clear from the figures how many patients or sample were in each group. I do acknowledge that adding the n everywhere could impact readability, but some figures would really benefit from addition of this (e.g. fig 1d, extended fig 1a and d, fig 2b and d, extended fig 2b, fig 5b, d and e)

Edwin Cuppen

(Remarks on code availability)

I did a superficial assessment of the code. No new tools were developed or reported.

Parameter and running settings of the open source tools used are insufficiently detailed to allow complete reproduction of results. This needs some attention as detailed in the comments above

Referee #3

(Remarks to the Author)

This is an important study that sheds significant light on the evolution of metastases in non-small cell lung cancer. One of the unique aspects of this study is the longitudinal collection and follow-up of patients from diagnosis to death, in addition to the large number of tumor samples/patients from multiple organs and even from the same organ/area of metastases. The availability of clinical data including therapies received adds to the clinical relevance of the paper. The authors use standard well defined techniques to assess genomic evolutionary and metastatic migration patterns and use complementary technologies to confirm their conclusions. While there are limitations to limited spatial sampling, the multiple samples from one site when available in some patients tries to address that limitation. one limitation is the lack of characterization of the tumor microenvironment and the selective pressures that can have on metastatic evolution even without the acquiring of new mutations. The study does not look at some of the epigenetic mechanisms that can impact evolutionary changes. regardless overall I think this work sheds new knowledge on the metastatic process and the genetic processes that mediate that. The quality of the data and figures is great.

(Remarks on code availability)

i did not review the code

Version 1:

Reviewer comments:

Referee #1

(Remarks to the Author)

I appreciate the thoughtful response to my critique with detailed responses. Not surprisingly the group known for its rigor and creativity has done a stellar job. I worry that the terms source and sink will be overly eagerly accepted by the larger community given the great regard the community has with this group of investigators. I would be more comfortable if the author removes these terms in this paper.

(Remarks on code availability)

Referee #2

(Remarks to the Author)

All my previous comments and concerns have been addressed satisfactorily. The revised manuscript has been improved significantly and I can recommend positively regarding publication.

(Remarks on code availability)

the code is much better structured and accessible now.

Referee #3

(Remarks to the Author)

The authors have addressed all prior comments in this rebuttal. They have performed extensive additional analyses, including validation using alternative computational algorithms (PhylogicNTD, MACH2) and orthogonal sequencing technologies (single-cell whole genome sequencing). Key clarifications include detailed explanations of mutational signature analysis, enhanced characterization of the relationship between chromosomal instability and metastatic seeding patterns, and important new findings regarding clinically actionable alterations. The authors also strengthened their methodological transparency by creating new GitHub repositories for their analysis pipelines and improving code documentation. Overall, the revisions substantially enhance the manuscript's rigor, reproducibility, and clarity while maintaining focus on the study's central findings regarding metastatic progression in non-small cell lung cancer.

(Remarks on code availability)

Evolutionary characterisation of lung cancer metastasis

We thank the reviewers for their insightful and constructive comments on our manuscript. We would like to highlight to the reviewers that we have been able to include an additional patient in the revised manuscript whose data was not available at the time of the first submission. This increases the cohort to 24 patients and all analyses and figures have been reproduced accordingly in the revised manuscript.

Referee #1 (Remarks to the Author):

This manuscript presents a comprehensive evolutionary analysis of metastatic progression of non-small cell lung cancer (NSCLC). The investigators have obtained samples from primary and metastatic sites (resection and biopsy specimens while alive and post-mortem) from 23 patients/subjects. The analysis based on whole exome sequencing of multiple samples from primary and metastatic sites (dominant ones- lung, lymph nodes and liver; infrequent- bones notably). The paper combines the efforts of the TRACERx and PEACE programs. Using phylogenetic reconstruction, the authors map metastatic dissemination, genomic alterations associated with seeding potential between primary and metastatic sites and across various metastatic sites and provide insights into the role of chromosomal instability in metastatic process. This paper leverages the strengths of the TRACERx program and the knowledge gained from previous studies from this group. The major strengths of this paper, apart from the tour de force comprehensive analyses using whole exome, the use of metastatic disease samples for genomic analyses. Published work from The Cancer Genome Atlas and other projects have largely focused on primary untreated surgically resected samples. Our knowledge of genomic alterations in metastatic tumor samples from patients with solid tumor malignancies (the common cause of cancer related morbidity and mortality) is limited. This paper adds a significant and rich body of new knowledge to the field. That said, there are some concerns with the paper as written.

R1C1: The analysis is restricted to whole exome sequencing. The limitations associated with lack of analyses of tumor microenvironment should be acknowledged

Reply: Thank you for this suggestion. To acknowledge these limitations of the current work we have added the following statements to the Discussion:

The bulk WES performed in this study is high depth, enabling detection of low frequency mutations and SCNAs, but nonetheless may underestimate the subclonal diversity compared to sequencing technologies that examine the whole genome and/or single cells. In addition, with WES alone, the resolution of mutational processes is limited and the role of structural variants, extrachromosomal DNA, the tumour microenvironment and other non-genetic processes in metastasis cannot be investigated. Our follow-on study, TRACERx EVO (NCT05628376), will endeavour to address these limitations. It aims to recruit 600 patients across the stage spectrum (I-IV) with NSCLC, SCLC or pleural mesothelioma and to perform up to 100 research autopsies using the PEACE study infrastructure, performing high depth whole genome sequencing on the longitudinally collected samples.

In forthcoming work, we apply high dimensional imaging technologies to this cohort to investigate how the patterns of metastasis inferred in this study are influenced by primary and metastasis tumour microenvironments.

Furthermore, as detailed in our reply to a comment about the limitations of WES raised by reviewer 2 (see **R2C1**), we performed a comparison of the subclone copy number profiles, tumour phylogeny and metastatic migrations inferred from single cell whole genome sequencing data from a matched subset of samples that shows high concordance with those inferred from WES in the current manuscript.

R1C2: While this is a commendable effort, nearly 50% of patients with NSCLC present with metastatic disease and would not have been eligible for the study. That important distinction should be acknowledged as well.

Reply: Thank you for highlighting the important possibility that tumours that present with *de novo* metastatic disease may have distinct biology to those that relapse following primary resection, which are the focus of this study. We now acknowledge this in the Discussion with the following text:

The degree to which the results of this study, comprised of patients who relapsed with metastatic NSCLC following primary surgery, 88% (21/24) of whom received systemic therapies, are generalisable to patients who have de novo metastatic disease at diagnosis, who are treatment naive, or who have another lung cancer subtype, such as small cell lung cancer (SCLC), is unknown.

R1C3: A more detailed description of signature assignment when both overlap- like APOBEC/platinum-based signature should be provided. It is not clear what the dominant signature in metastatic disease is- did the dominant signatures (apart from platinum signature) vary depending on the sites of metastasis, latency from primary-metastases?

Reply: Thank you for this suggestion. To ensure our findings are robust to the fact that multiple mutational signatures can be fit to each subclone, in the revised manuscript we assess the prevalence of each mutational signature (grouped by aetiology) using two approaches. First, we identify the total number of subclones that each signature is detected with at least 0.06 activity. This is the approach taken in the original analysis corresponding to Extended Data Figure 1b. Second, we identify the 'majority aetiology' of each subclone, that is, the mutational signature that constitutes >50% of the subclones' mutations, and calculate the prevalence of each signature among truncal, shared subclonal, primary-unique and metastasis-unique subclones (**Figure R1A**). We find both approaches yield the same conclusion, that smoking and APOBEC signatures are more prevalent in primary tumours than metastases.

Figure R1: Prevalence of each majority aetiology in each subclone class. Prevalence of mutational processes over time assessed by measuring the percent of truncal, primary-unique, shared subclonal and metastasis-unique subclones in which the signature aetiology accounts for the majority (>50%) of mutations ('major aetiology').

Figure R1 has been added to Extended Data Figure 1 in the revised manuscript and the text in the corresponding results in the **Different mutational processes are active in metastases than in ancestral primary tumours** section has been updated as follows (changes in red):

To evaluate how these processes contribute to temporal and spatial genomic heterogeneity, we assessed the prevalence of mutational signatures known to be active in NSCLC in truncal, shared subclonal, primary-unique and metastasis-unique subclones (Methods). We detected a significantly higher prevalence of smoking and APOBEC signature activity in primary subclones as compared to metastasis-unique subclones. This was evident when considering either the majority aetiology of each subclone, that is, the mutational process that constituted >50% of its mutations (Extended Data Figure 1a), or the total number of subclones in which each mutational process was detected above a 0.06 threshold (mean percentage of subclones with smoking signature detected: primary 21% vs metastasis 7%, p-value = 0.015, mean percentage of subclones with APOBEC signature detected: primary 59% vs metastasis 37%, p-value = 0.0094, Wilcoxon signed-rank test, Extended Data Figure 1b).

To clarify the mutational signatures that characterise metastasis-unique subclones we have also added the following text to the same results section:

Examining the major mutational process of each metastasis-unique subclone, the mutational signatures observed in metastases, in order of prevalence, were clock-like, APOBEC, platinum and reactive oxygen species related (Extended Data Figure 1b). Platinum-related signatures SBS31 and SBS35 were detected in metastasis-unique subclones in 69% (9/13) of patients who received platinum chemotherapy either in the adjuvant or metastatic setting, but in no patient were these the most prevalent signatures detected in metastasis-unique subclones.

Furthermore, to address the reviewer’s question about metastasis site-related variation in mutational signatures we evaluated the majority aetiology, and prevalence of each mutational process in metastasis-unique subclones that were unique to a single anatomical location. Only metastasis sites that were represented in at least 3 patients were included in this analysis. Each mutational process detected in metastasis-unique subclones occurred in subclones from multiple anatomical locations. We do not observe obvious site-specific patterns in this cohort, but acknowledge that we are likely be underpowered to do so. We have added this analysis as a Supplementary Figure and the following text to the results section:

Each mutational process detected in metastasis-unique subclones occurred in subclones from multiple anatomical locations.

Figure R1B: Site-specific mutational signature activity. The percentage of metastatic-unique subclones which are unique to a single anatomical location and their signature activities according to the majority aetiology of each subclone (A) or the prevalence of aetiologies detected above 0.06 activity threshold (B).

R1C4: A clearer description of what the autopsy component analyses provided that would be clinically relevant compared to the Tracer X data (ante-mortem) would be useful since the TRACERx has data from metastatic samples at least in some instances.

Reply: Thank you for highlighting that this could be conveyed more clearly. The key distinction between the TRACERx cohort and the current TRACERx-PEACE cohort is the breadth of metastasis sampling in the latter (from different anatomical sites and where feasible multi-region sampling from individual metastases), enabled by PEACE research

autopsies. A median of 14 metastasis regions are sampled per patient (range: 3-39) in the current cohort compared to a median of 1 metastasis region per patient (range: 1-6) in the published TRACERx 421 Relapse cohort which predominantly samples at first relapse only (**Figure R2A**)¹. In addition, the current cohort includes metastases from twice as many anatomical locations (19 total) compared to what is represented by the TRACERx 421 Relapse cohort (8 total), including sites that are rarely able to be biopsied in living patients such as brain (22 regions) and cardiac metastases (15 regions). Sampling in the TRACERx-PEACE cohort allows significantly more metastasis-unique mutations, SCNAs, and subclones to be detected per patient compared to the TRACERx 421 Relapse cohort (**Figure R2B**).

We have added **Figure R2** as a supplementary figure and the following sentences in the **Genetic heterogeneity between and within primary tumours and metastases** section to highlight these differences in the revised manuscript:

Illustrative of the more comprehensive view of metastasis characteristics afforded by autopsy sampling, the number of metastasis-unique subclones identified per patient in this cohort (median [range]: 28 [4-58]) was 11 fold greater than in our previous analysis of metastasis samples collected at the time of primary surgery or relapse in 126 patients enrolled in TRACERx (median [range]: 2.5 [0-13]), where the median number of metastasis regions sampled per patient was 1 (range: 1-6) compared to 16 (range: 3-39) in this study (Supplementary Figure 3).

Figure R2: Comparison of the TRACERx-PEACE and TRACERx 421 Relapse cohorts. **A)** Overview of the number of primary and metastasis regions available per patient in each cohort, **B)** Comparison of the number of metastasis-unique subclones, somatic mutations (muts) and copy number alterations (SCNAs) analysed per patient in the TRACERx(Tx)-PEACE (purple) and TRACERx 421 Relapse (teal) cohorts.

The availability of multiple metastasis samples for all patients in this study enables several analyses that were not possible in the TRACERx 421 Relapse cohort, in which only 39% (49/126) of patients had multiple metastases sampled. These include quantifying inter-metastasis and intra-metastasis genetic heterogeneity, assessing the prevalence of metastasis-to-metastasis seeding and investigating the impact of anatomical location of metastatic seeding patterns.

Another distinguishing feature of the current cohort is that 77% (273/353) of the metastasis regions collected at autopsy were exposed to systemic anti-cancer treatment compared to

18% (39/218) of the metastasis regions collected in the TRACERx 421 Relapse cohort. This allows us to investigate the impact of treatment on metastasis genomes. For example, we found that the duration of treatment with chemotherapy is positively associated with the number of metastasis-unique mutations harboured by metastases and that metastasis-unique driver mutations are correspondingly more frequent in subclones bearing the platinum-signature (Figure 2 in the manuscript). Of clinical importance, in a new analysis performed to address a subsequent comment, we do not detect any driver mutations that would predict response to an FDA-approved drug licensed for NSCLC (ie. Level 1 actionable alterations) within metastases that are not detected in the primary tumour (see **R1C8**).

R1C5: With regard to the statement that the degree of primary-metastasis genetic divergence did not differ according to the systemic treatment patients received (lines 212-213-Supplementary Figure 3) one possible explanation is that the exposure to targeted therapy is limited in this cohort. Despite the impressive number of tissue samples, no of patients in this study is still relatively small.

Reply: We thank the reviewer for pointing this out. We agree that the absence of significant differences relating to the anti-cancer therapies patients received could indeed relate to the limited patient cohort size and heterogeneity in treatment modalities. After relapse, 21% (5/24) patients were treated with chemotherapy alone, 25% (6/24) with checkpoint inhibitors alone, 29% (7/24) with both chemotherapy and checkpoint inhibitors, 8% (2/24) with EGFR inhibitors and 29% (7/24) did not receive any systemic therapy. Of the 7 patients who did not receive systemic therapy after relapse, 4 had previously received adjuvant chemotherapy, leaving 3 patients who had not received any systemic therapy at any point. While this is broadly representative of the frequency these therapies are used in the real world setting at the time the study was conducted, we acknowledge the number of untreated patients and the number of patients in each therapy group is insufficient to draw definitive conclusions in relation to the impact of treatment on genetic divergence and metastatic seeding patterns. To clarify this, we have added the following text to **Genetic heterogeneity between and within primary tumours and metastases** section of the revised manuscript:

We did not detect differences in the degree of primary-metastasis genetic divergence according to the systemic treatment patients received (Supplementary Figure 4). Such inferences may be limited due to cohort size.

R1C6: A cross validation with other algorithms (Phylogic NDT, Canopy etc.) would strengthen the observations from MACHINA algorithm.

Reply: We thank the reviewer for this suggestion and in the revised manuscript we perform new analyses using other algorithms described below for both phylogenetic tree reconstruction and metastatic migration inference. Reassuringly, these results exhibit high consistency with the results presented in our original submission.

First, as recommended by the reviewer, we compare the phylogenetic trees obtained using our tool CONIPHER, developed specifically to reconstruct tumour phylogenies from SNVs whilst correcting for errors in input SNV counts², with those that can be obtained using PhylogicNDT³, another algorithm for phylogenetic inference. We apply PhylogicNDT with default parameters to reconstruct phylogenetic trees from the identified mutation clusters for all patients. Based on this, PhylogicNDT was able to complete phylogenetic reconstruction

only for 10/24 patients within a reasonable time limit of 24 hours (**Figure R3A**). This outcome does not change after lowering the number of iterations of the PhylogicNDT algorithm to the minimum (i.e., 10 iterations). This outcome demonstrates that CONIPHER is particularly well suited for phylogenetic analyses in this study, because it can scale to the large number of samples analysed in this cohort. This is in contrast to PhylogicNDT, and other existing algorithms that we have been previously benchmarked against CONIPHER, such as CITUP, LiCheE, and PhyloWGS².

We compare the phylogenetic trees successfully reconstructed for these 10 patients with those reconstructed using CONIPHER. Since both CONIPHER and PhylogicNDT can exclude mutation clusters during phylogenetic reconstruction, the following comparison considers phylogenetic nodes corresponding to mutation clusters preserved by both algorithms. We compare the multiple equally plausible phylogenetic trees produced by PhylogicNDT to the CONIPHER tree used in the analyses of this study (ie. the lowest SCE tree). The phylogenetic topology of each pair of these trees is assessed using the standard Robinson-Foulds similarity metric (values between 0 and 1, where 1 indicates identical trees)⁴. We exclude small clusters (i.e., <15 mutations and CCF <20% in all samples) during the calculation of this metric to minimise noise. Overall, we find that PhylogicNDT reconstructs phylogenetic trees with high similarity to the CONIPHER trees analysed in this study (**Figure R3B**, with >0.7 similarity for 8/10 patients). In addition, PhylogicNDT scores each enumerated tree by calculating its likelihood. We find that trees with the highest likelihood (i.e., top decile) are among the trees with high similarity to the CONIPHER trees analysed in this study (**Figure R3B**). For example, the highest likelihood trees reconstructed by PhylogicNDT for patients CRUKP1548 and CRUKP9198 are broadly similar to those reconstructed using CONIPHER, differing by only a few phylogenetic edges (**Figure R3C,D**). Overall, these new analyses demonstrate that the main phylogenetic patterns analysed are preserved between these two algorithms for phylogenetic inference.

Figure R3: Comparison of different algorithms for phylogenetic inference. A) Fraction of trees successfully reconstructed among the 24 patients in this cohort by using either CONIPHER in this study, or PhylogiNNT. **B)** The phylogenetic similarity is calculated using the standard Robinson-Foulds accuracy metric (y-axis) between every tree enumerated by PhylogiNNT (each dot) and the best tree analysed in this study across all patients successfully analysed by PhylogiNNT (x-axis). Trees scored with the highest likelihood (top decile) are indicated (black stars). **C-D)** Examples of the phylogenetic trees either analysed in this study (left) or reconstructed by PhylogiNNT (right) after preserving mutation clusters for two patients, **(C)** CRUKP1548 and **(D)** CRUKP9198.

Figure R4: Comparison of metastatic seeding patterns inferred using different algorithms. A) Frequency of primary-to-metastasis vs metastasis-to-metastasis migrations (grey vs black) estimated by MACHINA using either the trees analysed in this study or those inferred by PhylogiNNT on the 10 patients successfully analysed by PhylogiNNT. **B)** The precision concordance rate (y-axis) is calculated for each patient (x-axis) as the fraction of metastatic migrations that are inferred in this study which are also inferred based on the trees inferred by PhylogiNNT. **C)** Example of the metastatic migration patterns inferred by MACHINA from the tree used in this study (left) or inferred by PhylogiNNT (right). **D)** Frequency of primary-to-met vs met-to-met migrations (grey vs black) estimated by MACH2 using the trees either analysed in this study or inferred by PhylogiNNT on the same patients.

As the focus of this study is on analysing metastatic seeding events, we apply MACHINA⁵ to reconstruct the metastatic migration patterns from the PhylogenicNDT trees with the maximum similarity to the CONIPHER trees, and compare the resulting metastatic migration patterns to those analysed in this study. The frequency of metastasis-to-metastasis seeding estimated from the metastatic migration patterns derived from PhylogenicNDT trees is nearly identical to the estimate based on metastatic migration patterns derived from CONIPHER trees (54.9% vs 60% respectively, **Figure R4A**). Furthermore, we find that the two different phylogenetic algorithms lead to a high fraction of consistent metastatic migrations: on average, across patients ~80% of metastatic migrations inferred in this study are also identified by applying MACHINA to PhylogenicNDT trees (**Figure R4B**). For example, the metastatic migration patterns inferred by MACHINA using either the CONIPHER tree or the PhylogenicNDT tree for CRUKP3207 are highly concordant in terms of the routes of migrations between the primary and metastases (**Figure R4C**).

Similar to the importance of comparing different algorithms for phylogenetic reconstruction, we agree with the reviewer that it is prudent to compare different algorithms that infer metastatic migration patterns. Therefore, we apply MACH2⁶, a recent new algorithm for reconstructing metastatic migration patterns, to the same trees described above. The frequency of metastasis-to-metastasis seeding is very similar between MACHINA and MACH2 irrespective of whether PhylogenicNDT trees or CONIPHER trees are used as input (**Figure R4D**). Taken together, these new analyses demonstrate that the metastatic migration patterns analysed in this study are preserved when considering different combinations of different algorithms for phylogenetic and metastatic migration inference.

We have added the following text to the manuscript in the **Metastasis-to-metastasis seeding is pervasive in relapsed NSCLC** section and added **Figure R4** as a supplementary figure:

Furthermore, the metastatic migration patterns inferred were highly consistent with those obtained using different combinations of algorithms for phylogenetic and metastatic migration inference (Supplementary Figure 10).

R1C7: The EGFR mutations reported as an example (EGFR 84Q and EGFR G452A) are not canonical mutations known to play an oncogenic role in lung cancer.

Reply: We thank the reviewer for highlighting this point. We have corrected the annotation of the EGFR (G452A) mutation to be the clinically actionable EGFR (G719A) mutation. When mapping the actionable mutations to the transcript annotations, the first transcript available in the database is used (in this case G452A) instead of the canonical transcript (in this case G719A), which is conventionally used to describe actionable alterations. We have adjusted the mapping to always use the canonical transcript. We would like to highlight that the underlying mutation calls have not changed, only the mapping of nucleotide variants to transcripts are corrected.

In contrast to the truncal *EGFR* (G719A) mutation harboured by patient CRUKP3358, the metastasis-unique *EGFR* (E84Q) (already mapped to the canonical transcript) is not recognised in the OncoKB database^{7,8} as a biomarker for response to EGFR targeted therapy. Nonetheless it is considered a putative driver mutation. We included it as an example of the emergence of a metastasis-unique driver mutation that is likely related to treatment

exposure to EGFR inhibitors. However, we have chosen to remove the example to avoid confusion by readers that we may be suggesting it could be actionable or related to treatment resistance.

R1C8: It is noted that the most (83%) patients harbored one metastasis unique driver alteration and nearly 1/4 of all driver alterations present only in metastatic tumor specimens. While this is an important observation, there should be some discussion on actionable alterations.

Reply: We thank the reviewer for this suggestion, and as a result we conducted an additional analysis now included in the manuscript. To evaluate whether any clinically actionable, metastasis-unique alterations were present, we annotate all somatic variants using OncoKB^{7,8} (<https://www.oncokb.org/actionable-genes#sections=Tx>). This database assigns tiers of clinical actionability ranging from Level 1, denoting an FDA-recognised biomarker predictive of response to an FDA-approved therapy in the same cancer type (e.g., *EGFR* p.L858R in NSCLC), to Level 3B, denoting a biomarker associated with response to an FDA-approved or investigational therapy in a different cancer type.

Across all primary tumour and metastasis mutations, 30 OncoKB-defined actionable alterations (Levels 1–3B) are identified in 19 of 24 patient tumours, with individual tumours containing one to three actionable variants. Two tumours with actionable fusion drivers (*CD74-ROS1* in CRUKP4213 and *EML4-ALK* in CRUKP9506) harbour no actionable mutation drivers. To assess the timing of these actionable mutations we map them to the CONIPHER-derived subclones. No Level 1 actionable alterations are identified exclusively in metastasis-unique subclones while being absent from the matched primary tumour. Only one patient, CRUKP3358, harbours a potentially actionable metastasis-unique event: *SMARCA4* p.E1083K, a Level 3A alteration supported by clinical evidence but lacking an FDA-approved indication.

These findings suggest that actionable driver alterations typically arise within the primary early in tumour evolution, with little evidence that novel clinically actionable alterations emerge during metastatic progression, although we cannot exclude the possibility that a larger cohort may identify such drivers. This conclusion applies only to mutation-based actionable drivers, as fusion events are assessed solely from pooled primary-tumour RNA (method as described in Frankell et al. 2023⁹) and are not profiled in metastasis samples in this study.

We have added the following text to the manuscript in **Putative driver alterations occur in metastases** results section:

*To determine if metastasis-unique driver mutations could reveal new therapeutic opportunities for patients with advanced disease, we annotated the clinical actionability of each somatic mutation using OncoKB^{7,8} (Figure 2b). No Level 1 actionable mutations, which predict response to an FDA-approved drug licensed for management of NSCLC, were identified in metastases-unique subclones. Only one metastasis-unique mutation was annotated as potentially actionable: *SMARCA4* (p.E1083K) identified in a lung metastasis in CRUKP3358 was classified as a Level 3A alteration based on preliminary clinical trial evidence supporting as yet unlicensed *SMARCA4* targeting strategies¹⁰.*

R1C9: It is not clear that some metastases are fated to be source, and others sink. It appears from their own data, it is a question of time for a sink metastasis to be yet another source. Given the limited amount of tumor burden the host can live with, there will always be a trail here with some metastases, the newer born ones, will appear as "sink" metastases- an illusion of sorts. This section should be carefully rewritten or more strongly supported and eloquently defended to provide an alternate point of view than this reviewer's. We ought to be careful before the terms such source and sink metastases (as appealing as they are) are established without strong foundational data.

Reply: We agree with the reviewer's interpretation of the data and we are grateful for their comment. We expect that time *in situ* facilitates the occurrence of metastasis-to-metastasis seeding and, as such, recently established sink metastases might go on to become source metastases with longer duration *in situ*. To expand on this important point in the manuscript, we develop a method to delineate metastases into those with sufficient or insufficient duration *in situ* for metastasis-to-metastasis seeding albeit restricted to the timelines relevant to the disease course in each patient. Specifically, we use the number of mutations that accrued prior to the emergence of metastasis-to-metastasis seeding subclones as a molecular marker of sufficient duration *in situ* for metastasis-to-metastasis seeding. For each patient, we compute the distribution of the 90th percentile of mutations that accrued prior to the emergence of metastasis-to-metastasis seeding subclones and use the mean of this distribution as a threshold to indicate sufficient time *in situ* for metastasis-to-metastasis seeding. Metastases with fewer total mutations than this threshold are categorised as having insufficient time *in situ* for metastasis-to-metastasis seeding (**Figure R5A,B**). Based on this approach, 68% (141/207) of the metastases that did not seed other metastases may have developed the ability to do so with longer time *in situ* (**Figure R5C**). Excluding these recently established metastases, the prevalence of metastases that are observed to seed other metastases is 41% (46/112).

We have added this new analysis as a supplementary figure and have updated the text in the corresponding results section entitled **Metastasis duration *in situ* is associated with metastasis-to-metastasis seeding** with the following:

*One implication of time being a necessary condition for metastasis-to-metastasis seeding, as is suggested by these data, is that the prevalence of metastasis-to-metastasis seeding will be influenced by the number of metastases that were recently established. Indeed, of the sink metastases, 68% (141/207) had fewer mutations in total than had accrued, on average, prior to the emergence of metastasis-to-metastasis seeding subclones in the same patient (Methods, Supplementary Figure 12), suggesting longer duration *in situ* may have led to the observation of metastasis-to-metastasis seeding from these metastases. Excluding these recently established metastases, the prevalence of metastases that were observed to seed other metastases was 41% (46/112).*

We hope that this addition to the manuscript makes clear that the labels 'sink' and 'source' are descriptive terms to help identify whether metastasis-to-metastasis seeding is or is not observed. We have currently left these terms in the manuscript for ease of interpretation of their related analyses and concepts, but we are happy to remove them if the reviewer feels they are misleading despite the additional analyses.

Figure R5: Classification of metastases with sufficient or insufficient duration *in situ* for metastasis-to-metastasis seeding. **A)** Depiction of the approach taken to determine the patient-specific threshold of sufficient time for metastasis-to-metastasis seeding, exemplified by patient CRUKP0294. The 90th percentile of the number of mutations (numbers) accrued in the interval (dashed lines) between a metastasis being seeded and the emergence of a metastasis-to-metastasis seeding subclone (grey outlined nodes) are re-sampled to generate a distribution (histogram) that represents the likely number of mutations that would be accrued in a metastasis prior to it seeding another metastasis. The total number of mutations that occurred from the point each metastasis was established is compared to the mean of this distribution (dark red line) to determine whether each metastasis is likely to have had sufficient (more than the distribution mean) or insufficient (less than the distribution mean) duration *in situ* to seed another metastasis. **B)** Categorisation of each source (circle) and sink (diamond) metastasis as having had sufficient (red) or insufficient (peach) duration *in situ* for metastasis-to-metastasis seeding according to the patient specific threshold (dark red line). **C)** Proportion of sink and source metastases with sufficient or insufficient duration *in situ* for metastasis-to-metastasis seeding.

Stratifying metastases by whether they had sufficient or insufficient time for metastasis-to-metastasis seeding also provided a way to further investigate the relationship between the anatomical order in which metastases arise and the variation we observe in metastasis-to-metastasis seeding rates across different anatomical sites. We find the proportion of metastases in each anatomical location that had sufficient time *in situ* is strongly related to the prevalence of metastases in that site that seeded metastases (Figure R6). Coupled with the pre-existing analysis showing a higher prevalence of source metastases in intrathoracic sites, which arise earlier than extrathoracic sites, this new result adds strength to the finding that the tendency of metastases to arise in a conserved anatomical sequence (e.g. lung early, liver late) underpins anatomical variation in metastasis-to-metastasis seeding rates.

Figure R6: Correlation between the proportion of metastases in each site with 'sufficient' duration *in situ* and the proportion of metastases from the same site that are sources. Only anatomical sites with a minimum of 4 metastases and representation in a minimum of 3 patients were included.

We have added Figure R6 as a panel in Figure 4 and the following text to the **Anatomical boundaries influence metastasis-to-metastasis migration routes** section:

*These differences were reflective of the temporal order in which metastases arose: the prevalence of source metastases in each anatomical location was strongly related to the proportion of metastases at that site that had sufficient duration *in situ* for metastasis-to-metastasis seeding (Pearson R: 0.80, p-value = 0.001, Figure 4g, Methods).*

R1C10: Finally there should be some detailed discussion of specific biological pathways enriched in these metastasis prone clusters that are shared across various organ sites/unique to certain organ sites.

Reply:

We agree with the reviewer that elucidating differences in the biological pathways disrupted in primary tumours compared to metastases could help shed light on the cellular mechanisms critical for metastasis. To this end, we examine the prevalence of alterations in 12 cancer-related signalling pathways manually curated in by Sanchez-Vega et al.¹¹ in primary and metastasis subclones. A pathway is considered disrupted if at least one putative driver mutation affecting one of the genes in the pathway is detected in the subclone. Mutations that are not annotated as drivers and copy number events that affected the pathway genes are not considered due to insufficient evidence that these would lead to functional disruption.

Comparing primary subclones (shared subclonal and primary-unique) to metastasis-unique subclones we observe that the WNT pathway is exclusively disrupted in the metastasis-unique subclones (0.3%, 2/701) while the NRF2 pathway is exclusively disrupted in the primary subclones (0.6%, 2/333). However, no significant differences in how frequently these pathways are disrupted is detected between primary and metastasis-unique subclones (**Figure R7**), which we suspect reflects the low prevalence of these alterations and the limited

number of patients in the cohort. Three out of the twelve analysed pathways were undisrupted across the cohort, so only the other nine are represented in Figure R7.

Figure R7: Barplots showing the comparison of nine manually curated cancer signalling pathways classified as disrupted (black) or intact (grey) in each subclone according to the detection of a driver mutation affecting one of the pathway genes. P-values from chi-square tests comparing the prevalence of disrupted pathways in metastasis-unique and primary subclonal.

Referee #2 (Remarks to the Author):

This is an impressive, well-designed and -excuted, study. The novelty in terms of biological insights over scattered previous work is limited, but the systematic execution and massive analyses put findings in a structured setting and convincingly confirm existing hypotheses and quantifies different metastatic seeding mechanisms. The study does provide only limited mechanistic insights in the biological underpinnings of the metastatic seeding patterns that are observed, but despite that, this work is of interest for a broad public.

Novelty: The primary novelty of this study is the use of a uniquely comprehensive dataset with extensive sampling across metastatic sites, enabling high-resolution quantification of metastasis-to-metastasis (met-to-met) seeding. The authors report that over 60% of metastases may arise from other metastases rather than the primary tumor, with notable differences between intra-thoracic and extra-thoracic sites—the latter more often seeded

directly by the primary tumor. While the concept of met-to-met seeding is not entirely new, the resolution achieved here is unprecedented due to the number of metastases profiled.

Major comment:

R2C1: That said, the reliance on whole-exome sequencing (WES) imposes limitations, particularly given the relatively low number of mutations and copy number alterations detectable compared to whole-genome sequencing (WGS). Given the importance for the main conclusions that are coupled to the inferences that are made, it would be valuable to assess whether the conclusions hold in a subset of cases profiled with WGS. Alternatively, a bit more validation of their analytical/bioinformatic framework to infer met-to-met versus prim-to-met seeding approach could be provided.

Reply: We thank the reviewer for this comment and we agree that it is important to confirm our findings using different and more-powerful sequencing assays on a subset of patients as well as alternative bioinformatic algorithms.

Single-cell whole genome sequencing validation of metastatic migrations

To confirm the main conclusions of the current study using a different sequencing assay, we leverage a single-cell whole-genome DNA sequencing (scWGS) dataset generated in a parallel study (now published in ¹²). This dataset comprises scWGS data generated for 14,994 single cancer cells obtained from 10 samples from a patient included in the current study, CRUKP9145. The same 10 samples, encompassing five primary tumour regions and five distinct metastases (left adrenal, right adrenal, left frontal lobe, right occipital lobe and liver), are analysed with bulk WES in the current manuscript. The scWGS dataset represents the ideal resource to orthogonally confirm the results in the current study since the whole-genome resolution, as well as the single-cell resolution, provide the highest currently achievable power for cancer evolutionary and metastatic analyses^{12,13}. Lucas et al.¹² reconstruct a phylogenetic from the scWGS data using a combination of single-cell-specific methods that jointly leverages both SNVs and SCNAs and apply MACHINA to these data to infer metastatic migration patterns, allowing direct comparison with the phylogeny and metastatic migration patterns inferred using bulk WES in this study.

The metastatic migration patterns obtained from the matched scWGS dataset are highly consistent with those in the current study based on WES. In the current study, we infer that the brain, liver, and right adrenal metastases were seeded by cancer cells spreading from the primary tumour, while the left adrenal metastasis is inferred to be seeded by cancer cells spreading from the right adrenal metastasis through metastasis-to-metastasis seeding (**Figure R8A**). The scWGS analysis confirms all but one of these metastatic migrations (**Figure R8B**) despite the differences in the sequencing assay and resolution, despite scWGS data only being available for a subset of primary tumour regions and metastases analysed in the current study, and despite sequencing different cells belonging to the same tumour regions. These differences likely explain the single metastatic migration inferred differently from scWGS. Moreover, the scWGS analysis finds that one of the two brain metastases was seeded by cancer cells that spread from the right adrenal metastasis, rather than from the primary tumour (as inferred in the current study, **Figure R8B**), in line with the conclusion that metastasis-to-metastasis seeding is prevalent in NSCLC put forth in the current study.

The high concordance between the WES- and scWGS-derived metastatic migrations is explained by the fact that the phylogenetic trees inferred in each of these analyses, despite the use of different phylogenetic reconstruction algorithms, have highly similar topologies (**Figure R8A,B**). Both analyses identify the presence of two main evolutionary branches, one comprised predominantly of subclones from the liver metastases and the other including subclones from all other sites (**Figure R8A,B**). This concordance illustrates a key trade-off between the WES used in current study and WGS-based datasets: while WES only captures a fraction of SNVs that are identifiable from WGS in a tumour, the high depth sequencing coverage used in the current study (~400x) provides high power and accuracy for the detection of SNVs in exomes even if they are present at low frequency. By enabling detection of low frequency SNVs, the high depth WES in the current study allows accurate reconstruction of tumour evolution despite the total number of SNVs being lower than with WGS. As further evidence of this, the SNVs identified in the scWGS-inferred subclones are consistently most similar to the SNVs detected in the same sample using WES (**Figure R8C**) and the most similar WES-inferred and scWGS-inferred subclones are consistently identified in the same tumour regions (**Figure R8D**).

Figure R8: Validating metastatic migrations with matched single-cell whole-genome sequencing (scWGS) data. **A**) (Top) Body map representation of the metastatic migrations inferred in this study with WES (each arrow represents a metastatic seeding event, coloured according to source tissue), and (Bottom) inferred tumour phylogenetic tree, where each node is coloured by the tissue sample in which the subclone is most prevalent (colour, primary tissue is preferred to break ties). **B**) (Top) Body map representation of the metastatic migrations inferred from the matched scWGS data (each arrow represents a metastatic seeding event, coloured according to source tissue) from a subset of samples (circles), and (Bottom) inferred tumour phylogenetic tree for single-cell tumour subclones (nodes by tissue of origin). **C**) SNV distance is calculated between each sample analysed with WES (rows) and each single-cell subclone (columns) as the fraction of different SNVs. **D**) For five primary tumor samples in this study (coloured circles on photo) and three additional samples (gray circles), each bulk subclone identified in this analysis (hexagons comprising subclones with different inner shapes of size proportional to cell proportion) is assigned to the most similar single-cell subclone using SNVs (colours).

Consistency of subclonal copy number profiles between scWGS and ALPACA applied to WES

With respect to SCNAs, another parallel study from our group recently demonstrated that subclone-specific SCNAs can be accurately inferred from multi-region, bulk WES data, despite its lower breadth of sequencing coverage compared to WGS¹⁴. This was made possible by introducing ALPACA, a novel algorithm that leverages SNVs as barcodes for tumour subclones to which SCNAs can be assigned¹⁴. The accuracy of this reconstruction was demonstrated both using realistic simulations and matched scWGS datasets¹⁴. In particular, Pawlik et al.¹⁴ demonstrated high similarity between the subclone copy number profiles inferred using the scWGS available from CRUKP9145 and those inferred with ALPACA applied to SNVs trees and sample-level copy number profiles derived from WES from the same patient (mean hamming distance = 0.24, **Figure R9**). Since ALPACA has been used to reconstruct subclone-specific SCNAs from comparable, high depth, multi-region WES in the current study, the inference of SCNAs is expected to be accurate and comparable to what could be obtained from WGS assays.

Figure R9: Heatmaps showing total copy number profiles of subclones inferred from single-cell WGS (Top) compared to total copy number profiles of subclones inferred by ALPACA when applied to SNV trees and bulk copy number profiles derived from WES data from the same tumour samples from patient CRUKP9145.

Comparison of alternative bioinformatic algorithms for metastatic migration inference

We also agree with the reviewer on the value of providing further evidence to verify the computational inference of primary-to-metastasis and metastasis-to-metastasis seeding. To do this, we perform new analyses by implementing different combinations of algorithms to infer phylogenetic trees and related metastatic seeding patterns (described in detail in the reply to a similar comment by Reviewer 1, please see **R1C6**). Based on these additional analyses, we demonstrate that the metastasis seeding patterns analysed in the current study are preserved when considering different combinations of algorithms for phylogenetic and metastatic pattern inference.

Overall, the analyses performed using a matched dataset generated with higher resolution sequencing technologies like scWGS and alternative computational algorithms support the robustness of the evolutionary histories and related metastatic migration patterns that underpin the study conclusions.

Minor comments

R2C2: The authors suggest that high CNV burden in the primary tumor is associated with a greater likelihood of seeding to extra-thoracic sites. However, it's unclear how feasible it is to detect high CNV burden reliably from a single biopsy of the primary tumor. Given spatial heterogeneity, what is the probability that a single sample would be representative enough for such an inference? Or are all the primary tumor subclones high CNV on such cases? Or is this claim based on data from within patients, showing systematically that the primary subclones that seed metastases are those that have highest CNV burden?

Do the connecting lines in figure 5b indicate metastases from the same patient and is the conclusion based on this figure? In that case, I am not convinced that the conclusion really holds up (and definitely not in a diagnostic setting as suggested in line 610) as about the same number of lines go up and down in the second panel top row. I think this means that in about the same number of patients either the highest or the lowest CNV burden subclones are the main seeding source.

Reply: We thank the reviewer for raising these points which we address in turn below.

Assessment of SCNA burden and seeding robust to primary tumour sampling

This cohort includes multi-region sampling of the primary tumours: a median of 4 regions were sampled per tumour (range 1-6). Mutations and SCNAs are identified in each sampled region and these are pooled to infer the subclonal composition and related phylogeny of each tumour. In previous work we have shown that sampling 3 regions of primary NSCLC tumours captures the majority of subclones that a greater number of samples would reveal¹. In the first submission of this manuscript, we reported an association between the number of primary-to-metastasis seeding subclones and the degree of SCNA intratumour heterogeneity (ITH), captured by the fraction of the aberrant genome that is subclonal. Because both the number of seeding subclones and this measure of SCNA ITH could be affected by the number of primary tumour regions sampled, we perform a new comparison included in the current manuscript version that takes into consideration the effect of sampling. We show that both the median SCNA burden and the median rate of SCNA acquisition (per mutation) of all primary tumour subclones is strongly related to the percent of primary subclones that seeded metastases (**Figure R10**), suggesting that the degree of chromosomal instability influences the likelihood of primary subclones seeding metastases. By using a subclone-based measure of SCNA burden and taking the median of all detected subclones from all regions of each primary tumour, we expect this estimate to be representative of the SCNA burden that would be obtained from a single region biopsy.

Figure R10: Association between SCNA burden and the percent of primary subclones that seed metastases. Correlation between the percent of primary tumour subclones that seed metastases and the median burden SCNAs per subclone for each primary tumour (left) or the median SCNA acquisition rate (SCNA/SNV) per subclone for each primary tumour.

We add **Figure R10** as a new panel in Figure 5 of the revised manuscript and add the following text to the **Chromosomal instability influences metastatic seeding capacity** section:

In addition, the percent of primary subclones that seeded metastases was strongly associated with the median SCNA burden (Pearson R : 0.62, p -value=0.001) and rate of SCNA acquisition across all primary subclones (Pearson R : 0.64, p -value= 8.4×10^{-4} , Figure 5C), suggesting the degree of chromosomal instability harboured by primary subclones is indicative of their likelihood of seeding metastases.

Pairwise comparison of seeding and non-seeding subclone SCNA burdens

Another way that we demonstrate the association between chromosomal instability and metastatic seeding is to compare the SCNA burden of primary subclones that seed metastases to those that do not from the same primary tumour. We are grateful that the reviewer highlighted that there are counter examples in this analysis, because further investigation of these examples has revealed a new observation. We find that primary tumours that seed intrathoracic metastases, but not extrathoracic metastases, account for the vast majority of counter examples where the median SCNA burden of non-seeding subclones is similar or in some cases larger than in seeding subclones (**Figure R11**). In the previous version of the manuscript we reported that primary tumour subclones that seed extrathoracic metastases had significantly higher SCNA burdens than those that seeded intrathoracic metastases, but we did not compare subclones that seed intrathoracic metastases and those that seed extrathoracic directly to non-seeding subclones from the same tumour. Doing so reveals that the enrichment SCNA burden observed in primary-to-metastases seeding subclones is driven by subclones that seed extrathoracic metastases. Meanwhile no significant difference is detected in the SCNA burdens of primary subclones that seed intrathoracic metastases and non-seeding primary subclones (**Figure R11**).

Additionally, we confirm this observation within the TRACERx 421 Relapse cohort (**Figure R12**).

Figure R11: Paired comparison of primary subclones that seed intrathoracic or extrathoracic metastases. The SCNA burden of non-seeding primary subclones (blue), primary-to-metastasis seeding subclones that seed intrathoracic metastases (purple) or extrathoracic metastases (orange) are compared within primary tumours that either seed both intrathoracic and extrathoracic metastases (left), primary tumours that only seed intrathoracic metastases (middle) and primary tumours that only seed extrathoracic metastases (right). The mean SCNA burden of all subclones in each category for each patient is plotted with connecting lines between patients. P-values are derived from an LME model that includes patient as a fixed effect.

Figure R12: SCNA burden of seeding and non-seeding subclones in the TRACERx 421 relapse cohort stratified by sites of metastases. Of the 126 patients in this separate published cohort¹, the 62 patients with both subclonal primary seeding and non-seeding subclones were included in this analysis. Patients with intrathoracic only relapse had only intrathoracic metastases on their relapse imaging while patients with Any extrathoracic relapse had at least 1 extrathoracic metastasis with or without concurrent intrathoracic metastases. P-values are derived from an LME model that includes patient as a fixed effect.

We have integrated these new results into the revised manuscript and re-written the corresponding results section **Chromosomal instability influences metastatic seeding** (not copied here due to the extent of the re-write, but all changed text is highlighted in red in the revised manuscript). We have also updated the Discussion to highlight this finding and the hypotheses it generates.

R2C3: Can the authors clarify whether there were cases in which metastases likely originated from a different, unsampled primary? Were these cases removed?

Reply: We follow the same process as described in Frankell et al. 2023⁹ to determine whether primary and metastasis regions sampled were genomically related. Mutations are used to classify all samples from each patient as genomically related or unrelated. If more than 10 mutations are shared across all samples, the samples are defined as related, that is, all having arisen from a single tumour. If fewer than 10 mutations were shared across all samples, hierarchical clustering is performed in a stepwise manner to determine the relationship between samples. With this approach, we identify three samples collected from three different patients that are unrelated to all other samples collected from these patients.

We have updated the Genomically independent tumours section of the **Methods** to provide further details of these cases:

In CRUKP1584, a synchronous lung tumour resected at the time of primary surgery was found to be unrelated to the other tumour resected at the same time and as such was considered to be a second primary (as opposed to a related metastasis). This second primary tumour was also genomically unrelated to all metastases subsequently sampled in this patient, suggesting it did not metastasize and thus was excluded from the cohort. In two other patients, CRUKP8172 and CRUKP7741, a single presumed metastasis sampled at autopsy (a lung and oesophageal sample, respectively) was found to be genomically distinct from the patient's primary tumour and all other metastasis samples. These may represent second primaries or metastases from undetected second primaries. As there were no paired primary-metastatic samples for these genomically distinct tumours, they were excluded from the final cohort.

R2C4: Can you explain the decrease in SMOKING signature? I guess this references to relative decrease (not absolute?) due to increase of other mechanisms, but the authors should clarify

Reply: In ever-smokers, which account for 22/24 of the patients in this cohort, 84% of all mutations that are attributed to the smoking signature SBS4 were accumulated in the trunk. This is consistent with previous observations of SBS4 as an early process in NSCLC evolution¹⁵. Smoking mutations are also detected in subclones that arose later in tumour evolution, however, the proportion of subclonal mutations related to this process is relatively small and the percentage of metastasis-unique subclones in which the smoking signature is detected (>0.06 activity) is significantly lower than shared subclonal and primary-unique subclones (**Figure R1**). This most likely reflects a longer period of tobacco-smoke exposure that the trunk endures relative to subclones which may correspond to smoking cessation (80% of patients in this cohort are ex-smokers).

We add the following statement to highlight this in the revised manuscript text:

In ever-smokers, 84% of all mutations attributed to the smoking signature SBS4 were accumulated in the trunk, consistent with previous observations that SBS4 is an early process in NSCLC evolution¹⁵. The lower prevalence of smoking signature in metastases-unique subclones may reflect smoking cessation (91% of the patients in this cohort are ex-smokers).

R2C5: In line 217 there is a claim about cumulative heterogeneity. It appears to me that this conclusion depends very much on the number of primary and metastatic biopsies that are evaluated and that a conclusion can only be justified when these numbers are equal. So instead of cumulative heterogeneity a balanced comparison is probably better suited.

Reply: Thank you for raising this point regarding the comparison of primary tumour and metastasis heterogeneity. We agree that the impact of sample number on the ‘cumulative heterogeneity’ is important to account for. In this analysis we do not calculate a sum of the heterogeneity in these samples, but rather the mean per patient to mitigate the impact of the different number of samples available from each patient.

We thank the reviewer for noting the unclear terminology and we have updated the wording in the **Genetic heterogeneity between and within primary tumours and metastases** section to more accurately reflect the analysis performed (changes in red):

*Collectively, metastases from the same patient were more similar to one another than they were to the paired primary tumour (**mean** inter-metastasis vs **mean** primary-metastasis heterogeneity **per patient**: mutation diversity p -value = 0.004, SCNA diversity p -value = 6.0×10^{-5} , Wilcoxon signed-rank test, Figure 1d), as previously observed in prostate cancer¹⁶. The genetic diversity between primary-metastasis pairs was positively associated with both primary intra-tumour heterogeneity (**mean per patient**: mutation diversity Pearson’s R : 0.57, p -value = 0.005, SCNA diversity Pearson’s R : 0.51, p -value = 0.013) and metastasis intra-tumour heterogeneity (**mean per patient**: mutation diversity Pearson’s R : 0.55, p -value = 0.01, SCNA diversity Pearson’s R : 0.33, p -value = 0.15, Figure 1e), suggesting that somatic evolution in the primary, metastases, or both after metastatic dissemination contributes to this genetic divergence. We did not detect differences in the degree of primary-metastasis genetic divergence according to the systemic treatment patients received (Supplementary Figure 4). **Such inferences may be limited due to cohort size**. Individual multi-region sampled metastases were less heterogeneous than their respective paired multi-region sampled primary tumours (**mean** metastasis intra-tumour vs **mean** primary intra-tumour heterogeneity **per patient**: mutation diversity p -value = 0.03, SCNA diversity p -value = 0.004, Wilcoxon signed-rank test, Figure 1d), but when all metastases were considered, **the heterogeneity among them** was not significantly different to that within the paired primary tumour (**mean** inter-metastasis vs **mean** primary intra-tumour heterogeneity **per patient**: mutation diversity p -value = 0.44, SCNA diversity p -value = 0.67, Wilcoxon signed-rank test, Figure 1d).*

R2C6: Lines 222-226. This has been reported before and should be referenced properly.

We apologise for this oversight and agree that there have been many studies highlighting the importance of wide-spread metastatic sampling for a detailed reconstruction of ongoing

tumour evolution. We have added three more references to this paragraph describing these approaches in breast, ovarian and prostate tumours:

1. McPherson *et al.* Divergent modes of clonal spread and intraperitoneal mixing in high-grade serous ovarian cancer, *Nature*, 2016
2. Yates *et al.* Genomic Evolution of Breast Cancer Metastasis and Relapse, *Cancer Cell*, 2017
3. Gundem *et al.* The evolutionary history of lethal metastatic prostate cancer, *Nature*, 2015

R2C7: The section Putative driver alterations is confusing to me and it should be considered framing this differently. The problem is that the authors do not clearly discriminate between mutations in driver genes (which can be drivers and passengers) and driver gene mutations. This latter category only includes those mutations that highly likely contribute to tumorigenesis, e.g. hotspot mutations in oncogenes and biallelic inactivation of tumor suppressors. These analyses are in there, but I think too much importance (and incorrect conclusions) are connected to just the mutation in driver genes analyses

Reply: We thank the reviewer for pointing out that the phrasing in this section is not clear. We focus on two types of alterations in this section: driver mutations and focal SCNAs (restricting to amplifications and LOH) affecting driver genes. We agree with the reviewers that only alterations of potential functional consequence should be considered in this analysis. To this end, we apply a strict approach for annotating somatic mutations as “driver mutations” that is based on the consensus calls of multiple state-of-the-art driver annotators. In addition, in this revised manuscript we annotate the clinical actionability of each somatic variant using OncoKB^{7,8} (please see **RC18**), which helps to distinguish that of the driver mutations that are of potential biological consequence, a small subset may have clinical actionability.

While less evidence is available to determine the impact of copy number alterations on driver gene function, we restrict the copy number driver analysis to only consider focal amplifications affecting oncogenes and focal loss of heterozygosity (LOH) affecting tumour suppressor genes as these are the most likely to have a functional impact. Furthermore, the SCNA driver gene list is restricted to the loci of significant amplification or deletion identified using GISTIC 2.0¹⁷ in a study of 1,000 lung cancer tumours with COSMIC Cancer Gene Census genes associated with mutation types ‘A’ (amplification) or ‘D’ (deletion)¹⁸. To ensure our conclusions are based only on alterations of potential functional consequence, in the revised manuscript we have removed the analyses that are solely based on copy number driver events, such as the analysis of parallel copy number evolution.

We have added the following text to the **Putative driver alterations occur in metastases** section to improve its clarity:

Annotating each somatic mutation (Methods), 185 putative driver mutations were identified including 167 SNVs, 4 DNVs, and 14 indels, of which 81% affected tumour suppressor genes (TSGs) (Figure 2a).

To determine if metastasis-unique driver mutations could reveal new therapeutic opportunities for patients with advanced disease, we annotated the clinical actionability of

each somatic mutation using OncoKB^{7,8}(Figure 2b). No Level 1 actionable mutations, which predict response to an FDA-approved drug licensed for management of NSCLC, were identified in metastases-unique subclones. Only one metastasis-unique mutation was annotated as potentially actionable: SMARCA4 (p.E1083K) identified in a lung metastasis in CRUKP3358 was classified as a Level 3A alteration based on preliminary clinical trial evidence supporting as yet unlicensed SMARCA4 targeting strategies¹⁰.

..We also examined the timing of focal amplifications and loss of heterozygosity (LOH) affecting lung cancer-related oncogenes and TSGs (Methods).

In addition, we have added the following text to the Driver annotation section of the **Methods**:

Mutations were annotated as putative drivers by running variant annotators including OncoKB^{7,8}, openCRAVAT (<https://www.opencravat.org/>), and the Ensembl Variant Effect Predictor (v114)¹⁹. A mutation was classified as a putative driver if it fulfilled any of the following criteria: (1) it was identified as a loss-of-function event by LOFTEE²⁰ in a gene annotated as a tumour suppressor gene (TSG) in the COSMIC Cancer Gene Census (v102)¹⁸(cancer.sanger.ac.uk); (2) it was called by SpliceAI²¹ (using a threshold of 0.8) in a gene listed in COSMIC (v102); (3) it was predicted to be a driver mutation by BoostDM²²; (4) it was identified as a driver by CHASMplus²³ following false discovery rate correction; or (5) it was annotated as oncogenic in OncoKB^{7,8}.

OncoKB was also used to annotate clinically actionable driver alterations with Therapeutic Levels^{7,8}. This database assigns tiers of clinical actionability ranging from Level 1, denoting an FDA-recognised biomarker predictive of response to an FDA-approved therapy in the same cancer type (e.g., EGFR p.L858R in NSCLC), to Level 3B, denoting a biomarker associated with response to an FDA-approved or investigational therapy in a different cancer type.

Furthermore, we have updated the labelling of Figure 2a to highlight that the figure includes driver SNVs and Indels in addition to the amplifications and LOH events.

R2C8: The conclusion in line 559 is just one possible mechanism. The way it is written now, it suggests that CIN is actively involved (directly or indirectly) in the ability of cells to migrate. It is also very well possible that the cell population diversity caused by CIN is more likely to yield a cell with the right characteristics to seed and/or survive seeding. Consider rephrasing to accommodate both a direct and indirect effect of CIN.

Reply: We thank the reviewer for this comment and agree our data suggest that subclones that seed extrathoracic metastases are enriched for CIN, however the way in which CIN contributes to this capacity is uncertain, and an area for future work. To reflect this, the updated **Chromosomal instability influences metastatic seeding capacity** results section includes the following text:

We observed instances of SCNAs affecting genes previously implicated in metastasis in seeding subclones^{1,14}, such as the focal amplification of CCND1 in a primary subclone that seeds a chest wall metastasis in CRUKP3207 and in a left frontal lobe brain metastasis that seeds a right frontal lobe metastasis in CRUKP8433. However, we did not find significant enrichment in the number of SCNAs affecting driver genes nor the number of driver mutations

in seeding compared to non-seeding subclones (Supplementary Figure 17), reflecting that this cohort may be underpowered to detect such differences or that CIN may support metastasis through alternative means, such as by generating subclonal diversity from which subclones with metastatic capacity emerge.

Code:

R2C9: Description of methods are insufficient to reproduce variant calling and filtering (e.g. lines 859-863 is insufficient to reproduce the exact same indel calls from the source sequencing data). I recommend creating a GitHub repository that separates source code and pipelines/scripts from intermediate data files. This would help clearly distinguish the analysis logic from the data itself. Additionally, including a README file with step-by-step instructions for processing the data (including settings of critical tool parameters that were not default settings) and reproducing the figures would be needed. At the moment, it's necessary to manually navigate through many scripts and infer which corresponds to each analysis, which could be streamlined significantly.

Reply: We created two GitHub repositories for the alignment (<https://github.com/FrancisCrickInstitute/peace-alignment>) and processing (<https://github.com/FrancisCrickInstitute/peace-pipeline>) pipelines, respectively. The former includes the alignment as well as the local INDEL realignment process which was performed based on GATK best practices while the latter includes the somatic variant calling, copy number calling, and phylogenetic reconstruction. Combined, these two pipelines take raw sequencing data as input and process them through to the final data used in this manuscript.

In addition, we have organised, labelled and more clearly documented the downstream data processing, analysis, and plotting scripts to clearly indicate the figure panels they produce and will continue to hone this further to facilitate reproducibility

R2C10: Line 874: it is unclear what the parameter setting was purity. Do they mean minimum purity to be 0.5? This would be very stringent and would exclude many samples from calling. When it is meant that for varscan it was assumed that purity was 0.5 and that this setting was set by the user and not empirically determined it should not only be explained why this was done, but also what the consequences of this could be, particularly for determining (sub)clonality of variants, which is an essential basis for most of the paper. Looking at the data analysis approach and filtering choices made, it is not unlikely that recall and precision are dependent on the purity of the sample. This is not at all analysed or discussed in the manuscript. Have the authors attempted to determine the tumour purity from the WES data and are there biases between primary and metastatic lesions (or intra and extra thoracic) and are there correlations between tumour purity and CNA, SNV burden and subclonal counts.

Reply: Thank you for this comment which we address point-by-point below.

Purity parameter settings

The Varscan purity parameter is set to 1 as default, meaning Varscan expects tumour samples to be "100% pure with 0% contaminating stromal or other non-malignant cells" (<https://varscan.sourceforge.net/using-varscan.html>). Furthermore, Varscan documentation

specifies: “You would change tumor-purity to something less than 1 if you have a low-purity tumor sample and thus expect lower variant allele frequencies for mutations”. As NSCLC samples generally have lower purity (median purity across all samples in this cohort is 0.38), the default parameter of 1 was lowered to 0.5. This parameter is only used within VarScan to detect mutations and is not the cutoff used to keep or remove samples based on overall purity. We have added the following text to the Somatic mutation calling section of the **Methods** help clarify this point:

Default parameters were used with VarScan2 with the exception of minimum coverage for the germline sample being set to 10, and minimum variant frequency being set at 0.01 and given that lung cancer samples generally have lower purity, the tumour purity parameter was set from 1 to 0.5 to better detect mutations in these low purity samples. This parameter is only used within VarScan to detect mutations, while the exact tumour purity is subsequently estimated using ASCAT

Inference of tumour purity per sampled region

ASCAT is used to determine tumour purity and whether sampled primary or metastasis regions should be passed or failed based on their tumour content. All regions with a purity ≥ 0.1 are included for analysis subject to manual review of the corresponding copy number solutions.

Associations with tumour purity

We investigated the relationship between purity and the number of subclonal mutations, SCNAs, and subclones identified in each primary and metastasis region sampled. On a region-by-region basis, we do not see an association between purity and the total number of these genomic features (**Figure R13A,B**). Likewise, the median purity of all primary tumour regions sampled per patient is not associated with the median number of primary subclonal mutations, SCNAs nor subclones detected (**Figure R13C**). The median purity of all metastasis regions sampled per patient, however, is positively associated with the median number of metastasis-unique mutations and subclones detected (**Figure R13D**).

Figure R13: Relationship between sample purity and subclonal genomic features. **A)** Linear regression of the total number of primary subclonal (primary-unique + shared subclonal) mutations, SCNAs, and subclones per region and the primary region purity. **B)** As for A) but for metastasis-unique genomic features and metastasis region purity. **C)** Linear regression of the total number of subclonal mutations, SCNAs, and subclones per patient and the median purity across all primary regions for that patient. **D)** As for C) but but for metastasis-unique genomic features and metastasis region purity. Pearson correlation co-efficient R and p-value.

To locate the greatest source of variation in purity in this cohort, so it can be accounted for, we measured the absolute difference in purity between pairs of regions. We observe the greatest differences in purity are between patients (**Figure R14A**). Within patients, the difference in primary and metastasis region purity is significantly higher than the difference among different primary or metastasis regions (**Figure R14A**). Where relevant, the analyses performed in the manuscript aim to account for patient-level effects in the data due to its nested structure, which will encompass this inter-patient variation purity.

We also assessed if the higher purity metastases with greater subclonal resolution could systematically influence the comparisons made in the manuscript. Despite the differences in purity between primary and metastasis regions within patients, across the cohort, no significant difference in purity is observed between primary and metastasis regions (**Figure R14B**). Purity varied between metastases from different anatomical locations (**Figure R14C**), but no significant difference is observed between intrathoracic and extrathoracic metastases (**Figure R14D**), nor between primary tumours that seeded extrathoracic metastases, intrathoracic metastases, or both (**Figure R14E**). Furthermore, no significant difference is detected between source and sink metastases (**Figure R14F**). Purity is therefore unlikely to confound analyses involving these comparisons.

Figure R14: Associations between purity and region characteristics. **A)** Comparison of the absolute difference in purity between pairs of regions within individual patients (dark blue) or between patients (light blue). Mann Whitney U test. **B)** Paired comparison of the median purity of all primary and all metastasis regions per patient. **C)** Median purity per metastasis region stratified by anatomical location. **D)** Paired comparison of the median purity of all extrathoracic and intrathoracic metastasis regions per patient. **E)** Purity of primary tumour regions derived from primary tumours that seed extrathoracic metastases, intrathoracic metastases or both. **F)** Paired comparison of the median purity of all extrathoracic and intrathoracic metastasis regions per patient. The lines in B,D,F connect patients.

R2C11: Lines 899-901: this is a strange way of determining DNVs. Why not use direct observations based on co-occurrence in the same raw read and/or phasing information provided by standard calling tools?

Reply: We thank the reviewer for their suggestion and have now investigated these variants in more detail. For each putative dinucleotide variant (DNV) defined in the cohort ($n = 374$) we extract read information at each position of the variant and calculate the proportion of overlap between reads at each position, excluding reads where the DNV overlaps with the beginning or end of the read and therefore do not include the full DNV. We defined a DNV to have good overlap in a specific region if more than 90% of the reads harbour the full DNV. For 323 of 374 (86.36%) of the DNVs all regions where the variant was detected have good overlap. For 41 additional DNVs (10.96%) the number of tumour samples with lower overlap is not equal to the number of samples the DNV is detected in, meaning there is at least one region with high confidence of the overlap of the DNV indicating that these variants are in fact true variants.

We check the remaining 10 variants manually in the Integrative Genomics Viewer (IGV). In 1 case the putative DNV overlaps with a poly-C stretch. Given the challenging nature of mutation calling in this repetitive region, we amend the annotation and remove this variant completely. For two additional putative DNVs we do not see clear evidence of co-occurrence on the same reads and amend the annotation accordingly to remove these variants. Finally, the remaining 7 DNVs show clear evidence of co-occurrence.

The following text has been added to the Somatic mutation calling section of the **Methods**:

Additionally, for all putative DNVs, read information was extracted and the proportion of reads overlapping between the bases was calculated. DNVs were called in cases where at least 90% of reads harboured both variants in at least one sample of the patient.

R2C12: Line 906: considering replacing blacklist with the more neutral term exclusion list

Reply: Thank you for this suggestion. We have replaced this terminology with “problematic genomic regions” in the revised manuscript.

R2C13: Line 955-958: why not use a phylogenetic approach? Or edit distance?

Reply: The approach used to determine distinct tumours is very similar to a phylogenetic approach. We use hierarchical clustering based on the presence/absence of mutations across all samples collected from a given patient. The resulting dendrogram is cut into two branches which results to determine the regions associated with two distinct tumours. These results are then manually checked and confirmed.

Data:

R2C14: EGAS00001008217 is not yet publicly available so it could not be checked if all relevant data will be available through this route.

Reply: We can assure the reviewer and editor that this repository containing the raw sequencing data will be made public if the manuscript is accepted. The processed data derived from these raw sequencing data is available in the Zenodo which is now structured in a much more navigable format.

Figures:

R2C15: There are many (interesting) comparisons and statistics. However, for most comparisons, it does not become clear from the figures how many patients or sample were in each group. I do acknowledge that adding the n everywhere could impact readability, but some figures would really benefit from addition of this (e.g. fig 1d, extended fig 1a and d, fig 2b and d, extended fig 2b, fig 5b, d and e)

Reply: Thank you for this suggestion. Figure panels where this is prudent have been updated to more clearly convey the number of patients and samples included in the analysis, including:

Figure 1d (plotted)
Extended Data 1a (plotted)
Extended Data 1b,d (legend)
Figure 2b,d (plotted)
Figure 4e,i (plotted)
Figure 4d,g (legend)
Figure 5b,d,e (plotted)

Referee #2 (Remarks on code availability):

R2C16: I did a superficial assessment of the code. No new tools were developed or reported. Parameter and running settings of the open source tools used are insufficiently detailed to allow complete reproduction of results. This needs some attention as detailed in the comments above.

Reply: We thank the reviewer for taking the time to review the code and data. GitHub repositories and reconfigured downstream analysis scripts have been provided along with this revised manuscript (as described in R2C9) to address these points.

Referee #3 (Remarks to the Author):

This is an important study that sheds significant light on the evolution of metastases in non-small cell lung cancer. One of the unique aspects of this study is the longitudinal collection and follow-up of patients from diagnosis to death, in addition to the large number of tumor samples/patients from multiple organs and even from the same organ/area of metastases. The availability of clinical data including therapies received adds to the clinical relevance of the paper. The authors use standard well defined techniques to assess genomic evolutionary and metastatic migration patterns and use complementary technologies to confirm their conclusions. While there are limitations to limited spatial sampling, the multiple samples from one site when available in some patients tries to address that limitation. one limitation is the lack of characterization of the tumor microenvironment and the selective pressures that can have on metastatic evolution even without the acquiring of new mutations. The study does not look at some of the epigenetic mechanisms that can impact evolutionary changes. regardless overall I think this work sheds new knowledge on the metastatic process and the genetic processes that mediate that. The quality of the data and figures is great

Reply: We would like the that the reviewer for their comments. In this manuscript we focus on insights into the metastatic process that can be gleaned from reconstructing detailed tumour evolutionary histories that span patients' disease course, from early to late stage disease. We agree that the tumour microenvironment (TME) and non-genetic processes such as epigenetic modifications that are not evaluable using DNA sequencing data are equally

important to characterise to understand metastasis biology. In forthcoming work, we apply multiplexed imaging to the samples described in this manuscript to investigate the contribution of cellular and structural aspects of the TME to metastasis. The phylogenies and metastatic migration patterns inferred in this manuscript provide critical context for this investigation, for example, by allowing comparison of the TME features of metastases that seed other metastases to those that do not. Our intention is that further characterisation of this cohort with additional analytical platforms and technologies, such as the RRBS and spatial transcriptomics currently underway, will build upon the findings in this manuscript and advance the community's understanding of this highly complex and morbid disease.

We have added the following text to the **Discussion** to acknowledge the limitations highlighted by the reviewer (and also by reviewer 1):

The bulk WES performed in this study is high depth, enabling detection of low frequency mutations and SCNAs, but nonetheless may underestimate the subclonal diversity compared to sequencing technologies that examine the whole genome and/or single cells. In addition, with WES alone, the resolution of mutational processes is limited and the role of structural variants, extrachromosomal DNA, the tumour microenvironment and other non-genetic processes in metastasis cannot be investigated. Our follow-on study, TRACERx EVO (NCT05628376), will endeavour to address these limitations.

Referee #3 (Remarks on code availability):

I did not review the code

References

- 1 Al Bakir, M. *et al.* The evolution of non-small cell lung cancer metastases in TRACERx. *Nature* **616**, 534-542 (2023). <https://doi.org:10.1038/s41586-023-05729-x>
- 2 Grigoriadis, K. *et al.* CONIPHER: a computational framework for scalable phylogenetic reconstruction with error correction. *Nat Protoc* **19**, 159-183 (2024). <https://doi.org:10.1038/s41596-023-00913-9>
- 3 Gruber, M. *et al.* Growth dynamics in naturally progressing chronic lymphocytic leukaemia. *Nature* **570**, 474-479 (2019). <https://doi.org:10.1038/s41586-019-1252-x>
- 4 Böcker, S., Canzar, S., Klau, G. W. . *The generalized robinson-foulds metric*. (Springer Berlin Heidelberg, 2013).
- 5 El-Kebir, M., Satas, G. & Raphael, B. J. Inferring parsimonious migration histories for metastatic cancers. *Nat Genet* **50**, 718-726 (2018). <https://doi.org:10.1038/s41588-018-0106-z>
- 6 Roddur, M. S. *et al.* Characterizing the Solution Space of Migration Histories of Metastatic Cancers with MACH2. *bioRxiv* (2025).
- 7 Chakravarty, D. *et al.* OncoKB: A Precision Oncology Knowledge Base. *JCO Precis Oncol* **2017** (2017). <https://doi.org:10.1200/PO.17.00011>
- 8 Suehnholz, S. P. *et al.* Quantifying the Expanding Landscape of Clinical Actionability for Patients with Cancer. *Cancer Discov* **14**, 49-65 (2024). <https://doi.org:10.1158/2159-8290.CD-23-0467>
- 9 Frankell, A. M. *et al.* The evolution of lung cancer and impact of subclonal selection in TRACERx. *Nature* **616**, 525-533 (2023). <https://doi.org:10.1038/s41586-023-05783-5>
- 10 Hulse, M. *et al.* PRT3789 is a First-in-Human SMARCA2-Selective Degrader that Induces Synthetic Lethality in SMARCA4-Mutated Cancers. *Cancer Res* (2025). <https://doi.org:10.1158/0008-5472.CAN-25-1141>
- 11 Sanchez-Vega, F. *et al.* Oncogenic Signaling Pathways in The Cancer Genome Atlas. *Cell* **173**, 321-337 e310 (2018). <https://doi.org:10.1016/j.cell.2018.03.035>
- 12 Lucas, O. *et al.* Characterizing the evolutionary dynamics of cancer proliferation in single-cell clones with SPRINTER. *Nat Genet* **57**, 103-114 (2025). <https://doi.org:10.1038/s41588-024-01989-z>
- 13 Laks, E. *et al.* Clonal Decomposition and DNA Replication States Defined by Scaled Single-Cell Genome Sequencing. *Cell* **179**, 1207-1221 e1222 (2019). <https://doi.org:10.1016/j.cell.2019.10.026>
- 14 Pawlik, P. *et al.* Clone copy number diversity is linked to survival in lung cancer. *Nature* **646**, 190-197 (2025). <https://doi.org:10.1038/s41586-025-09398-w>
- 15 de Bruin, E. C. *et al.* Spatial and temporal diversity in genomic instability processes defines lung cancer evolution. *Science* **346**, 251-256 (2014). <https://doi.org:10.1126/science.1253462>
- 16 Gudem, G. *et al.* The evolutionary history of lethal metastatic prostate cancer. *Nature* **520**, 353-357 (2015). <https://doi.org:10.1038/nature14347>
- 17 Mermel, C. H. *et al.* GISTIC2.0 facilitates sensitive and confident localization of the targets of focal somatic copy-number alteration in human cancers. *Genome Biol* **12**, R41 (2011). <https://doi.org:10.1186/gb-2011-12-4-r41>
- 18 Tate, J. G. *et al.* COSMIC: the Catalogue Of Somatic Mutations In Cancer. *Nucleic Acids Res* **47**, D941-D947 (2019). <https://doi.org:10.1093/nar/gky1015>
- 19 McLaren, W. *et al.* The Ensembl Variant Effect Predictor. *Genome Biol* **17**, 122 (2016). <https://doi.org:10.1186/s13059-016-0974-4>
- 20 Karczewski, K. J. *et al.* The mutational constraint spectrum quantified from variation in 141,456 humans. *Nature* **581**, 434-443 (2020). <https://doi.org:10.1038/s41586-020-2308-7>
- 21 Jaganathan, K. *et al.* Predicting Splicing from Primary Sequence with Deep Learning. *Cell* **176**, 535-548 e524 (2019). <https://doi.org:10.1016/j.cell.2018.12.015>
- 22 Muinos, F., Martinez-Jimenez, F., Pich, O., Gonzalez-Perez, A. & Lopez-Bigas, N. In silico saturation mutagenesis of cancer genes. *Nature* **596**, 428-432 (2021). <https://doi.org:10.1038/s41586-021-03771-1>

- 23 Tokheim, C. & Karchin, R. CHASMplus Reveals the Scope of Somatic Missense Mutations Driving Human Cancers. *Cell Syst* **9**, 9-23 e28 (2019).
<https://doi.org:10.1016/j.cels.2019.05.005>